# Influence of resonant plasmonic nanoparticles on optically accessing the valley degree of freedom in 2D semiconductors

Tobias Bucher [1,2,3,10] ✉, Zlata Fedorova [1,2,3,10] ✉, Mostafa Abasifard [1,2,3], Rajeshkumar Mupparapu [2,3], Matthias J. Wurdack [1,2,3,4,5], Emad Najafidehaghani[6], Ziyang Gan [6], Heiko Knopf[2,3,7,8], Antony George [3,6], Falk Eilenberger [2,3,7,8], Thomas Pertsch [2,3,7,8], Andrey Turchanin [3,6,9] & Isabelle Staude [1,2,3,8]

The valley degree of freedom in atomically thin transition metal dichalcogenides, coupled with valley-contrasting optical selection rules, holds great potential for future electronic and optoelectronic devices. Resonant optical nanostructures emerge as promising tools for controlling this degree of freedom at the nanoscale. However, their impact on the circular polarization of valley-selective emission remains poorly understood. In our study, we explore a hybrid system where valley-specific emission from a molybdenum disulfide monolayer interacts with a resonant plasmonic nanosphere. Contrary to the simple intuition that a centrosymmetric nanoresonator mostly preserves the degree of circular polarization, our cryogenic experiments reveal significant depolarization of the photoluminescence scattered by the nanoparticle. This striking effect presents an ideal platform for studying the mechanisms governing light-matter interactions in such hybrid systems. Our full-wave numerical analysis provides insights into the key physical mechanisms affecting the polarization response, offering a pathway toward designing novel valleytronic devices.

As modern CMOS-based information technology is facing fundamental limits of further downscaling, novel materials providing additional electronic degrees of freedom, such as spin or valley-pseudospin, have become an active field of research[1–6]. The valley-pseudospin arises from multiple degenerate but inequivalent energy extrema in the bands of a crystal, in which excitons with distinct spin states can form and may be used to encode and process information[7–9]. In two-dimensional transition metal dichalcogenides (2D-TMDs), the broken inversion symmetry of the crystal structure and strong spin-orbit coupling lead to spin-valley locking.

[1]Institute of Solid State Physics, Friedrich Schiller University Jena, 07743 Jena, Germany. [2]Institute of Applied Physics, Friedrich Schiller University Jena, 07745 Jena, Germany. [3]Abbe Center of Photonics, Friedrich Schiller University Jena, 07745 Jena, Germany. [4]Department of Chemical Engineering, Stanford University, Stanford, CA, USA. [5]ARC Centre of Excellence in Future Low-Energy Electronics Technologies and Department of Quantum Science and Technology, Research School of Physics, The Australian National University, Canberra, ACT 2601, Australia. [6]Institute of Physical Chemistry, Friedrich Schiller University Jena, 07743 Jena, Germany. [7]Fraunhofer Institute for Applied Optics and Precision Engineering IOF, 07745 Jena, Germany. [8]Max Planck School of Photonics, Jena, Germany. [9]Jena Center for Soft Matter (JCSM), 07743 Jena, Germany. [10]These authors contributed equally: Tobias Bucher, Zlata Fedorova. ✉e-mail: tobias.bucher@uni-jena.de; zlata.fedorova@uni-jena.de

Consequently, energy-degenerate excitons with opposite spin are located at the K/K' points (or valleys) at the corners of the Brillouin zone[10–12] following valley-contrasting optical selection rules. The pronounced direct bandgap photoluminescence (PL) in the monolayer phase[13,14] facilitates efficient addressing and readout of the valley-degree of freedom in 2D-TMDs using circularly polarized light as depicted by the inset in Fig. 1. Photons with circular polarization $\sigma^+$ ($\sigma^-$) only interact with carriers in the valley K (K') and the valley-selective occupation can be quantified by the degree of circular polarization, DOCP $= (\mathcal{I}_{\sigma^+} - \mathcal{I}_{\sigma^-})/(\mathcal{I}_{\sigma^+} + \mathcal{I}_{\sigma^-})$, based on the valley-selective PL intensities $\mathcal{I}_{\sigma^\pm}$. However, despite an efficient control knob for the valley-pseudospin by means of circularly polarized light, the robust detection, manipulation and transport of the valley-pseudospin information remains challenging, mainly because of the short lifetime of valley-polarized excitonic states and a strongly reduced DOCP at room temperature due to phonon-assisted intervalley scattering.

During the last few years, the integration of 2D-TMDs with photonic nanostructures has gained immense popularity as an approach to address these challenges by enhancing and tailoring light-valley interaction at the nanoscale. For example, chiral plasmonic nanostructures and metasurfaces have been shown to favour one emission helicity over the other, suggesting a potential pathway to achieve room-temperature valley polarization by making use of superchiral nearfields[15,16]. However, the direct link between valley-selective excitation of the material and the observed farfield PL polarization contrast remains unclear, and definitive proof that valley polarization can respond to superchiral fields is still lacking.

In parallel, achiral nanostructures characterized by symmetric responses to the emission from both valleys emerged as promising tools for valley-based information processing. For instance, integrating 2D-TMDs with achiral dielectric metasurfaces demonstrated potential in controlling directionality, lifetime and spectral shape of the PL response[17,18]. Importantly, Liu and coworkers further demonstrated that the DOCP can be enhanced (equally for $\sigma^+$- and $\sigma^-$-polarized excitation) using Mie-resonant metasurfaces[18]. Other nanophotonic structures facilitated the generation of valley-polarized plasmon/photon-exciton polaritons opening new ways for valley control[19,20]. Of crucial importance for the development of on-chip valleytronic devices is directional routing of the valley-pseudospin

information. Routing has been reported in structures supporting guided modes with spin-momentum locking[2,3,19,21] and photonic crystals[22,23]. Despite these advancements, an apparent lack of experimental progress remains in the manipulation of valley-selective emission at the level of single nanoantennas. A notable divergence is observed in this context: The experimentally observed effects, though valuable, fell short of the expected efficiency in numerical simulations[24]. Additionally, other nanophotonic designs require the employment of electron beam excitation techniques[25,26], which significantly increases the technical complexity as well as the costs of the proposed schemes. Demonstrating efficient schemes for nanoantenna-based valley-routing employing widely accessible and integrated optical techniques therefore remains an open challenge. As the investigated nanophotonic architectures gain in complexity, the crucial prerequisite of precisely modelling the electromagnetic interaction between the valley-selective emitters and the resonant modes of these nanostructures becomes challenging and poses a limitation for further developments in this field. In light of this, we investigate a simplified model system as sketched in Fig. 1. Namely, we focus on a spherically symmetric (and hence achiral) gold nanoparticle (GNP) resonantly interacting with valley-specific emission from a monolayer of molybdenum disulfide (1L-MoS$_2$). By performing polarization-resolved cryogenic PL microscopy, we study the farfield polarization properties of this hybrid system under circularly-polarized optical pumping. Although it seems intuitive that an achiral nanoantenna should not affect the valley-pseudospin as it equally interacts with both K and K' valleys, we observe robust quenching of the DOCP in the farfield. Along with that there is no significant increase in the linear polarization components caused, for example, by the ellipticity of the GNP. This leads us to the conclusion that the observed effect predominantly represents a depolarization. This phenomenon raises fundamental questions about the underlying mechanisms of polarization effects within such hybrid systems. Specifically, what causes the observed depolarization? Is this a result of nearfield effects during the excitation phase, or does it occur during the re-emission of photons? Furthermore, which valleys are actually excited in the process? These are the key questions we aim to address in our investigation.

We compare our findings with a systematic numerical analysis of both excitation and emission aspects of the conducted experiments. We thereby isolate the main mechanisms that govern the observed depolarization. In particular, we find that the degree of circular polarization drops dramatically once the excitons are positioned only a few tens of nanometers away from the symmetry point leading to complete depolarization after averaging over contributions from excitons within the optical resolution limit. With this work, we not only aim to refine the existing simulation approaches for valleytronic devices, but also contribute to the deeper understanding of the rich physics of light-valley interactions at the nanoscale.

## Results

### Sample preparation

We prepared hybrid nanoparticle-on-substrate structures incorporating embedded 1L-MoS$_2$ by means of a simple spin-coating scheme (see Methods for a detailed description of all processes). Initially, we synthesized 1L-MoS$_2$ by chemical vapour deposition (CVD) on silicon/silicon-dioxide wafers[27]. The growth process yields a dense coverage of the substrate with high-optical quality 1L-MoS$_2$ crystals[28] reaching edge lengths of up to 60 μm. Next, we transferred the as-grown 1L-MoS$_2$ crystals onto a glass wafer by polymethylmethacrylate assisted wet-transfer[29]. Subsequently, we coated the sample with 15 nm silicon oxide using an optimized physical vapour deposition process[30]. The dielectric spacer layer prevents charge-transfer induced quenching of emission from 1L-MoS$_2$ by avoiding direct contact with the GNPs. Finally, by spin-coating we sparsely distributed monodispersed GNPs with an average size of (220 ± 15) nm on top of the prepared substrates.

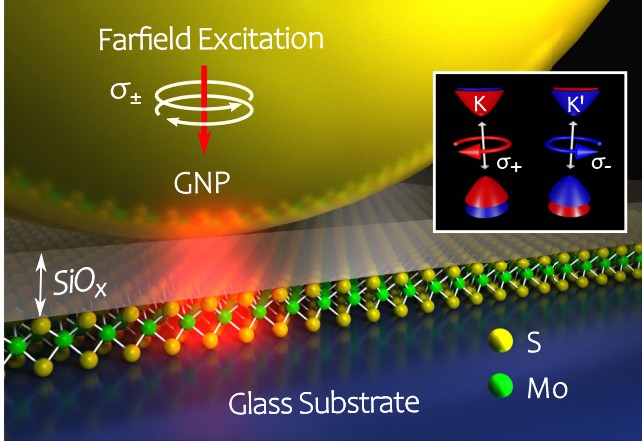

**Fig. 1 | Schematic of the investigated hybrid system excited by circularly polarized light.** A resonant gold nanoparticle (GNP, size 200 nm) is placed above a monolayer of molybdenum disulfide situated on a glass substrate. A thin dielectric spacer layer of silicon oxide (thickness 15 nm) was introduced prior to the nanoparticle deposition in order to prevent direct metal-TMD contact. Note that the crystal structure was scaled up for better visibility. The inset illustrates the valley-contrasting optical selection rules in monolayer TMDs.

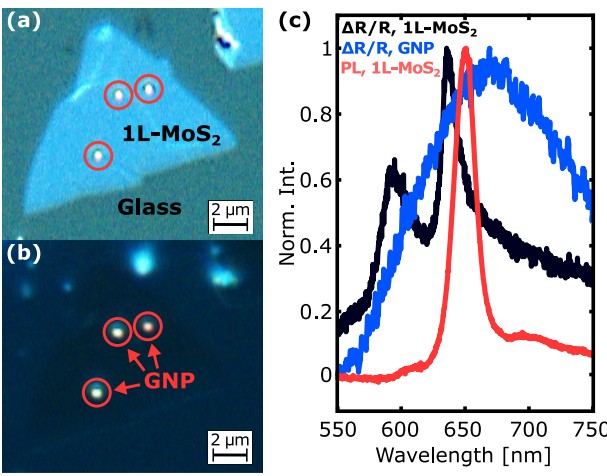

**Fig. 2 | Optical microscopy and spectroscopy pre-characterization. a** Optical brightfield and (**b**) darkfield microscope image of a prepared substrate incorporating embedded 1L-MoS$_2$ crystals and being decorated by several GNPs. The red circles indicate the positions of the GNPs overlapping with embedded 1L-MoS$_2$. **c** Cryogenic measurements ($T = 3.8$ K): differential reflectance spectra $\Delta R/R$ of embedded 1L-MoS$_2$ (black curve) and of isolated GNPs on a bare region of the substrate (blue curve), PL spectrum (red curve) of embedded 1L-MoS$_2$.

See Sec. S8 of the Supporting Information for details on the size distribution and shapes of GNPs deposited on a separate glass substrate using scanning electron microscopy. Figure 2a and b show an optical bright- and darkfield microscope image of a typical sample, respectively. The embedded 1L-MoS$_2$ crystals are decorated by several monodispersed GNPs which are visible in both images as bright spots as indicated by circles. Next, we characterized the optical properties of the prepared sample at cryogenic temperature. Figure 2c shows the measured differential reflectance spectrum $\Delta R/R$ (black curve) of the embedded 1L-MoS$_2$, which was measured using a tungsten-halogen white light source (see Methods). We observe two distinct peaks at 645 nm and 595 nm wavelength which are related to the A and B excitonic resonances formed at the direct bandgap in the K/K' points of the Brillouin zone[11]. The pronounced reflectance peaks indicate a large oscillator strength of the A and B excitons formed in our samples showing the high-optical quality of the embedded 1L-MoS$_2$. In this study, we focus on the emission from the embedded 1L-MoS$_2$ which is dominated by the lower-energetic A exciton as shown in the PL spectrum (red curve) measured from the same sample using a 561 nm wavelength continuous-wave excitation (see Methods). The observed PL peak centered at 650 nm wavelength is slightly Stokes-shifted with respect to the A exciton absorption peak and has a line width of about 60 meV which is comparable to the values of as-grown 1L-MoS$_2$ from our previous work[28] indicating the non-degradative character of the silicon oxide deposition process. For comparison, we show the averaged white light reflectivity spectrum (blue curve) of two separate GNPs deposited on a bare region of the coated substrate. The GNPs exhibit an electric dipolar resonance visible as broad peak centered at 670 nm wavelength. Despite a slight red-shift of the GNP reflectivity spectrum with respect to the A exciton energy of the embedded 1L-MoS$_2$ crystals, the broad width of the plasmonic resonance provides sufficient spectral overlap confirming the resonant character of the deposited GNPs at the central emission wavelength of the embedded 1L-MoS$_2$.

## Polarization-resolved cryo-PL measurements

We investigated the modification of the farfield degree of circular polarization (DOCP) of valley-specific emission from 1L-MoS$_2$ when scattered by a resonant GNP by performing polarization-resolved PL imaging at cryogenic temperature $T = (3.8 \pm 0.1)$ K. To achieve valley-

selective excitation of excitons, we pumped the sample near-resonantly at 633 nm wavelength and with $\sigma^+$ polarization. For excitation we used a 100x/0.9 NA objective and an average laser power of 50 μW (16 kW/cm$^2$ peak intensity). In detection, light was collected in reflection geometry using the same objective and a ($660 \pm 5$) nm wavelength bandpass. We further analyzed the collected light in a helical basis ($\sigma^\pm$) to obtain the valley-specific emission intensities ($\mathcal{I}_{\sigma^\pm}$) and calculated the farfield DOCP as DOCP $= (\mathcal{I}_{\sigma^+} - \mathcal{I}_{\sigma^-})/\mathcal{I}_{tot}$ where $\mathcal{I}_{tot} = (\mathcal{I}_{\sigma^+} + \mathcal{I}_{\sigma^-})$ is the total emission intensity. We have detailed the polarization control and notation of our experimental setup in Sec. S1 of the Supporting Information. In this notation, the DOCP of the incoming excitation laser and the measured PL emission will have equal sign. In order to investigate the modification of the farfield DOCP of emission from embedded 1L-MoS$_2$ by the resonant GNP, we measured $\mathcal{I}_{\sigma^\pm}$ as a function of position by confocal scanning microscopy as shown in Fig. 3a. For both $\mathcal{I}_{\sigma^+}$ and $\mathcal{I}_{\sigma^-}$, we observe a uniform distribution across the embedded 1L-MoS$_2$ crystal area where $\mathcal{I}_{\sigma^+}$ is about 6-times higher than $\mathcal{I}_{\sigma^-}$ as a signature of the induced exciton valley-polarization due to the valley-selective excitation. For 1L-MoS$_2$ without GNPs, we measured an average DOCP of $0.71 \pm 0.03$, which is consistent with values reported in the literature. The DOCP in 1L-MoS$_2$ has been shown to vary significantly from 32% up to 100%, even under the same experimental conditions due to sample-specific factors such as the dielectric environment, defects, and fabrication processes[31]. A different situation, however, is observed at the positions of the GNPs which are highlighted by circles. We find a significant local modulation of the valley-specific emission intensities with $\mathcal{I}_{\sigma^+}$ being slightly reduced and $\mathcal{I}_{\sigma^-}$ being noticeably enhanced with respect to the case without GNPs. To further quantify this effect we calculated $\mathcal{I}_{tot}$ and DOCP as shown in Fig. 3b and c, respectively. While there is only a small modulation of $\mathcal{I}_{tot}$ by the GNPs, we observe a strong reduction $\Delta_{DOCP} = 0.63 \pm 0.11$ from $0.71 \pm 0.03$ for embedded 1L-MoS$_2$ without GNP to $0.08 \pm 0.08$ with GNP. As confirmed by full-polarization resolved measurements, the emission from the embedded 1L-MoS$_2$ with GNP exhibits negligible linear polarization components (see Sec. S2 of the Supporting Information). Hence, the observed reduction in the DOCP is equivalent to a reduction of the total degree of polarization which can be clearly attributed to the presence of the GNPs. Interestingly, the cylindrical symmetry of the nanoparticle-on-substrate geometry considered in this work would suggest a preserved DOCP of emission which is in stark contrast to our experimental observations. In the following, we systematically analyze aspects of excitation and emission in our hybrid system and compare them with predictions from numerical simulations which allows us to isolate the mechanism leading to the observed reduction in the DOCP.

## Nearfield excitation polarization

The initial requirement to observe valley-specific emission from 1L-MoS$_2$ is a valley-selective excitation. By selection rules, the excitation rate of carriers in valleys K and K' is proportional to the intensity of the $\sigma^+$ and $\sigma^-$ polarized components of the external field, respectively. In 1L-MoS$_2$, the out-of-plane contributions from spin-forbidden dark excitons are negligible without strong external magnetic fields as shown by Robert et al.[32]. Therefore, mostly the in-plane components of the external fields, denoted by the superscript ||, contribute to the excitation of the material[33]. In equilibrium the local exciton densities $n_K(x,y)$ and $n_{K'}(x,y)$ will be proportional to $I_{\sigma^+}^{||}(x,y)$ and $I_{\sigma^-}^{||}(x,y)$, respectively. Note that, throughout this manuscript, $I$ quantifies near-field intensities, while $\mathcal{I}$ relates to farfield intensities. The induced degree of valley-polarization, $\eta = (n_K - n_{K'})/(n_K + n_{K'})$, is therefore proportional to the 2D-DOCP of the in-plane components of the excitation field, i.e. the DOCP of the in-plane components of the excitation field. In order to analyze the influence of the GNP on the valley-selective excitation of 1L-MoS$_2$, we numerically calculate the helical intensities, $I_{\sigma^\pm}^{||} \propto |E_{\sigma^\pm}^{||}|^2$, in a plane 15 nm below the GNP for a $\sigma^+$-polarized excitation beam. Figure 4a shows the resulting

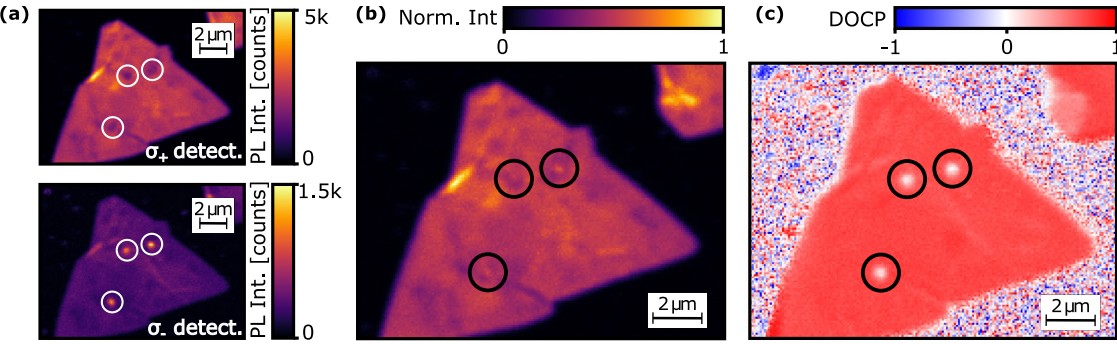

**Fig. 3 | Polarization-resolved photoluminescence microscopy. a** Measured confocal scans of valley-specific emission intensities $\mathcal{I}_{\sigma^\pm}$ from embedded 1L-MoS$_2$ decorated with monodispersed GNPs upon $\sigma^+$ excitation and collected through a 660 nm bandpass filter. **b** Total intensity $\mathcal{I}_{tot} = (\mathcal{I}_{\sigma^+} + \mathcal{I}_{\sigma^-})$ and (**c**) degree of circular polarization DOCP = $(\mathcal{I}_{\sigma^+} - \mathcal{I}_{\sigma^-})/\mathcal{I}_{tot}$ scans as calculated from the results shown in (**a**).

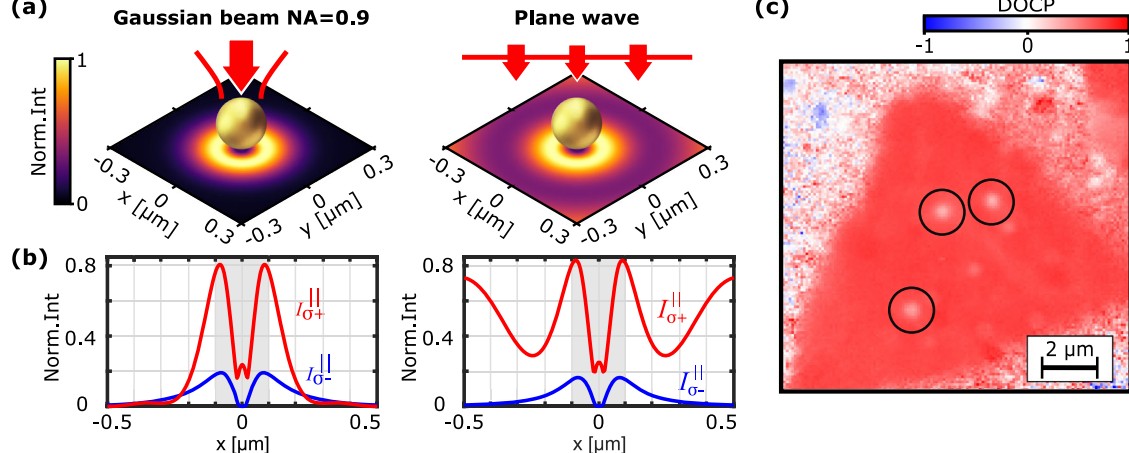

**Fig. 4 | Impact of the gold nanoparticle on the valley-selective excitation of 1L-MoS$_2$. a** Total intensity of the external in-plane nearfield in the plane of the monolayer upon $\sigma^+$ polarized excitation with a focused Gaussian beam (NA = 0.9) positioned at the center of the GNP (left) and a plane wave (right). **b** Respective individual contributions of $\sigma^+$ and $\sigma^-$ polarized components in dependence on the distance from the GNP's center along the $x$-axis ($y = 0$). The normalization is chosen such that the sum of both curves peaks to 1. Note that identical results are obtained for $\sigma^-$ polarized excitation up to exchanged labels $\sigma^\pm$, respectively. The shaded area marks the size of the GNP. **c** Measured DOCP of valley-specific emission from embedded 1L-MoS$_2$ decorated with monodispersed GNPs upon $\sigma^+$-polarized widefield illumination and collected through a 660 nm bandpass filter.

total (incident + scattered) in-plane intensities generated by a focused Gaussian beam (left) and a plane wave (right). In both cases, the maximum intensity is concentrated in a ring around the GNP, while directly below the GNP the intensity is lower but non-zero (see Fig. S6 of the Supporting Information for reference). We note that this pattern is characteristic when observing closely to a spherical GNP, within distances less than 50 nm, where near-field interactions with nanoantennas are typically prominent. Further insights into the formation of such a ring can be found in Sec. S5 of the Supporting Information. The individual contributions of $\sigma^+$ and $\sigma^-$ polarized field components to these intensities are plotted in Fig. 4b, represented by red and blue curves respectively, against the distance from the GNP's center along the $x$-axis ($y = 0$). For both excitation conditions, the GNP induces the cross-polarized intensity component $I_{\sigma^-}^{\|}$ peaking at the projected GNP's edge (gray shaded region) and diminishing with increasing distance. Notably, when normalized to the maximum of the total in-plane intensity, this cross-polarized component is slightly larger for the focused beam. Reference simulations of a tightly focused Gaussian beam without a GNP (see Fig. S6 of the Supporting Information) reveal that $I_{\sigma^-}^{\|}$ remains nearly zero, emphasizing that the slight increase of this component observed in the presence of a GNP is not due to high NA itself. Additionally, for the focused beam the intensity of $\sigma^+$-polarized field drops faster than $\sigma^-$, resulting in their curves intersecting at

$|x| \approx 0.22\,\mu m$. In contrast for a plane wave, the total field at larger distances from the GNP is dominated by the incident field characterized by pure $\sigma^+$ polarization. The resultant 2D-DOCP maps for all of the discussed cases are provided in Sec. S6 of the Supplementary Information. Hence, the excitation scheme is expected to influence the observed reduction in DOCP of emission from 1L-MoS$_2$ mediated by the GNP. We therefore repeated our cryo-PL imaging experiments using wide-field illumination (excitation NA ≈ 0) mimicking the plane wave excitation. Figure 4c shows the respective measured DOCP image of PL from the same 1L-MoS$_2$ sample as shown before and measured with an average excitation power of 200 µW (40 W/cm$^2$ peak intensity). Indeed, we observe a less pronounced local reduction of $\Delta_{DOCP} = 0.39 \pm 0.06$ from $0.72 \pm 0.02$ for embedded 1L-MoS$_2$ without GNP to $0.33 \pm 0.04$ with GNP. For a quantitative prediction of the $\Delta_{DOCP}$ for both excitation schemes we have to relate the degree of valley polarization $\eta$ to the DOCP of emission in the farfield. In order to so solely based on the numerically calculated excitation nearfield, we initially neglect scattering of the emission by the GNP. Additionally, we take into account the finite optical resolution in our experiments.

In case of the widefield excitation scheme, we use a Gaussian point spread function to obtain the averaged helical excitation intensities $\tilde{I}_{\sigma^\pm}(x,y) \propto \iint G(\xi - x, \eta - y) I_{\sigma^\pm}^{\|}(\xi, \eta)\,d\xi d\eta$, where $G(x,y)$ is a Gaussian weight function given by Equation (3) from the Methods section.

For confocal scanning the resolution limit is encoded in the finite beam size. Importantly, the beam position relative to the GNP changes during the scanning such that the calculated nearfield intensity distribution $I_{\sigma\pm}^{\parallel}(x_b, x, y)$ and the respective exciton density distribution $n_{K/K'}(x_b, x, y)$ become dependent on the displacement distance $x_b$ of the beam from the center of the GNP. Here, it is sufficient to consider a displacement only along one axis due to cylindrical symmetry of our system. Consequently, we obtain the averaged helical intensities by integrating over the intensity spot for each beam position $\tilde{I}_{\sigma\pm}(x_b) \propto \iint I_{\sigma\pm}^{\parallel}(x_b, \xi, \eta) \, d\xi d\eta$.

Finally, we estimate the DOCP of PL in the farfield for both excitation schemes by substituting $\tilde{\mathcal{I}}_{\sigma\pm} \longrightarrow \tilde{I}_{\sigma\pm}$ in Equation (5) and evaluating it for $x_b = 0$. We obtain a predicted reduction of $\Delta_{\text{DOCP}} = 0.18 \pm 0.04$, i.e. from $0.72 \pm 0.02$ (measured) without to $0.54 \pm 0.02$ (calculated) with GNP for plane wave excitation, and $\Delta_{\text{DOCP}} = 0.45 \pm 0.06$, i.e. from $0.71 \pm 0.03$ (measured) without to $0.26 \pm 0.03$ (calculated) with GNP, for Gaussian beam excitation. Our calculations based on excitation effects are consistent with the experimental observation of a larger reduction in DOCP for the case of the confocal as compared to the widefield excitation scheme. However, the experimentally observed changes of $\Delta_{\text{DOCP}} = 0.39 \pm 0.06$ and $\Delta_{\text{DOCP}} = 0.63 \pm 0.11$ remain systematically larger indicating additional contributing factors.

Note, that in this simple estimation we have not assumed any exciton diffusion which is known to potentially impact the electromagnetic interaction of emitters in 2D-TMDs with optical nanoresonators[34]. In our system, however, we expect a rather weak impact of the nearfield of the GNP on the diffusion properties of the excitons in 1L-MoS$_2$ as first, we observe no significant modulation of the total cryo-PL intensity at the positions of the GNPs, and second, we find the depolarization effect to be robust across all our samples including different spacer thicknesses (see Sec. S4 of the Supporting Information) and GNPs with naturally varying size and topography. Nevertheless, the observed DOCP of emission from 1L-MoS$_2$ with GNP cannot be explained by excitation effects alone but additional factors need to be considered as discussed next.

## Farfield emission polarization

By symmetry, the nanoparticle-on-substrate geometry is expected to largely preserve the circular polarization of valley-selective emission in the forward and backward scattered farfield. This can be understood by modelling emission using dipoles exhibiting circularly polarized farfield components. For this purpose, different types of emitters are discussed in literature with focus on their symmetry properties[35–37] or different multipolar coupling behaviour[38,39]. The prevalent model used

in the context of 2D-TMDs is the rotating electric dipole[2,3,5,18,19,26,40], $\mathbf{p}_{K/K'} = \mathbf{p}_x + / - i\mathbf{p}_y$, where the spin or valley index is associated with the fixed phase sign. On the right side of Fig. 5a we show the farfield intensity distribution and respective DOCP of a single counterclockwise rotating electric dipole. Due to its fixed axis of rotation, $\mathbf{p}_K$ approximately emits light of one helicity into one half-space and the opposite helicity in the other half-space matching the PL polarization properties of valley-selective excitons in bare 1L-TMDs (see Sec. S3 of the Supporting Information). When placed below a resonant GNP, the rotating electric dipole is expected to induce a respective mirror rotating dipole with equal spin (as both linear dipole components of the mirrored rotating dipole will experience the same $\pi$ phase flip). Consequently, the PL polarization properties of the hybrid system would be similar to that of a bare rotating electric dipole.

However, this is only the case for a single rotating dipole placed on the symmetry axis of the hybrid system. For a finite displacement of the rotating dipole from the projected center of GNP, the cylindrical symmetry of the system is broken. To investigate the effect of the dipole displacement on the farfield polarization of the hybrid system, we calculate the helical farfield intensities $\mathcal{I}_{\sigma\pm}^K(x_e, \mathbf{u}(\theta, \phi))$ emitted by a rotating dipole $\mathbf{p}_K$ that is displaced by $x_e$ along the $x$-axis where the unit-vector $\boldsymbol{u}$ denotes the farfield direction $(\theta, \phi)$ in spherical coordinates (see Methods section for details). In our calculations we assume that the emission from 1L-MoS$_2$ is both spatially and temporally incoherent. By integrating over a numerical aperture of 0.9, we obtain the angle-averaged intensities $\mathcal{I}_{\sigma\pm}^K(x_e)$ as plotted on top of Fig. 5b. Similar to the previously analyzed nearfields, we observe the emergence of cross-polarized intensity components $\mathcal{I}_{\sigma-}^K(x_e)$ peaking at about $|x_e| \approx 60$ nm. Below, in Fig. 5b we show the resulting total farfield intensity (orange curve) and the farfield DOCP (green curve) as functions of the displacement distance $x_e$. Here, all the farfield intensities are normalized to the emission of a rotating electric dipole without GNP, i.e. $x_e \rightarrow \infty$.

As expected from symmetry, we obtain a high farfield DOCP for dipoles below the center of the GNP ($x_e = 0$). However, we find that the DOCP reduces radically even for small displacements below the footprint of the GNP (shaded region) and reaches a minimum of about 0.04 for $x_e = 35$ nm. Simultaneously, the relative intensity contribution of dipoles below the center of the GNP is significantly lower than for emitters located below the edge of the GNP. Our findings therefore highlight that the overall farfield polarization of the hybrid system with an ensemble of spatially separated valley excitons can differ significantly from the intuition based on a single rotating electric dipole below the center of the GNP.

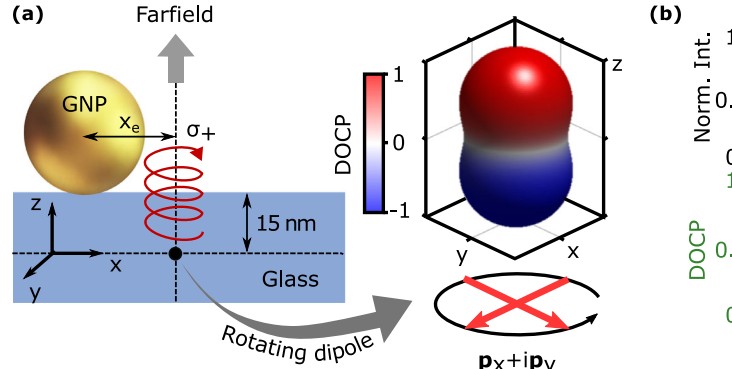
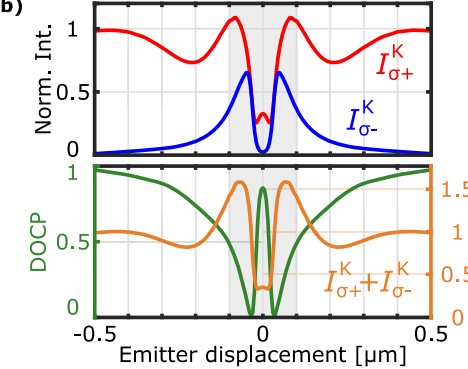

**Fig. 5 | Impact of the gold nanoparticle on the photoluminescence polarization from distributed emitters. a** Sketch of the simulated nanoparticle-on-substrate geometry showing the position of the rotating electric dipole in a plane 15 nm below and displaced by a distance $x_e$ from the projected center of the GNP. The inset shows the calculated farfield radiation pattern of a rotating electric dipole $\mathbf{p}_K$, the color encodes the respective DOCP. **b** Top: Calculated angle-averaged farfield

intensities $\mathcal{I}_{\sigma+}^K$ (red line) and $\mathcal{I}_{\sigma-}^K$ (blue line) emitted by an incoherent rotating dipole $\mathbf{p}_K$ as a function of the displacement distance. Bottom: The corresponding DOCP (green curve) and total intensity (orange curve) of the integrated farfield. The farfield intensity is normalized to the intensity obtained without a GNP. Note that the results for a dipole with opposite spin, $\mathbf{p}_{K'}$, can be obtained by exchanging the labels $\sigma\pm$, respectively. The shaded areas mark the size of the GNP.

**Full model for optically addressing the valley degree of freedom**
In this section, we provide a quantitative model that describes the observable PL intensity and polarization of the hybrid system in the farfield by combining excitation and emission effects as well as incorporating the finite optical resolution. For the hybrid system, we recall that a $\sigma^+$ polarized excitation beam leads to non-zero exciton densities in both valleys K/K' proportional to the in-plane intensity distribution of the total field, $n_{K/K'} \propto I^{\parallel}_{\sigma^{+/-}}$. Similarly, we have discussed above that excitons located in the valley K (or K') at a distance $x_e$ from the GNP's center lead to both $\sigma^+$ and $\sigma^-$ polarized PL intensities in the farfield which we denoted as $\mathcal{I}^K_{\sigma^\pm}(x_e)$ (or $\mathcal{I}^{K'}_{\sigma^\pm}(x_e)$), respectively, where the farfield intensities from emitters in opposite valleys are obtained as $\mathcal{I}^{K'}_{\sigma^\pm}(x_e) = \mathcal{I}^K_{\sigma^\mp}(x_e)$. Hence, the overall PL emission of the hybrid system in a helical farfield basis, $\mathcal{I}_{\sigma^\pm}$, is a sum of contributions from both valleys, K and K', weighted by their respective excitation rates. Equation (2) of the Methods section expresses $\mathcal{I}_{\sigma^\pm}$ for the case of the widefield excitation scheme where the excitation beam is fixed. In order to account for the finite optical resolution we convolute the local contributions $\mathcal{I}_{\sigma^\pm}(x, y)$ with a Gaussian point spread function where the Gaussian width matches the optical resolution of the experimental setup (see Equation (3) in Methods).

For the case of confocal microscopy, the excitation beam is not fixed and the exciton density distribution has to be calculated for each position of the excitation beam separately. The farfield observable PL contributions are then described by Equation (4) of the Methods section.

## Discussion

Ultimately, we compare the calculated farfield DOCP with our measured results for both excitation schemes in Fig. 6. Here, we plotted the cross-sections of the measured DOCP of emission from 1L-MoS$_2$ upon $\sigma^+$ excitation across the GNPs both for wide-field illumination (blue circles) and confocal scanning (red circles). The cross-sections and error corridors (shaded regions) are obtained as the mean value and the standard deviation, respectively, of several GNPs and cross-section directions across each GNP. For our sampling we have used the DOCP shown in Fig. 3c (scanning) or Fig. 4b (wide field) and compared the cross-sections along x- and y- axes of three measured nanoparticles on a single flake (in total 6 samples per measurement scheme).

For comparison, we also plotted the numerical prediction of the farfield DOCP according to Equation (5) (see Methods) for plane wave excitation (blue solid curve) as well as for confocal Gaussian beam excitation (red solid curve). Our numerical calculations closely agree with the experimental results indicating that the presented model effectively captures the dominating physical effects leading to the observed reduction of the DOCP of emission from 1L-MoS$_2$. Additionally, in Sec. S7 of the Supporting Information, we provide detailed computations of the expected values for all Stokes components and the total degree of polarization. These results further validate that our model accurately predicts the overall depolarizing effect of the nanoparticle, in strong agreement with experimental observations.

Interestingly, the accuracy of our numerical model varies between the two measurement schemes. While the computed DOCP reduction for confocal scanning is at the upper error limit of the experimental curve, we observe a very close theory-experiment match for the widefield measurement. One important difference between these schemes is the optical energy density which is inevitably higher in the case of confocal scanning due to shorter exposure times. We attribute the discrepancy in the confocal scanning regime to two main factors. First, our model does not account for exciton-exciton interactions, which are expected to intensify at higher optical energy densities. Second, we do not consider finite exciton momenta induced by the in-plane wavevector components of light scattered from a GNP and of a tightly focused laser beam. This non-zero momentum can result in a spatial redistribution of excitons and is known to lead to increased

intervalley scattering rates[41]. Recently, Salzwedel et al.[42] and Greten et al.[43] presented a framework based on a self-consistent Maxwell-Bloch theory able to describe excitonic effects in similar plasmonic-TMD hybrid systems. Their works, exploring different coupling regimes, offers a valuable testbed for validating our proposed model.

In summary, we investigated the complex polarization behaviour of nanoscopic hybrid systems consisting of 2D-TMDs interacting with resonant nanostructures. For our hybrid model system, we used a spherical GNP as a probe to systematically investigate its impact on the emission polarization behaviour of a 2D-TMD crystal below. We reported a strong and robust reduction in the DOCP of PL from 1L-MoS$_2$ mediated by the GNPs which is in contrast to the expectation that a cylindrically symmetric system predominantly conserves the circular polarization in the farfield in the direction out of the substrate.

Our observation highlights the need for a predictive model able to quantitatively describe the collective response of such nanoscopic hybrid systems. In this work, we have conceived and tested a physical model that provides a quantitative description for the farfield PL properties of (ensembles of) valley-selective emitters in 2D-TMDs when in proximity to resonant nanostructures. It gives deeper insights into the complex polarization behaviour of nanoscopic hybrid systems by discerning the individual contributions discussed above which serves as a crucial prerequisite to accurately design and optimize nanoscopic hybrid systems for valleytronic applications. While our work specifically shows the challenges and limitations for the realization of valleytronic devices on the basis of resonant plasmonic nanostructures, it hints at pathways to circumvent these limitations. These might range from optimizing the valley selective excitation of 2D-TMDs in the vicinity of resonant nanostructures by structured light excitation techniques or exploring alternative material platforms and resonance types for tailoring the collective emission response to nanostructuring of 2D-TMDs for precise emitter position control for the realization of functional nanoscopic valleytronic devices.

## Methods

### CVD growth and embedding of 1L-MoS$_2$
Single crystals of 1L-MoS$_2$ with high optical quality were synthesized on silicon wafers with 300 nm thermal oxide (Sil'tronix, root mean square (RMS) roughness < 0.2 nm) by a modified chemical vapor deposition process in which Knudsen cells were employed for the delivery of precursors[27,28]. The grown 1L-MoS$_2$ single crystals were characterized initially using optical microscopy and Raman spectroscopy and transferred onto a glass substrate using a poly(methyl methacrylate) assisted wet-transfer process[29]. Subsequently, the sample was coated by a thin film of silicon oxide using an adapted physical vapour deposition process[30] (Buhler SyrusPro 1100) using a relatively low deposition rate of 0.5 nm/s and 15 nm target thickness.

### Gold nanoparticle deposition
Initially, the purchased GNP suspension (Merck/Sigma Aldrich, particle size 200 nm) was ultrasonicated for 5 min and then 20 μL of the suspension were diluted with 60 μL of isopropanol to reach a particle concentration of ~ 4.7 × 10$^8$ particles/mL. Subsequently, the diluted suspension was spin-coated onto the sample at 1000 rpm with 500 round/s$^2$ acceleration for 20 s. This process resulted in a homogeneous distribution of mostly isolated GNPs on the substrate.

### Optical experiments at cryogenic temperatures
Optical experiments were conducted at cryogenic temperatures ($T = (3.8 \pm 0.1)$ K) using a closed-cycle helium cryostation (Montana Instruments s50) equipped with a high numerical aperture optical access (Zeiss 422392-9900-000, EC Epiplan-Neofluar 100x/0.90 DIC Vak objective) in reflection geometry. A non-polarizing 30(R):70(T) plate beam splitter (Chroma) was utilized for all our cryogenic measurements. A sketch of our optical setup as well as a detailed

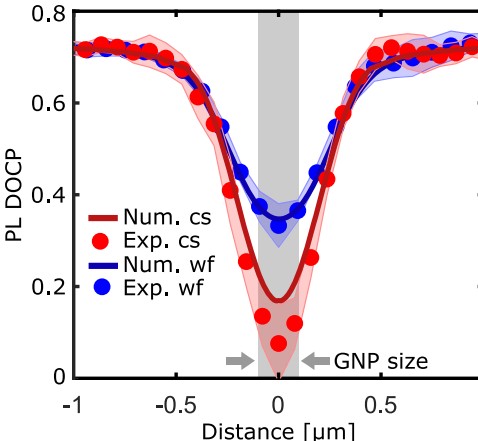

**Fig. 6 | Quantitative modelling of the collective photoluminescence polarization.** Comparison of the spatial lines profiles of the calculated (solid curves) and measured (circles) DOCP as a function of the radial distance. Blue and red color corresponds to wide field (wf) and confocal scanning (cs) measurement settings, respectively. The shaded area marks the size of the GNP.

characterization of the polarizing effect of the utilized optical components are provided in Sec. S1 of the Supporting Information. For white light spectroscopic measurements, a stabilized tungsten-halogen white light source was used. Reflectance spectra were measured from 1L-MoS₂ as well as from isolated GNPs by limiting the illumination area using an iris aperture. The differential reflectance spectra in Fig. 2c were then obtained by referencing to the reflectance of a bare region of the coated substrate via

$$\Delta R/R = \frac{R_{\mathrm{MoS_2/GNP}} - R_{\mathrm{Subs}}}{R_{\mathrm{Subs}}}. \tag{1}$$

For photoluminescence spectroscopic measurements, the sample was excited using a 561 nm wavelength continuous-wave diode pumped solid-state laser. For polarization-resolved photoluminescence imaging, the sample was pumped near-resonantly at 633 nm wavelength using a continuous wave helium-neon gas laser. The polarization of the excitation beam was prepared by a linear polarizer (Thorlabs LPVISC050-MP2) and a quarter-wave phase plate (Thorlabs WPMQ05M-633) by monitoring the degree of circular polarization of the collimated excitation beam before entering the objective. In detection, the polarization was analyzed by a combination of a superachromatic quarter-wave phase plate (Thorlabs SAQWP05M-700) and a linear polarizer (Thorlabs LPVIS100-MP2). Note that in experiments we do not have direct access to the polarization state of the excitation beam in the focal plane. Hence, we numerically calculated the 2D-DOCP for a tightly focused Gaussian beam (see. Sec. S6 of the Supporting Information) which is nearly 1. Note further that the out-of-plane component for a tightly focused beam is non-zero, however, it does not contribute significantly to the absorption by 1L-MoS₂ as discussed in the Results section. For confocal scanning measurements, the excitation beam with an average power of 50 μW was focused to a diffraction limited spot with a diameter of $d = 2\lambda/(\mathrm{NA} \cdot \pi) \approx 0.45\,\mu\mathrm{m}$ and a peak intensity of 16 kWcm⁻². Lateral scanning was performed using two piezoelectrical nanopositioners moving the sample. This allows for stable conditions of the focused excitation with a fixed degree of circular polarization. The collected emission was imaged onto an EMCCD camera (Andor iXon 897 Ultra) and integrated over an area of $3 \times 3$ pixels ($= 48 \times 48\,\mu\mathrm{m}^2$ on camera chip) which relates to an area of $270 \times 270\,\mathrm{nm}^2$ in the conjugated sample plane. For wide-field measurements, an additional lens was introduced to focus the excitation beam with an average power of 200 μW onto the back-focal

plane of the objective. This results in a beam size of $\approx 25\,\mu\mathrm{m}$ on the sample and a peak intensity of 40 Wcm⁻². In all imaging experiments, the collected emission was filtered by a 650 nm longpass filter to block the laser light collected from the sample in reflection. An additional 660 nm bandpass filter was used to limit the detection to a spectral band as close to the A exciton photoluminescence peak as possible.

## Numerical simulations

For the numerical analysis of our hybrid system, we have used a commercial finite-difference time-domain solver (Lumerical FDTD solutions). Perfectly matched layers were applied at each boundary of the simulation area. The nanoparticle was modelled as a gold sphere with a radius of 100 nm lying on a glass that fills the lower halfspace of the FDTD domain. We did not differentiate between the glass substrate and the capping oxide layer, considering both as a homogeneous medium with the refractive index of $n = 1.5$. The remaining space is filled with air. The presence of the monolayer was neglected in our simulations. The optical properties of the materials were taken from the default material database in Lumerical, namely, Handbook of Optical Constants of Solids I - III by E. Palik.

Focusing on the excitation aspect, we first computed the in-plane nearfield components $E_x(x,y)$ and $E_y(x,y)$ at the position of the monolayer, 15 nm below the nanoparticle. The GNP was illuminated under normal incidence by either a plane wave or a Gaussian beam in a thin lens approximation with NA = 0.9. For both excitation sources we have set circular polarization and center wavelength of 633 nm. The mesh override sections with the mesh size of 5 nm were applied for the areas around the beam focus and nanoparticle to achieve finer resolution. In case of a plane wave, we computed the local valley exciton density as $n_K(x,y) \propto I_{\sigma^+}^{||}(x,y)$ and $n_{K'}(x,y) \propto I_{\sigma^-}^{||}(x,y)$, where $I_{\sigma^{\pm}}^{||} \propto |E_{\sigma^{\pm}}|^2 = |E_x \pm iE_y|^2$. To reproduce the result of confocal scanning we have displaced the focal spot of the Gaussian beam from the nanoparticle center along the $x$-axis and computed the corresponding $n_{K/K'}(x_{\mathrm{b}},x,y)$ as a function of the displacement distance $x_{\mathrm{b}}$.

In the second set of simulations we studied the radiation farfield polarization from a single emitter in 1L-MoS₂ scattered by the GNP. For that we mimicked an emitter from K or K' valley as a pair of perpendicular electric dipoles with 90° or -90° phase shift, respectively. The simulations were performed for different GNP-emitter distances $x_{\mathrm{e}}$ as shown in Fig. 5a. High sensitivity of the interaction between the GNP and the dipole emission on the relative distance yields high sensitivity to the mesh size. Performing the convergence test, we chose 3 nm mesh size around the dipole and the nanoparticle as a good trade-off between the simulation accuracy and computation time. The nearfield was extracted from a horizontal plane right above the nanoparticle and propagated by 1 m to the farfield using the built-in Lumerical functions. Each FDTD simulation provides the field components in spherical coordinates $\{E_\theta(x_{\mathrm{e}},\mathbf{u}), E_\phi(x_{\mathrm{e}},\mathbf{u}), E_r(x_{\mathrm{e}},\mathbf{u})\}$, where $x_{\mathrm{e}}$ is the emitter displacement and $\mathbf{u}(\theta,\phi) = (\sin\theta\cos\phi, \sin\theta\sin\phi, \cos\theta)^{\mathrm{T}}$, $\theta \in [0,\pi]$, $\phi \in [0,2\pi]$ denotes the emission direction in terms of a direction unit vector as defined in Lumerical FDTD Solutions[44]. An angular filter was applied to the resulting field restricting the accepted emission angles to fit the finite numerical aperture of our objective ($\sin\theta <$ NA, NA = 0.9). Next, we transformed the filtered field into the helical basis $E_{\sigma^{\pm}} = E_\theta \pm iE_\phi$. The radial field component $E_r$ is not considered because it vanishes in the farfield. The corresponding intensities are given by $\mathcal{I}_{\sigma^{\pm}}^K(x_{\mathrm{e}},\mathbf{u}) = |E_{\sigma^{\pm}}^K(x_{\mathrm{e}},\mathbf{u})|^2$, where we have added the valley index $K$ denoting the rotating dipole used as nearfield source. In the following we assume that emission from each point in space is temporally incoherent because we observe the system on timescales ~ 0.1 s incommensurately longer than the exciton coherence times ~ 0.1 ps[45]. Consequently, there is no fixed phase relation between the radiation into different directions and we average by integrating the intensities over all emission angles $\mathcal{I}_{\sigma^{\pm}}^K(x_{\mathrm{e}}) \propto \iint \mathcal{I}_{\sigma^{\pm}}^K(x_{\mathrm{e}},\mathbf{u}(\theta,\phi))\sin\theta\,d\theta d\phi$ (see Ref. 44 for details on the farfield integration). Note that from the

symmetry considerations, it is sufficient to consider just one valley, because the intensity contributions of the opposite one will be swapped, i.e. $\mathcal{I}_{\sigma-}^{K'}(x_e) = \mathcal{I}_{\sigma+}^{K}(x_e)$ and $\mathcal{I}_{\sigma+}^{K'}(x_e) = \mathcal{I}_{\sigma-}^{K}(x_e)$.

Having quantified both the excitation and emission processes we combined them to compare with the experimental results. In case of the widefield experiment, the farfield intensities of $\sigma^{\pm}$-polarized light emitted from every point $(x,y)$ is given by

$$\mathcal{I}_{\sigma^{\pm}}(x,y) = n_K(x,y) \cdot \mathcal{I}_{\sigma^{\pm}}^{K}\left(\sqrt{x^2+y^2}\right) + n_{K'}(x,y) \cdot \mathcal{I}_{\sigma^{\pm}}^{K'}\left(\sqrt{x^2+y^2}\right), \quad (2)$$

where we have leveraged the rotational symmetry of our system around the GNP's center (0,0).

Next, we took into account the finite resolution of our optical system by applying the Gaussian point spread function $\tilde{\mathcal{I}}_{\sigma^{\pm}}(x,y) = \iint G(\xi-x, \eta-y)\mathcal{I}_{\sigma^{\pm}}(\xi,\eta)\,\mathrm{d}\xi\mathrm{d}\eta$, where

$$G(x,y) = \frac{1}{2\pi\Sigma^2}\exp\left(-\frac{x^2+y^2}{2\Sigma^2}\right) \quad (3)$$

with $\Sigma = 174$ nm, as extracted from the widefield experimental data.

In case of the confocal scanning configuration, the local exciton densities $n_{K/K'}(x_b, x, y)$ become dependent on the displacement $x_b$ of the beam from the projected center of the GNP. In the experiment, we capture the PL signal that passes through the circular detection polarizer by integrating over a spot on the camera for each beam position. Likewise, in simulations we obtain $\sigma^{\pm}$ PL intensities by integrating over the corresponding farfield intensity distributions for each $x_b$

$$\tilde{\mathcal{I}}_{\sigma^{\pm}}(x_b) \propto \iint \left[ n_K(x_b, \xi, \eta) \cdot \mathcal{I}_{\sigma^{\pm}}^{K}\left(\sqrt{\xi^2+\eta^2}\right) + n_{K'}(x_b, \xi, \eta) \cdot \mathcal{I}_{\sigma^{\pm}}^{K'}\left(\sqrt{\xi^2+\eta^2}\right) \right] \mathrm{d}\xi\mathrm{d}\eta \quad (4)$$

where $\xi$ and $\eta$ are two spatial coordinates used in the simulations. Note, that there is no need in applying the Gaussian point spread function because the optical resolution is already accounted for by integrating over the finite size of the focal spot. Finally, the anticipated DOCP of the PL for both experiments that is shown in Fig. 6 was computed as

$$\mathrm{DOCP} = \frac{\tilde{\mathcal{I}}_{\sigma^+} - \tilde{\mathcal{I}}_{\sigma^-}}{\tilde{\mathcal{I}}_{\sigma^+} + \tilde{\mathcal{I}}_{\sigma^-}} \times \left\langle \mathrm{DOCP}_{\mathrm{PL}}^{\mathrm{bare}} \right\rangle \quad (5)$$

where $\left\langle \mathrm{DOCP}_{\mathrm{PL}}^{\mathrm{bare}} \right\rangle$ was the averaged experimentally observed DOCP of PL from embedded 1L-MoS$_2$ without a GNP.

## Data availability

The raw data that supports this study is available in the Supplementary Information and has also been deposited in the figshare database under accession code https://doi.org/10.6084/m9.figshare.27312498.

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

## Acknowledgements
This work was funded by the Deutsche Forschungsgemeinschaft (DFG, German Research Foundataion) - CRC/SFB 1375 NOA "Nonlinear Optics down to Atomic scales" (Project number 398816777), EXC 2051 (Project number 390713860), International Research Training Group 2675 "META-Active" (project number 437527638), and the Emmy Noether Program (Project number STA 1426/2-1). F.E. acknowledges the Bundesministerium für Bildung und Forschung (BMBF, Federal Ministry of Education and Research) support via NanoScopeFutur-2D (Project Number 13XP5053A). M.J.W. acknowledges support by the Australian Research Council (ARC) Centre of Excellence Grant No. CE170100039 and Schmidt Science Fellows, in partnership with Schmidt Sciences and Rhodes Trust. We thank M. Younesi and M. Rikers for providing the scanning electron micrographs, P. Lohbeck for helping with characterizing the polarizing properties of the optical components, and J. Gour, L. Valencia Molina and R. Schlegel for the preparation and coating of the substrates. We also thank U. Peschel, C. Rockstuhl and I. Fernandez-Corbaton for fruitful discussions on the symmetry properties of chiral emitters. Further, we thank A. Knorr, R. Salzwedel and L. Greten for fruitful discussions on dipole-dipole interactions in exciton-plasmon based systems.

## Author contributions
T.B., Z.F., R.M., and I.S. conceived the research idea and designed the experiments; T.B., Z.F., and M.A. prepared the samples with the help of E.N., Z.G., H.K., A.G.; E.N., Z.G., A.G. performed the crystal growth under the supervision of A.T.; H.K. performed the thin-film integration under the supervision of F.E.; T.B. Z.F., and M.A. performed the experiments under the supervision of T.P. and I.S.; Z.F. and M.A. performed the numerical simulations; T.B. and Z.F. performed the data analysis and developed the physical model with the help of M.J.W.; T.B. and Z.F. prepared the figures and wrote the manuscript with the help of all authors; I.S. supervised this project.

## Funding

## Competing interests
The authors declare no competing interests.
