## [Transparent Peer Review file · Nature Communications]

Influence of resonant plasmonic nanoparticles on optically accessing the valley degree of freedom in 2D semiconductors

Corresponding Author: Mr Tobias Bucher

Version 0:

Reviewer comments:

Reviewer #1

(Remarks to the Author)
See the attached file.

Reviewer #2

(Remarks to the Author)

The manuscript deals with the possible manipulation of valley polarization in a MoS₂ monolayer using resonant plasmonic nanospheres. Measurements reveal a lack of polarization in the emitted light below the nanosphere. The authors investigate this problem numerically decomposing it into excitation and emission, involving a spinning electric dipole and a two-dimensional version of the Stokes parameter for circular polarization. The conclusion of the analysis is that such a model qualitatively explains the results, including the spatial dependence of the observed decrease of valley polarization for the monolayer-nanoparticle system.

I am not sure if the manuscript is a good fit for Nature Communications. On the one hand, it fits some of the acceptance criteria: it is probably significant work in this subfield and the conclusions are supported by the evidence presented. The work is deeper than most established literature in the combination of nanostructures and valley polarization, although previous works were alarmingly superficial. The present study is also still incomplete because of the possible presence of other effects, that the authors only briefly discuss. I see no major methodological flaws (at least with current knowledge in the field, which still has major knowledge gaps). On the other hand, although the interpretation is likely mostly correct at least qualitatively, the design of the experiment can be considered faulty as gold nanoparticles are not an optimal choice to excite and detect valley polarization. The manuscript attempts to explain negative experimental results by delving into the effects that could explain the lack of emission polarization using simulations. To a certain extent, the article's premise hinges on the simplistic assumption that an achiral nanoparticle should conserve the degree of valley polarization and also uses an overly simplistic idea of a point spinning dipole located below the center of the nanoparticle as the starting point for the expected results. The focus on explaining this negative result reduces the relevance of this work for other situations that could be better tailored to enhance valley polarization. Such results can be nevertheless interesting and shed some light into light-valley interactions at the nanoscale.

In summary, if the manuscript is not accepted in Nature Communications, I would recommend publication elsewhere as it contains a useful analysis that is likely to be correct and more refined than previously published work on this topic.

Points for improvement include:

1. Confusing use of the word chiral throughout the manuscript. Reference 39 makes a very clear distinction between spinning dipoles and chiral dipoles. The authors use spinning electric dipoles for their simulations, so it is incorrect to refer to chiral emitters as this is a physically different source. Some examples of this confusion between spinning dipoles and chirality are: 'a critical understanding gap remains in how nanostructures and their nearfields affect the polarization properties of valley-selective chiral emission' and 'considering spatially distributed chiral emitters'

2. Some references are given in an inaccurate context, which creates a misleading connection between superchirality and valley polarization. I am referring specifically to the block of references 15-19 attached to “chiral photonic nanostructures have been proven to selectively enhance and efficiently tailor emission from individual valleys”. Allow me to explain case by case what I believe would be the correct context:

- Reference 15 uses a chiral nanoparticle, which naturally acts like a local circular polarizer and behaves like a polarization filter/converter. There is no enhancement of valley polarization in the semiconductor, just circularly polarized emission in this situation. Even valley unpolarized emission should emit circularly polarized light if interacting efficiently with a chiral nanoparticle.
- Reference 17 invokes superchiral fields to explain their results. Critically for the results of the authors, there is no experimental evidence whatsoever that valley polarization is related or proportional to superchiral fields.
- Reference 18 also invokes superchiral fields in simulations.

The results of the authors do not require the use of superchiral fields at all, so the works cited in this manuscript in the context above should be carefully phrased (or even not cited), so as not to perpetuate this misinterpretation that is even incompatible with the presented results. I believe that the value of the present manuscript is trying to clarify the near-field mechanisms that result in circularly polarized emission from valley polarization. It does a disservice to the goal of the manuscript to cite sloppy work without pointing out such incorrect interpretations.

3. Part of the premise of the study is based on an incorrect intuition that symmetric resonators should preserve local circular polarization when excited from the far field. Several sentences are problematic in this respect: ‘Contrary to the simple intuition suggesting that a centrosymmetric nanoresonator preserves the degree of circular polarization in the farfield...’ and ‘By symmetry, the nanoparticle-on-substrate geometry is expected to preserve the helicity of valley-selective chiral emission.’ Such statements are incorrect, as proven by the fact that nanoparticles can preserve or not the helicity of the incident field, as demonstrated in Nano Letters 23, 11, 5101 (2023).

Additionally, the gold nanoparticle is 15 nm away from the monolayer. At the symmetry point below the center of the sphere, one could expect that the monolayer experiences out-of-plane electric fields due to the dielectric-metal interface. In the simulations in Figure 4, it is clear that circular polarization is decreased directly below the nanoparticle, whereas maximum intensity is achieved in a ring surrounding the nanoparticle with a circular polarization that is still not unity, so such a simulation disproves the naïve initial assumption about local circular polarization.

4. Does the absorption of out-of-plane electric fields play any role? The authors seem to neglect fields at the monolayer plane with this orientation, but is this justified by a different absorption coefficient for fields with that orientation? The authors cite the simulation reference 33 in this context, but is there any experimental evidence for the accuracy of such an approximation?

5. Can the authors show the 2D-DOCP of a focused Gaussian beam for reference in the supplementary information? Is the reversal of DOCP also present without the nanoparticle?

6. It also looks like the areas with lower or sign-reverted DOCP in Figure 4 should not be very relevant because there is no intensity at those locations, so it is surprising that they have an effect on the observed polarization. Can the authors comment on this?

7. The manuscript states that the work “unveils a serious impediment of the practical fabrication of resonant valleytronic nanostructures.” Is it an impediment or a constraint that needs to be taken into account when designing such a system for optimal polarization? For example, there could be resonant microcavities where the spatial invariance along the plane of the monolayer does not affect polarized emission in the way that a plasmonic nanoparticle does, so such resonant nanostructures could in principle enhance valley polarization.

8. What is shown in Figure 2c and attributed to absorption? Did the authors measure transmission and reflection in order to calculate absorption? Or is it differential reflection?

9. The methods section does not specify whether a beamsplitter was used for reflection-configuration photoluminescence measurements. If used, what are the polarizing properties and model of that component?

10. Supplementary Figure 1 shows an offset for the S2 Stokes parameter (and S1 to a lesser extent), which is not zero and changes sign with the excitation helicity. What is the origin of this apparent artifact? Also, a subtle detail is visible in the S1 and S2 images: the emission is slightly linearly polarized with lobes following the orientation of the polarization, corresponding to enhanced linearly polarized electric fields at different positions around the nanoparticle, which is to be expected for a metal nanoparticle.

11. Do the Lumerical simulations include the presence of the MoS2 monolayer? This is not specified in the methods.

12. Did the authors specify how they scan in their confocal images? Is the sample scanned or the beam?

13. Can the authors please define explicitly how they calculated ‘the cross-sections and error corridors (shaded regions)’ in Figure 6?

14. I cannot see the logic of using the word “however” in the following sentence: “Other nanophotonic structures facilitated the generation of valley-polarized plasmon/photon–exciton polaritons opening new ways for valley control. However, of crucial importance for the development of on-chip valleytronic devices is directional routing ...”. The paragraph makes sense without “however”.

15. I believe that the following sentence is inaccurate: ‘Finally, by drop-casting and subsequent spin-coating we sparsely distributed monodispersed GNPs’. Usually the term drop-casting is used for a method that relies on placing a drop of liquid and letting it dry, which is incompatible with spin-coating.

Note typos: non-degradative character, inducable degree of valley-polarization, Our numerical calculations closely agrees, spatial extend.

Reviewer #3

(Remarks to the Author)

Tobias Bucher and coauthors investigate spin-valley polarization in hybrid systems composed of gold nanoparticles and monolayer MoS₂. In their optical studies, the combined system is subject to strong depolarization. The experiments seem to have been conducted with care and are well presented. The manuscript is written in a clear, technical language, but is still a good read.

The results are convincing. However, I cannot judge the novelty of the work as my focus is more on plasmonics. The combination has been investigated quite a bit and my guess would have been that polarization in such combinations would have been treated earlier. The guess might be wrong though. If I understood the model correctly, the dipolar nature of the exciton is not changed by the presence of the plasmonic nanoparticle? There is quite a body of work on strong coupling effects in 2D excitons strongly coupled to plasmonic cavities. The authors should detail on how to exclude changed emission properties by a changed exciton fine structure. A brief mentioning of current related works from the field of chiral plasmonics would be beneficial for the manuscript, too.

From the experimental point of view, the same experiment with off-resonant particles would be very interesting as well as measuring polarization for the combination driven off-resonantly.

Version 1:

Reviewer comments:

Reviewer #1

(Remarks to the Author)

See my pdf attached.

Reviewer #2

(Remarks to the Author)

The authors have addressed my concerns for the most part and increased the clarity and accuracy of the manuscript. There are a few manuscript modifications that could be tweaked for extra accuracy:

1. Two issues with the implied link between superchiral fields and valley polarization are still present in the rephrased introduction, which now reads: 'In the field of chiral plasmonics, the modulation of the valley-pseudospin is commonly discussed on the basis of chiral Purcell enhancement (15) involving superchiral nearfields. (16) While such chiral metamaterials have been proven to selectively modulate the valley dynamics and can lead to a sizable DOCP from 2D-TMDs even at room temperature,(17) a fundamental challenge arises when transferring these concepts to valleytronics.' Firstly, even though it has been used a few times in papers, it is not correct to talk about chiral Purcell effect for valley polarization: reference 15 does not deal with valley polarization but with spontaneous emission from chiral quantum emitters. Secondly, phrasing the problem with applying superchiral fields to valleytronics as an issue only due to the use of chiral nanostructures instead of achiral ones is not correct: the reality is that there is no experimental proof that valley polarization should respond to superchiral fields at all. In fact, the evidence presented in this manuscript proves that there is no reason to expect any relation between them. Therefore, I find that the implied connection between superchirality and valley polarization is highly problematic as it goes against the conclusions of the article.

2. Regarding excitation of valley polarization with in-plane field components only, the original version said '... only the tangential components of external fields will contribute to the excitation of the material (33)'. The revised version says '... refers to the in-plane components of the external fields.' avoiding any mention of its relevance to the problem at hand and removing the reference. I think it would be more accurate to say, closer to the original phrasing, that 'mostly the in-plane components of the external fields contribute to the excitation of the material (33)'.

3. For reproducibility, it is good practice to indicate supplier and model numbers for optical components that can introduce measurement artefacts. Please provide that information for the non-polarising beam splitter, mirror and objective.

Reviewer #3

(Remarks to the Author)

Based on the extensive feedback from referee one and two (and my short comments), the authors have revised their manuscript extensively. On a side note, referee comments and replies were an educational reading.

I still cannot judge novelty from the 2D materials perspective, but this left aside, the present manuscript represents an accessible, educational description on the author's studies of coupling plasmons to the valley degree of freedom in 2D semiconductors. I can imagine that content and presentation will receive positive response from the readership of Nature Communications and support publication as is.

Version 2:

Reviewer comments:

Reviewer #1

(Remarks to the Author)

Reviewer #2

(Remarks to the Author)

The authors have addressed my concerns

Point by point response to the Reviewers: Influence of resonant plasmonic nanoparticles on optically accessing the valley degree of freedom in 2D semiconductors

**T. Bucher, Z. Fedorova, M. Abasifard, R. Mupparapu, M. J. Wurdack,
E. Najafidehaghani, Z. Gan, H. Knopf, A. George, F. Eilenberger, T.
Pertsch, A. Turchanin, and I. Staude**

E-mail: tobias.bucher@uni-jena.de and zlata.fedorova@uni-jena.de

We sincerely thank all the reviewers for their thorough analysis of our work. Especially, we appreciate their constructive comments which are pointed towards the improvement of the quality of our manuscript. We read all comments carefully and answer them point-by-point below. For better readability we printed the original reviewer comments in **blue**, the corresponding answers in **black**, and the changes to the manuscript in **red** where the struck out parts were removed and the italic parts were newly added. Where appropriate, page and line numbers are given which refer to the respective position in the originally submitted manuscript. For the revised version please refer to the file with tracked changes.

Reviewer 1, Xavier Zambrana-Puyalto

In their article *Influence of resonant plasmonic nanoparticles on optically accessing the valley degree of freedom in 2D semiconductors*, Tobias Bucher et al. measure the effect that a gold nanoparticle (GNP) has on the degree of circular polarization (DOCP) of the photoluminescence of a monolayer of MoS₂. The authors measure the emitted DOCP of a 1L-MoS₂ flake upon two different illumination schemes: i) a tightly focused circularly polarized Gaussian beam, and ii) a circularly polarized plane wave. They observe that the DOCP of the 1L-MoS₂ emission is completely determined by the handedness of the incident circular polarization (CP). At the same time, the authors observe that the DOCP is drastically reduced in the areas where GNP have been drop-casted on top of the 1L-MoS₂ (which is shielded by a 15 nm layer of silica). After presenting their experimental results, the authors build a physical model that reproduces their experimental findings. Their model takes into account the role of the GNPs in exciting the 1L-MoS₂, as well as in tailoring its emission. Finally, the authors compare their model with the experimental data (Fig. 6), and obtain a good agreement, thus validating their theoretical interpretation.

In general terms, I think that the article is correct, yet I have found it quite confusing at times. I needed to read it multiple times in order to make sure that I understood it properly. I think that part of my confusion stems from the use that the authors do of certain words or concepts which turn out to be crucial to understand their light-matter interaction model. Next I list some of the concepts that, in my opinion, should be clarified in the next version of the manuscript, so that their physical model becomes also clearer.

- **Depolarization.** The article revolves around this concept, but it is not clear to me what the authors mean. Between lines 136-142, the authors state "*the observed reduction in the DOCP is equivalent to a reduction of the total degree of polarization...*". Then, in Fig.1 of the supplementary information (SI), the three Stokes parameters of their measurements are given

which seem to lead us towards the idea that what is actually happening is that the degree of polarization (DOP) is being reduced. But then, the whole article deals with changes in the DOCP, rather than with the decrease in DOP. In my opinion, these points need to be clarified:

i) What do the authors mean by "depolarization"? If an optical field goes from being CP to being linearly or elliptically polarized, has this field been depolarized? Note that this is what typical polarization optics elements do, e.g. linear polarizers, wave-plates, etc. Microscope objectives also do this, they transform the polarization of a beam. But they not introduce any randomness or decoherence effects that are the typical cause of unpolarized light [1, Ch.12]. In my opinion, this is what "depolarization" is about, the decrease of DOP due to unpolarized light or random phases. But of course, I also think that it is fine if the authors use it in a different way, I would just like them to clearly state what they mean.

ii) Do the authors believe that what their hybrid system is doing is changing the DOP and the DOCP at the same time? Or just the DOCP? Or maybe just the DOP? Because, as far as I can see, the whole physical model of the article is based on the assumption that DOP is constant (and equal to 1) at all times. This is what I actually expect, by the way, as their illumination is a HeNe laser. But then, as I mentioned before, the authors also state that the DOP is reduced (and they also provide evidence of that), so then we are left with the question - well, if something is making the DOP to be reduced, then what is it? And why is this not being studied or taken into account in order to explain the experimental results?

Since the questions i) and ii) are closely related we provide the joint answer below. We define depolarization as a reduction in the degree of polarization (DOP). Namely, the PL emitted by the 1L-MoS₂, by interaction, with a GNP goes from being circularly polarized to unpolarized because we do not observe significant enhancement of linear polarization components induced by a GNP as shown in our SI figure. Next, in our hybrid system we observe changing of DOP and DOCP at the same time. We rely on the fact that mathematically the DOCP is equivalent to the Stokes S_3 , while

$$\text{DOP} \equiv \sqrt{S_1^2 + S_2^2 + S_3^2},$$

where we have normalized the total intensity to $S_0 = 1$ (Ref.(1), Ch.8). Since S_1 as well as S_2 are nearly zero, we can equate the change of DOCP with the change of DOP. To improve clarity, we have incorporated the following explanation into the introduction of our manuscript: *Along with that there is no significant increase in the linear polarization components caused, for example, by the ellipticity of the GNP. This leads us to the conclusion that the effect we observed represents a predominating depolarization.*

In addition to that, we have complemented the SI figure that displays all the measured Stokes components (Figure S2 in the revised version) with the map of the DOP to support this point.

Addressing the reviewer's concern regarding the GNP-induced decoherence, we clarify that a nanoparticle does not introduce any additional decoherence because the light emitted from the ensemble of point emitters is already incoherent. Nevertheless, it features circular polarization due to specific character of the emitters associated with a certain valley type. We confirm that such emission can be modelled by rotating dipoles, i.e. the phase shift between the orthogonal dipole components p_x and p_y is fixed to $\pi/2$. The far-field emission from such dipoles, despite they are randomly oriented and emit with random phase offsets relative to one another, can have a degree of polarization (DOP) of 1 given that all emitters belong to one valley and do not interact with photonic nanostructures. As such, the emission retains its polarization characteristics even in the absence of coherence.

- Helicity. I am aware that the term helicity is used in various ways in the literature. And of course, I think that it fine that the authors use it in the way that they think is the most correct. But again, I think that they should clearly state what they mean, otherwise it could get confusing for some readers like me. For me, helicity has to do with the handedness (or the sign of CP) of a field in the momentum space [2]. Formally, it is defined as the normalized projection of the total angular momentum (AM) into the direction of the linear momentum, i.e. $\Lambda = \frac{\mathbf{J} \cdot \mathbf{P}}{|\mathbf{P}|}$ [3, p.136]. As a result, an optical field has a helicity +1 when its decomposition into plane waves yields that all the plane waves are left CP [2, p.4]. Instead, I think that the authors use helicity to talk about CP (in real space). Of course, when we only have one plane wave, or we have a paraxial beam, both things coincide. But they do not coincide when we are talking about dipoles, tightly focused beams or near fields. In particular, in lines 165-166, the authors say that *we observe a strong modification of the 2D-DOCP at larger distances even leading to field components of opposite helicity*. I guess that what they mean is that they have decomposed the field in three polarizations on that plane, and they have observed that the polarization changes depending on the point of the plane. But then, later on, the authors use helicity to talk about the rotation direction of an electric dipole: in line 213, they say *rotating mirror dipole of the same helicity*. But the emitted field of a rotating electric dipoles is definitely not CP. Here, I understand that they talk about helicity of emitting electric dipole to talk about the sign of a rotating dipole of the form $p_{\pm} = p_x \pm ip_y$. The authors also mention in lines 204-205 that *By symmetry, the nanoparticle-on-substrate geometry is expected to preserve the helicity of valley-selective chiral emission*. Here, I am not sure of what they mean, as the two helicity meanings that have been previously given could make sense. Do they mean that a single sphere will preserve the CP handedness in the forward direction in the far field, or maybe they mean that the GNP will have an induced electric dipole that will be of the same sign \pm as the emitting rotating dipole? Notice that there are no general conditions for polarization preservation. Cylindrical symmetry is associated to the preservation of AM, but not polarization. In fact, a rotating electric dipole of the kind $p_{\pm} = p_x \pm ip_y$ emits fields with a well-defined AM= ± 1 [4, 5], and this will certainly be preserved by the substrate-GNP system if the dipole is centred. Instead, the condition for helicity (understood as handedness in the momentum space) preservation is duality symmetry [6], which has to do with the material properties of the matter, and not cylindrical symmetry.

We thank the reviewer for pointing out the formal definition of helicity and the respective difference between the circular polarization state of individual \mathbf{k} -components of the electromagnetic field and electromagnetic fields of well-defined helicity. In the manuscript, we therefore have replaced the term “helicity” with “circular polarization” when talking about field components and with “spin” when talking about the rotation sense of dipoles. For the sentences and phrases specifically pointed out by the reviewer, we have incorporated the following corrections:

- ~~In fact, we observe a strong modification of the 2D-DOCP even leading to field components of opposite helicity.~~ (Note that this paragraph was changed according to the revision of Fig. 4.)
- ~~When placed below a resonant GNP, the rotating electric dipole is expected to induce a respective mirror rotating mirror dipole of the same helicity with equal spin (note that as both linear dipole components of the mirrored rotating dipole are off-set by a will experience the same π phase flip leaving the helicity unchanged).~~

- By symmetry, the nanoparticle-on-substrate geometry is expected to preserve the helicity circular polarization of valley-selective chiral emission in the forward scattered farfield.

• Chirality. Chirality is a term that tends to have a more established meaning. Typically, we say that an object is chiral when you cannot superimpose it with any of its mirror images. As the authors write in line 66, a single GNP is achiral. I agree with that. But then, chirality is mostly used in the article to refer to the emission of 1L-MoS₂. For example, in line 67 "specific chiral emission from a monolayer of 1L-MoS₂", or in line 117 "valley-specific chiral emission". When it comes to optical fields, chirality can be used as helicity (handedness in momentum space), consequently there are only two possible handednesses that an optical field can have, + or -. Or left and right. It is well-known that a CP plane wave is chiral, and it has two different handednesses, left and right. But things get more complicated when we have optical fields that are superpositions of plane waves, such as the emission of dipole. When I read that the emission of 1L-MoS₂ is chiral, what I understand is that the emission of 1L-MoS₂ can be decomposed into plane waves, and that all these plane waves have the same handedness. Now, I do not know if this is what they authors meant or not, but this is not the case if we model the emission of 1L-MoS₂ as a rotating electric dipole of the kind $\mathbf{p}_{\pm} = \mathbf{p}_x \pm i\mathbf{p}_y$. In fact, it can be proved that such a dipole is achiral, i.e. that the emitted field can be superimposed with its mirror image. Or in other words, that its field can be decomposed 50-50% into fields with left-right handedness [5, p.3]. I suspect that the reason why this gets confusing is because the authors are only interested in the emission of 1L-MoS₂ in one semi-space. And indeed, if we only consider one semi-space, then we could consider that the field emitted by a \mathbf{p}_{\pm} electric dipole is mostly formed by plane waves with the \pm handedness. I have depicted this in Fig. R1. In Fig. R1a), I show the near-field of a rotating dipole of the kind $\mathbf{p}_{\pm} = \mathbf{p}_x \pm i\mathbf{p}_y$. It is

FIG. R1. a) Power emitted by a rotating electric dipole $\mathbf{p}_{\pm} = \mathbf{p}_x \pm i\mathbf{p}_y$. b) Power emitted by a rotating chiral dipole with left handedness created as a in-phase superposition of rotating electric and magnetic dipoles, i.e. $\mathbf{p}_+ + \mathbf{m}_+$, with \mathbf{m}_+ being a rotating magnetic dipole. c) Power emitted by a rotating chiral dipole with right handedness, i.e. $\mathbf{p}_+ - \mathbf{m}_+$, with \mathbf{m}_+ . All plots are done at 632nm, for an area of $1.2 \times 1.2 \mu\text{m}^2$. The black region in the center of the plot is avoided because emitting dipolar fields diverge at the emitting point.

observed that it emits light all over space. Then, in Fig. R1b) and c), I depict the near-field of rotating dipoles (with the same + rotating sign) that emit light only with left (or +) and right (or -) handedness respectively. To create these dipoles, we need to superimpose electric and magnetic rotating dipoles with the same rotating sign +. And then, it is easy to see that the dipole in a) is the addition of the dipoles in b) and c). In fact, this is also plotted by the

authors in Fig. 5 of the manuscript, but for the far-field. That is, effectively, a rotating dipole emits light with one handedness in one semispace, and it emits light the other handedness in the other semispace. Consequently, it is achiral but has a well-defined AM equal to \pm [5, p.3].

We thank the reviewer for pointing out the difference between a chiral dipole (e.g. $\mathbf{p}_x \pm i\mathbf{m}_x$) and an (achiral) rotating electric dipole whose optical field has no defined helicity but carries an angular momentum. As further pointed out by the reviewer, our experimental focus is on the polarization of light emitted into one half space such that it seems helicity is defined in this case. To avoid such confusions, we have removed the attribute "chiral" when talking about "valley-selective emitters" and replaced "chiral" with "circular polarization" when talking about specific field components.

This is also highlighted by the farfield pattern and respective DOCP of a single rotating electric dipole as shown in Figure 5a in the main text. We have improved the following section of the main manuscript to discuss the polarization properties of a rotating electric dipole:

- This can be understood by modelling PL emission from using dipoles exhibiting ~~chiral circularly polarized farfield components farfield properties matching those of excitons in bare 1L-TMDs~~. For this purpose, different types of emitters with ~~chiral farfield properties~~ are discussed in literature [...] where the *spin or valley index* is associated with the fixed phase sign. *On the right side of Figure 5a we show the farfield intensity distribution and respective DOCP of a single counterclockwise rotating electric dipole. Due to its fixed axis of rotation, $\vec{\mathbf{p}}_K$ emits circularly polarized light with opposite handedness into different half spaces matching the PL polarization properties of valley-selective excitons in bare 1L-TMDs (see Sec. S3 of the Supporting Information).*

In Section S3 of the Supporting Information we then briefly discuss the different polarization properties of a rotating electric dipole $\vec{\mathbf{p}}_{K/K'} = \vec{\mathbf{p}}_x \pm i\vec{\mathbf{p}}_y$ as compared to a truly chiral dipole $\vec{\mathbf{p}}_{\sigma\pm} = \vec{\mathbf{p}}_x \pm i\vec{\mathbf{m}}_x$. We reason that these two dipole types could be distinguished experimentally by comparing the handedness of backwards emitted light with respect to the polarization of the reflected excitation laser. We have performed the respective measurements on bare 1L-MoS₂ as shown in Figure S3 of the Supporting Information and below.

There we explain that a chiral dipole would yield an opposite handedness of emission in backwards direction as compared to the handedness of the reflected laser beam that was used for excitation. Therefore, our experimental observation rules out the use of the chiral dipole but supports the use of the rotating electric dipole in the context of modelling valley-selective emission from 1L-TMDs.

To wrap up this general comment about concepts/wording - my intention, of course, is not to try to impose any kind of wording, but rather to show the authors that their current wording, which is necessary to understand their physical model, does not seem to be very consistent throughout the paper and could lead to misunderstandings. As I mentioned, I think that I partially struggled to understand the paper because of this.

We highly value the clarifications by the reviewer with respect to the established concepts and notations in literature and do hope that the revised manuscript now provides a more precise and clearer explanation of our physical model and the underlying mechanisms.

Next, I move on to comment some other issues. I find Fig.4 confusing, which is problematic,

Fig. S3: Comparison of the polarization contrast of reflected laser and emission. (a) Sketch indicating the circular polarization of excitation light and emitted PL for the case of a rotating electric dipole (left side) and a chiral dipole (right side) upon σ^+ polarized excitation. (b) Normalized spectra of the σ^+ -polarized HeNe laser detected after the reflection from the sample (solid lines) and of the photoluminescence from the bare 1L-MoS₂ excited by this laser (dashed) for σ_+ (black) and σ_- (red) detection polarizations.

because it is what summarizes the excitation model of the authors. First of all, for propagating beams, I am used to talking about transverse and longitudinal polarization components. That is, if a beam is propagating in the z direction, and we decompose it into three polarizations components, e.g. $\mathbf{E} = E_x\hat{\mathbf{x}} + E_y\hat{\mathbf{y}} + E_z\hat{\mathbf{z}}$ or $\mathbf{E} = E_{\sigma^+}\hat{\sigma}^+ + E_z + E_{\sigma^-}\hat{\sigma}^-$, then I would consider $\hat{\mathbf{z}}$ to be the longitudinal component of polarization, and $\{\hat{\mathbf{x}}, \hat{\mathbf{y}}\}$ or $\{\hat{\sigma}^+, \hat{\sigma}^-\}$ would be the transverse components. The authors use the expression "tangential" components. I believe that they are referring to the $\{\hat{\sigma}^+, \hat{\sigma}^-\}$ components.

We agree with the reviewer noting the improper usage of the word "tangential". To remove the ambiguity we have replaced "tangential" by "in-plane" throughout the manuscript.

But then, when I look at the intensity and the DOCP plots, I get very confused. For whoever has worked with nanoparticles and Gaussian beams, I think that it would be shocking to see that the result of this interaction is a doughnut beam. Typically, one would expect something similar to a Gaussian beam, or an Airy-type pattern. And then, because I know that the longitudinal component of a tightly focused Gaussian beam looks quite similar to a doughnut beam, I started wondering if what the authors meant by "tangential" was longitudinal. I had to go and simulate a similar system (a tightly focused Gaussian beam) interacting with a sphere to see what is happening - see Fig. R2(a). In Fig. R2(a), I plot the total transverse field, i.e. scattered + incoming, on a $x - z$ plane of $1 \times 1 \mu\text{m}^2$. We observe that indeed a vortex beam is formed at the plane at 15 nm from the tip of the sphere (the beam is travelling from negative to positive z 's). But already at 100 nm from the tip of the sphere, the field resembles that of a Gaussian beam. In my opinion, there needs to be a discussion about all this in the revised version of the manuscript.

Prior to addressing this comment, it is important to clarify that while the in-plane intensity exhibits a donut-like distribution, it does not constitute a vortex beam as there is no source of azimuthal phase variation. In response to the reviewer's suggestion, we have elaborated further on the formation of the specific intensity patterns below the GNP. For that we have prepared

FIG. R2. a) Plot of $|E_{\sigma+}|^2 + |E_{\sigma-}|^2$ for a x-z plane of $1 \times 1 \mu\text{m}^2$ size. The inset is a x-y plane of $1 \times 1 \mu\text{m}^2$ size at 15 nm of the tip of the sphere. b) DOCP in the same x-y plane as the inset. All plots are done with a CP Gaussian beam with a $\hat{\sigma}^+$ polarization before focusing. The wavelength is $\lambda = 632\text{nm}$. The GNP has a radius of 100 nm.

the new SI figure (Figure S5 in the revised version) with the proper description (see section S5)

Furthermore, we refer to this figure in the main text: *Figure 4a shows the resulting total in-plane intensities generated by a focused Gaussian beam (left) and a plane wave (right). In both cases, the maximum intensity is concentrated in a ring around the GNP. We note that this pattern is characteristic when observing closely to a spherical GNP, within distances less than 50 nm, where near-field interactions with nanoantennas are typically prominent. Further insights into the formation of such a ring can be found in Sec. S5 of the Supporting Information.*

By the way, I also simulated the DOCP in the near-field using the definitions given in the manuscript, but I did not get the same results as the authors - see Fig. R2(b). It can be seen that the results differ quite a bit from the ones shown in the paper, but I understand that they simulated a sphere lying on an infinite glass layer, so obviously the result is different. Not sure if it makes sense that the two systems behave so differently - in my plot the DOCP is always positive, and in the plot in Fig. 4 of the paper we see a DOCP going to negative values - but anyway, I cannot say much more about it. What I am especially puzzled with is the DOCP for the plane wave case. It looks as the DOCP plot was just the DOCP of the incoming plane wave and no scattering had been taken into account. Surely the near-field scattered by the GNP will alter the DOCP of the incoming plane wave and will yield some components in the opposite polarization $\hat{\sigma}^-$. Could the authors double-check this? In fact, the image seems to be in contradiction with the text, since in line 160 is mentioned that the GNP leads to a slight reduction of the DOCP even for the plane wave case, but the DOCP seems to be constant and equal to 1 across the plane. This constant DOCP plot is one of the reasons why I am quite confused with Fig. 4, as it made me think that these plots were showing just the incoming beams, and not the incoming beam + scattering.

Your comment helped us to realize that our plots of 2D-DOCP were hard to interpret. While technically correct, they do not provide the direct information on the magnitude of the valley-specific exciton densities induced by the scattered laser field. To improve the clarity of our findings, we decided to change the displayed data. Instead of showing the 2D-DOCP in Figure 4

Fig. S5: Nearfield intensity distribution of a gold nanoparticle on substrate. Left: Numerically computed $|E_x|^2 + |E_y|^2$ for a x - z plane intersecting the center of a GNP. The white dots below the GNP correspond to $z = -15$ nm, $z = -50$ nm, and $z = -100$ nm. Right: $|E_x|^2 + |E_y|^2$ for x - y planes centered around the GNP at different z values. The scale bar corresponds to 200 nm.

we plot the individual contributions of σ^+ and σ^- polarized in-plane fields. See below the revised version of the figure:

We modified the textual description accordingly:

The individual contributions of σ^+ and σ^- polarized field components to these intensities are plotted in Figure 4b, represented by red and blue curves respectively, against the distance r from the GNP's center. For both excitation conditions, the GNP induces the cross-polarized intensity component $I_{\sigma^-}^{\parallel}$ peaking at the projected GNP's edge (gray shaded region) and diminishing with increasing distance. Notably, this cross-polarized component is slightly larger for the focused beam. Reference simulations of a tightly focused Gaussian beam without a GNP (see Sec. S6 of the Supporting Information) reveal that $I_{\sigma^-}^{\parallel}$ remains nearly zero, emphasizing that the slight increase of this component observed in the presence of a GNP is not due to high NA itself. Additionally, for the focused beam the intensity of σ^+ -polarized field drops faster than σ^- , resulting in their curves intersecting at $r \approx 0.22$ μm . In contrast for a plane wave, the total field at larger distances from the GNP is dominated by the incident field characterized by pure σ^+ polarization. The resultant 2D-DOCP maps for all of the discussed cases are provided in the Supplementary Information, Figure S6.

Note, that we kept the previous plots of 2D-DOCP in the Supporting Information, see Figure S6 in the revised version. Furthermore, we have added a similar plot for the emitted farfield intensities to Figure 5 for consistency.

The textual description is adjusted as follows: *By integration over a numerical aperture of 0.9, we obtain the helical farfield intensities $\mathcal{I}_{\sigma^\pm}^K(r)$ as plotted on top of Figure 5b. Similar to the previously analyzed nearfields, we observe the emergence of the cross-polarized intensity component $\mathcal{I}_{\sigma^-}^K(r)$, peaking at about $|r| \approx 60$ nm, with its contribution being notably pronounced. Below, in Figure 5b we show the resulting total farfield intensity (orange curve) and the farfield DOCP (green curve) as functions of the displacement distance r .*

Fig. 4: Impact of the gold nanoparticle on the valley-selective excitation of 1L-MoS₂. (a) Total intensity of the external in-plane nearfield in the plane of the monolayer upon excitation with a focused Gaussian beam (NA=0.9) positioned at the center of the GNP (left) and a plane wave (right). (b) Respective individual contributions of σ^+ and σ^- polarized components in dependence on the distance $r = \sqrt{x^2 + y^2}$. The normalization is chosen such that the sum of both curves peaks to 1. (c) Measured DOCP of valley-specific emission from embedded 1L-MoS₂ decorated with monodispersed GNPs upon σ^+ -polarized wide-field illumination and collected through a 660 nm bandpass filter.

In the article, it is also mentioned that Fig.3(a) is the result of confocal microscopy. Whereas, I assume that the DOCP shown in Fig. 4(b) is not a confocal measurements, right? That is, to measure DOCP for the plane wave excitation, pictures of the whole flake are taken (under a plane wave, or more accurately a weakly focused Gaussian beam) and the polarization analysis is done on the whole image right? For this case, it is clear to me that the polarization that all the points in the flake will feel is the same. However, in the confocal case, I am less sure about that, especially if the scan on the sample is done with a scanning laser beam, instead of with a scanning stage. If the sample is scanned with a stage, then all the pixels of the image, corresponding to different positions on the plane of the sample, feel the same polarization distribution, given by the tightly focused Gaussian beam. But I am not sure that this is the case when the scan is done by a scanning laser. Could the authors comment on this?

As summarized correctly by the reviewer, the measurements presented in Fig. 3 are obtained by confocal scanning (point-by-point excitation and detection) and the measurements presented in Fig. 4b are obtained by widefield imaging (widefield excitation and "single-shot detection"). In our confocal measurements, the sample is scanned inside the chamber of the cryostat by two piezoelectric stages. In this way we can ensure that the excitation beam (and particularly its polarization) does not change during the scanning measurements. We have added this information to the revised manuscript as follows:

- Materials and Methods", section "Optical experiments at cryogenic temperatures" (page 16, lines 323): *Lateral scanning was performed using two piezoelectrical nanopositioners moving the sample. This allows for stable conditions of the focused excitation with a fixed degree of circular polarization.*

A little bit related to this, in lines 178-180 we are told about this Gaussian smoothing. Does

Fig. 5: Impact of the gold nanoparticle on the photoluminescence polarization from distributed emitters. (a) Sketch of the simulated nanoparticle-on-substrate geometry showing the position of the rotating electric dipole in a plane 15 nm below and displaced by a distance r from the projected center of the GNP. The inset shows the calculated farfield radiation pattern of a rotating electric dipole \vec{p}_K , the color encodes the respective DOCP. (b) Top: Calculated farfield intensities $\mathcal{I}_{\sigma+}^K$ (red line) and $\mathcal{I}_{\sigma-}^K$ (blue line) emitted by a rotated dipole \vec{p}_K as a function of the displacement distance r . Bottom: The corresponding DOCP (green curve) and total intensity (orange curve) of the integrated farfield. The farfield intensity is normalized to the intensity obtained without a GNP.

this just mean that a plane wave is not used, but rather a very weakly focused Gaussian beam?

No, we used a perfect plane wave for the excitation. The Gaussian function played a role of the point spread function to account for the finite optical resolution. To prevent the ambiguity, we replaced the word "smoothing" by "point spread function" throughout the manuscript.

Also, in Fig.1(a) in the SI - why is there so little photoluminescence coming from the center of the flake? Is this due to photobleaching? In fact, could the authors comment on the power that was used to do the experiment? How does this affect the emission of 1L-MoS₂?

We thank the reviewer for pointing out the missing information on the laser power used in our optical experiments. We have used an average laser power of 50 μW (16 kWcm^{-2} peak intensity) and 200 μW (40 Wcm^{-2} peak intensity) for the focused and collimated excitation scheme, respectively. In this power regime we have not noticed any effects of saturation or photobleaching throughout the experiments. We have included the missing information in the new version of the manuscript as follows:

- "Materials and Methods", section "Optical experiments at cryogenic temperatures" (page 16, lines 322-323): "For confocal scanning measurements, the excitation beam with an average power of 50 μW was focused to a diffraction limited spot with a diameter of $2r = 2\lambda / (\text{NA} \cdot \pi) \approx 0.45 \mu\text{m}$ and a peak intensity of 16 kWcm^{-2} ."
- "Materials and Methods", section "Optical experiments at cryogenic temperatures" (page 16, lines 326-327): "For wide-field measurements, an additional lens was introduced to focus the excitation beam with an average power of 200 μW onto the back-focal plane of the objective. This results in a beam size of $\approx 25 \mu\text{m}$ on the sample and a peak intensity of 40 Wcm^{-2} ."

- "Polarization-resolved cryo-PL measurements" (page 7, lines 120-122): *"For excitation and detection we used a 100x/0.9 NA objective in reflection geometry and limited the collected light to a (660 ± 5) nm wavelength band (see Methods). we used a 100x/0.9 NA objective and an average laser power of $50 \mu\text{W}$ (16 kWcm^{-2} peak intensity). In detection, light was collected in reflection geometry using the same objective and limited to a (660 ± 5) nm wavelength band.*
- "Polarization-resolved cryo-PL measurements" (page 10, lines 171-172): *"[...] the respective measured DOCP image of PL from the same 1L-MoS₂ sample as shown before and measured with an average excitation power of $200 \mu\text{W}$ (40 Wcm^{-2} peak intensity).*

Regarding the inhomogeneous photoluminescence intensity distribution, please note that we have investigated flakes of 1L-MoS₂ that were synthesized by chemical vapour deposition. As growth conditions may vary across the growth substrate and/or during the growth process, such differences in photoluminescence intensity distribution are naturally present within a growth batch. To our experience, however, this has no impact on the polarization properties of emission as further confirmed by the homogeneous distribution of the S-parameters of PL shown in Fig. 1b of the SI.

In the sample preparation, it is mentioned that the GNP are drop-casted and spin-coated. What is the purpose of the spin-coating? And does the polymer or resist get removed afterwards?

For the deposition of GNPs we have used a spin-coating protocol (see "Materials and Methods", section "Gold nanoparticle deposition") allowing for a homogeneous coverage of the substrates with evenly distributed monodispersed GNPs. The GNP suspension (mixture of isopropanol and citrate buffer in a ratio 3:1) is dripped onto the substrates for spin-coating. Subsequently, the remaining liquid is allowed to evaporate. No additional resist was used in this case.

Please note that Reviewer 2 has pointed out correctly the different meaning of "drop-casting" which is incompatible with our spin-coating protocol and the term has been removed from the manuscript (see response to comment 15 of Reviewer 2).

In line 161 - what does inducable mean? I guess that it should be inducible. Also in line 189, there's a typo - calculated instead of calcualted.

We thank the reviewer for the careful reading, the typos were removed in the revised manuscript.

Some final comments:

- Fig. 6 is, in a way, what "proves" that the modelling done is correct. Yet, I think that very few details are provided so that we can try to reproduce their results. Unless I am mistaken, in order to get Fig. 6, what the authors have done is: i) They have computed the intensities $|E_{\sigma^+}(x,y)|^2$ and $|E_{\sigma^-}(x,y)|^2$, which are computed as illumination + scattering for both illumination schemes. ii) They have assumed that the emission of 1L-MoS₂ is given by \mathbf{p}_+ and \mathbf{p}_- -like dipoles, whose probability to emit is proportional to the intensities $|E_{\sigma^+}(x,y)|^2$ and $|E_{\sigma^-}(x,y)|^2$. These probabilities to emit in \mathbf{p}_+ and \mathbf{p}_- states are proportional to what they call local exciton densities $n_K(x,y)$ and $n_{K'}(x,y)$. iii)

They have taken the dipolar emission caused by the local exciton densities $n_K(x,y)$ and $n_{K'}(x,y)$ and have convoluted their (x,y) distribution on the plane with Fig. 5(b). Fig 5(b) models the DOCP that a far-field detector measures as a result of a \mathbf{p}_+ emission in a glass substrate located at 15 nm below the sphere contact point with the glass, and at a transverse distance r of that contact point. The authors do not show it, but obviously there must be another counterpart of Fig. 5(b) with an emitting \mathbf{p}_+ dipole and negative values of DOCP. Now, my feeling is that the article would be clearer if the reader was guided through this process more thoroughly. For that, the authors would need to show and quantify (when possible) all the intermediate results. The article is relatively long (14 pages), and the section regarding Fig. 6, which verifies all the rest, is only one and a half pages.

In order to improve the clarity of the manuscript, we have updated Figures 4(b) and 5(b) to show the helical nearfield excitation and farfield emission intensities as used for the model calculations. We have further added the following sentences to clarify the cases of opposite handedness:

- "Nearfield excitation polarization", caption of figure 4 (page 9): *Note that identical results are obtained for σ^- polarized excitation up to exchanged labels σ^\pm , respectively.*
- "Farfield emission polarization", caption of figure 5 (page 11): *Note that the results for a dipole with opposite spin, $\vec{\mathbf{p}}_{K'}$, can be obtained by exchanging the labels σ^\pm , respectively.*
- "Farfield emission polarization" (\circ): *Similarly, we have discussed above that a single exciton located in the valley K (or K') leads to both σ^+ and σ^- polarized PL intensities in the farfield which we denoted as $\mathcal{I}_{\sigma^\pm}^K$ (or $\mathcal{I}_{\sigma^\pm}^{K'}$), respectively, where the farfield intensities from emitters in opposite valleys are obtained as $\mathcal{I}_{\sigma^\pm}^{K'} = \mathcal{I}_{\sigma^\mp}^K$.*

Please note that we have already mentioned this approach in the Methods, section "Numerical simulations" (page 18, lines 364-367).

Further, a brief summary of the calculation was added to the discussion of Figure 6 to guide the reader through these steps.

- "Full model for optically addressing the valley degree of freedom": *For the hybrid system, we recall that a σ^+ polarized excitation beam leads to non-zero exciton densities in both valleys K/K' proportional to the in-plane intensity distribution of the total field, $n_{K/K'} \propto I_{\sigma^{+/-}}^{\parallel}$. Similarly, we have discussed above that a single exciton located in the valley K (or K') leads to both σ^+ and σ^- polarized PL intensities in the farfield which we denoted as $\mathcal{I}_{\sigma^\pm}^K$ (or $\mathcal{I}_{\sigma^\pm}^{K'}$), respectively, where the farfield intensities from emitters in opposite valleys are obtained as $\mathcal{I}_{\sigma^\pm}^{K'} = \mathcal{I}_{\sigma^\mp}^K$. Assuming a fixed excitation beam (as valid for widefield excitation), we can express the farfield PL contributions emitted from a point (x,y) of the 1L-MoS₂ crystal in a helical basis as $\mathcal{I}_{\sigma^\pm}(x,y) = n_K(x,y) \cdot \mathcal{I}_{\sigma^\pm}^K(x,y) + n_{K'}(x,y) \cdot \mathcal{I}_{\sigma^\pm}^{K'}(x,y)$. In order to obtain the farfield observable distribution we convolute the local contributions $\mathcal{I}_{\sigma^\pm}(x,y)$ with a Gaussian point spread function where the Gaussian width matches the optical resolution of the experimental setup (see Equation 3 in Methods). For the case of*

confocal microscopy, the excitation beam is not fixed but instead scanned across the sample. Hence, the exciton density distribution has to be calculated for each excitation beam position separately and the farfield observable PL contributions are then described by Equation 4 as described in the Methods section.

Please note that this section is intentionally kept shorter as it summarizes all the effects which have previously been discussed and/or are detailed in the Method section.

Also, we are told about an averaging over GNP sizes for the theoretical curve. But we are not told what sort of sizes they have averaged upon. Also, what about the experimental results? Are they also averaged or they are just the points for one of the GNP? If they are not averaged, does it make sense to average the theoretical results instead of measuring the GNP size and applying some error bars to the measurement and do the simulation with different sizes within the error bars? In summary, I think that more details should be given in this part.

There seems to be a misunderstanding; the theoretical curve was not averaged over various GNP sizes. The error corridors depicted in Figure 6 represent the standard deviation of the experimental results, which were obtained from measurements on three individual GNPs on a single flake, considering both x- and y- cross-sections. To address this concern and provide further clarity, we have included a more detailed explanation in the main text of the revised version:

The cross-sections and error corridors (shaded regions) are computed as the mean value and the standard deviation, respectively. For our sampling we have used the DOCP shown in Figure 3c (scanning) or Figure 4b (wide field) and compared the cross-sections along x- and y- axes of three measured nanoparticles on a single flake (in total 6 samples per measurement scheme).

- Finally, after reading the article, I am left a little bit thinking that we have not gotten many insights out of this experiment. The authors have done an experiment and in principle they have managed to reproduce the results with a physical model. That is good work, of course. But what can we get out of this? We have learned that this highly symmetric system does not preserve DOCP (contrary to what some could expect), and we have learned how to model this complex light-matter interaction problem to get good matching with the experimental data. But what is the underlying reason that is making this happen apart from a convolution of effects? If we were to vary the system slightly, should we go through the whole simulation process to predict the outcome? Or can we build some intuition based on these results? For example, what would have happened if the particle was not cylindrically symmetric? Or if the particle did not behave like an electric dipole, but rather as a combination of electric and magnetic? Also, what would it happen if the 1L-MoS₂ flake was at a different distance. As seen in Fig. R2a), the fact that a doughnut beam is obtained at a plane at 15 nm from the contact point of the sphere is a bit of a surprising near-field condition that already disappears at 100 nm from the contact point of the sphere. And it can be checked that if the particle would have been made of TiO₂ (just to name a common material for nanophotonics), the nearfield intensity would have been very similar to a Gaussian beam. In fact, one could intuitively think that what is happening here is that the 1L-MoS₂ is located at a precise distance of the GNP such that the intensity on that plane (right below the GNP) is zero for a large area of roughly 100 x 100 nm². And as a result, the DOCP and DOP that the authors measure are 0 basically because there is no emission. Whereas if that near-field intensity distribution was more

Gaussian-like, which is what intuitively one would expect, then the values of DOCP and DOP would be much better preserved. I am not saying that this is what is happening exactly, but basically, as the article stands right now, I do not think that we get any clear big picture of what is happening. And thus, we are left with the idea that if we were to change any of the particularities of the experiment that I mentioned above (particle shape or material and distance), that we would need to re-think the whole experiment another time.

We agree with the reviewer that it is not straight forward to develop a general yet reasonably accurate intuition for other systems based solely on the geometry investigated in our work. Here, we would like to point out that our work cannot make the claim to provide a general answer to all the questions raised. Instead, we aim to provide new insights for the scientific community pointing at a list of necessary aspects for the design and characterization of nanoscopic valleytronic devices. We would like to briefly elaborate on this:

- First, we have shown that even for highly symmetric systems the collective response may deviate quite strongly from the simple intuition based on that symmetry (Specifically, the cylindrical symmetry of the GNP-on-substrate geometry is expected to preserve angular momentum along the out-of-plane direction both for excitation and emission). Despite our demonstration based on a specific system, it concerns a general aspect: the need for a predictive model for the response of 2D-TMDs integrated with resonant nanophotonic systems which helps researchers to accurately design and optimize hybrid systems for valleytronic applications. This need is only emphasized by the apparent discrepancy between expected and demonstrated functionality found in many previous works in the field. In this regard, our model not only provides very good quantitative agreement with the experimental results but also gives intuition by identifying and quantifying the impact of several factors, namely the excitation scheme, the distribution of emitters, and the optical resolution in detection.
- Second, by our thorough and systematic experimental characterization using high-optical resolution photoluminescence microscopy at cryogenic conditions we are able to provide the nontrivial experimental verification of the model. By doing so, we aim to motivate and set a new standard that is appropriate for tailoring and investigating nanoscopic valleytronic devices. This comprehensive approach, which in our eyes is often missing in previous studies, has the potential to provide the necessary insights to advance this field of research further.
- Third, this opens very interesting directions for future studies by investigating different materials and resonance types, structured light illumination techniques or patterning of 2D-TMDs. Considering additional effects like exciton diffusion and the quantum regime of light-matter interaction will provide a route for very interesting follow-up research topics.

We thank the reviewer for raising these important points which lead us to further strengthen these points in the conclusion of our manuscript (see revised version).

I am sorry for the long review. But I hope that these comments will help the authors to improve their work.

Reviewer 2

The manuscript deals with the possible manipulation of valley polarization in a MoS₂ monolayer using resonant plasmonic nanospheres. Measurements reveal a lack of polarization in the emitted light below the nanosphere. The authors investigate this problem numerically decomposing it into excitation and emission, involving a spinning electric dipole and a two-dimensional version of the Stokes parameter for circular polarization. The conclusion of the analysis is that such a model qualitatively explains the results, including the spatial dependence of the observed decrease of valley polarization for the monolayer-nanoparticle system.

I am not sure if the manuscript is a good fit for Nature Communications. On the one hand, it fits some of the acceptance criteria: it is probably significant work in this subfield and the conclusions are supported by the evidence presented. The work is deeper than most established literature in the combination of nanostructures and valley polarization, although previous works were alarmingly superficial. The present study is also still incomplete because of the possible presence of other effects, that the authors only briefly discuss. I see no major methodological flaws (at least with current knowledge in the field, which still has major knowledge gaps). On the other hand, although the interpretation is likely mostly correct at least qualitatively, the design of the experiment can be considered faulty as gold nanoparticles are not an optimal choice to excite and detect valley polarization. The manuscript attempts to explain negative experimental results by delving into the effects that could explain the lack of emission polarization using simulations. To a certain extent, the article's premise hinges on the simplistic assumption that an achiral nanoparticle should conserve the degree of valley polarization and also uses an overly simplistic idea of a point spinning dipole located below the center of the nanoparticle as the starting point for the expected results. The focus on explaining this negative result reduces the relevance of this work for other situations that could be better tailored to enhance valley polarization. Such results can be nevertheless interesting and shed some light into light-valley interactions at the nanoscale.

In summary, if the manuscript is not accepted in Nature Communications, I would recommend publication elsewhere as it contains a useful analysis that is likely to be correct and more refined than previously published work on this topic.

We thank the reviewer for the critical yet constructive assessment of our work. In response to the concern about the weak motivation of our work, we have refined our introduction to emphasise the broader scientific implications and practical insights our study offers beyond merely explaining a negative result. Although spherical plasmonic nanoparticles are indeed not the optimal choice for preserving or enhancing valley polarization, their simple geometry and pronounced polarization effects make them ideal for studying the complex near-field interactions that are central to our study. To better represent the significance of our findings and the novel insights they provide, we have incorporated an additional paragraph into the introduction: *This leads us to the conclusion that the observed effect represents a predominating depolarization. This phenomenon raises fundamental questions about the underlying mechanisms of polarization effects within such hybrid systems. Specifically, what causes the observed depolarization? Is this a result of near-field effects during the excitation phase, or does it occur during the re-emission of photons? Furthermore, which valleys are actually excited in the process? These are the key questions we aim to address in our investigation.*

Points for improvement include:

1. Confusing use of the word chiral throughout the manuscript. Reference 39 makes a

very clear distinction between spinning dipoles and chiral dipoles. The authors use spinning electric dipoles for their simulations, so it is incorrect to refer to chiral emitters as this is a physically different source. Some examples of this confusion between spinning dipoles and chirality are: ‘a critical understanding gap remains in how nanostructures and their nearfields affect the polarization properties of valley-selective chiral emission’ and ‘considering spatially distributed chiral emitters’

We thank the reviewer for pointing out the difference between a chiral dipole (e.g. $\mathbf{p}_x \pm i\mathbf{m}_x$) and an (achiral) rotating electric dipole whose optical field has no defined helicity but carries an angular momentum. Therefore, we have removed the attribute "chiral" when talking about "valley-selective emitters" and replaced "chiral" with "circular polarization" when talking about specific field components.

2. Some references are given in an inaccurate context, which creates a misleading connection between superchirality and valley polarization. I am referring specifically to the block of references 15-19 attached to “chiral photonic nanostructures have been proven to selectively enhance and efficiently tailor emission from individual valleys”. Allow me to explain case by case what I believe would be the correct context:

- Reference 15 uses a chiral nanoparticle, which naturally acts like a local circular polarizer and behaves like a polarization filter/converter. There is no enhancement of valley polarization in the semiconductor, just circularly polarized emission in this situation. Even valley unpolarized emission should emit circularly polarized light if interacting efficiently with a chiral nanoparticle.

- Reference 17 invokes superchiral fields to explain their results. Critically for the results of the authors, there is no experimental evidence whatsoever that valley polarization is related or proportional to superchiral fields.

- Reference 18 also invokes superchiral fields in simulations.

The results of the authors do not require the use of superchiral fields at all, so the works cited in this manuscript in the context above should be carefully phrased (or even not cited), so as not to perpetuate this misinterpretation that is even incompatible with the presented results. I believe that the value of the present manuscript is trying to clarify the near-field mechanisms that result in circularly polarized emission from valley polarization. It does a disservice to the goal of the manuscript to cite sloppy work without pointing out such incorrect interpretations.

We thank the reviewer for the careful evaluation of the references given. Specifically, we agree with the verdict about references [15,17] which have been removed from the revised manuscript. Further, we added a brief introduction to the concept of valley manipulation by chiral nearfield effects and discuss the difference to achiral structures as follows:

- "Introduction" (page 3, lines 42-47): *During the last few years, the integration of 2D-TMDs with photonic nanostructures has gained immense popularity as an approach to address these challenges by enhancing and tailoring light-valley interaction at the nanoscale. In the field of chiral plasmonics, the modulation of the valley-pseudospin is commonly discussed on the basis of chiral Purcell enhancement (2) involving superchiral nearfields. (3) While such chiral metamaterials have been proven to selectively modulate the valley dynamics and can lead to a sizable DOCP from 2D-TMDs even at room temperature, (4) a fundamental challenge arises when transferring these concepts to valleytronics. The chiral asymmetry permits only coupling to interacting objects of the same handedness (e.g. chiral molecules). In 2D-TMDs, however, the handedness of*

chiral valley-excitons is defined by the spin angular momentum of excitation (5) which can take two possible states (\pm). Consequently, nanostructures with equal responses to a valley-excitonic state and its mirror image, namely achiral nanostructures, emerge as favorable choice for valley-based information processing.

For instance, ~~chiral photonic nanostructures have been proven to selectively enhance and efficiently tailor emission from individual valleys. Coupling integrating 2D-TMDs to with~~ achiral dielectric metasurfaces demonstrated potential in controlling directionality, lifetime and spectral shape of the PL response. (6; 7) Importantly, Liu and coworkers further demonstrated that the DOCP can be enhanced (equally for σ_+ - and σ_- -polarized excitation) using Mie-resonant metasurfaces. (7)

3. Part of the premise of the study is based on an incorrect intuition that symmetric resonators should preserve local circular polarization when excited from the far field. Several sentences are problematic in this respect: 'Contrary to the simple intuition suggesting that a centrosymmetric nanoresonator preserves the degree of circular polarization in the farfield...' and 'By symmetry, the nanoparticle-on-substrate geometry is expected to preserve the helicity of valley-selective chiral emission.' Such statements are incorrect, as proven by the fact that nanoparticles can preserve or not the helicity of the incident field, as demonstrated in Nano Letters 23, 11, 5101 (2023). Additionally, the gold nanoparticle is 15 nm away from the monolayer. At the symmetry point below the center of the sphere, one could expect that the monolayer experiences out-of-plane electric fields due to the dielectric-metal interface. In the simulations in Figure 4, it is clear that circular polarization is decreased directly below the nanoparticle, whereas maximum intensity is achieved in a ring surrounding the nanoparticle with a circular polarization that is still not unity, so such a simulation disproves the naïve initial assumption about local circular polarization.

We agree with the reviewer that the statements in the original manuscript do not hold and appreciate the sharing of this reference. We have improved the respective statements in the manuscript by strictly referring to the conservation of angular momentum along the axis of cylindrical symmetry, i.e. an emitter in the center below the GNP and the DOCP in the direction out of the substrate (i.e. forward scattered direction).

As further pointed out correctly by the reviewer, the excitation field below the GNP and at the plane of the 1L-MoS₂ will also have out-of-plane components. However, these should not lead to any sizable PL contributions for our experimental conditions as discussed in detail below.

4. Does the absorption of out-of-plane electric fields play any role? The authors seem to neglect fields at the monolayer plane with this orientation, but is this justified by a different absorption coefficient for fields with that orientation? The authors cite the simulation reference 33 in this context, but is there any experimental evidence for the accuracy of such an approximation?

In short: It does not for 1L-MoS₂. – The orientation of luminescent excitons in 2D-TMDs has been studied in literature both theoretically and experimentally. Distinct results have been reported depending on the material and temperature. While molybdenum and tungsten based 2D-TMDs (i.e. MoX₂ and WX₂, respectively, with X=S, Se) show no out-of-plane contributions in PL emission at room temperature (8; 9), a mixture of in-plane and out-of-plane contributions was measured for tungsten based 2D-TMDs at cryogenic temperatures (10). However, this behaviour was not observed for molybdenum based 2D-TMDs (10) and it was thought that this is due to the fact that the spin-orbit splitting at K/K' appears with opposite signs for molyb-

denum and tungsten based 2D-TMDs (leading to an in-plane oriented exciton ground state for molybdenum based 2D-TMDs and an out-of-plane oriented exciton ground state for tungsten based 2D-TMDs (10; 11; 12)). While this holds for MoSe₂, it was shown that for MoS₂ the out-of-plane exciton (spin-forbidden dark exciton) actually lies energetically below the in-plane exciton (spin-allowed bright exciton) – not as for MoSe₂ but as for WS₂ and WSe₂ (13). Despite this energetic ordering of dark and bright excitons, in MoS₂ the (out-of-plane) spin-forbidden dark excitons are negligible even at low temperatures without the application of strong magnetic fields (above ~12 T) due to their much weaker oscillator strength as compared to the (in-plane) bright excitons and potentially due to low population of the excitonic state (see discussion by Robert et al. (13): the optical phonon energy in MoS₂ is larger than the dark-bright exciton splitting– not as for WS₂ and WSe₂). Hence, the out-of-plane fields in our system (1L-MoS₂) are negligible. We have added this information in the revised manuscript as follows:

- "Polarization-resolved cryo-PL measurements" (page 8, lines 149-152): *In 1L-MoS₂, the out-of-plane contributions from spin-forbidden dark excitons are negligible without strong external magnetic fields as shown by Robert et al. (13) Therefore, in equilibrium the local exciton densities, $n_K(x,y)$ and $n_{K'}(x,y)$, must will be proportional to $I_{\sigma^+}^{\parallel}(x,y)$ and $I_{\sigma^-}^{\parallel}(x,y)$, respectively, where the superscript \parallel points to the fact that only the tangential refers to the in-plane components of the external fields will contribute to the excitation of the material.*

5. Can the authors show the 2D-DOCP of a focused Gaussian beam for reference in the supplementary information? Is the reversal of DOCP also present without the nanoparticle?

We have added the corresponding figure as well as its short description to the SI (see section S6 in the revised version). Clearly, no reversal of DOCP is present without the nanoparticle.

Fig. S6: Helical intensity components in the nearfield of a gold nanoparticle. (a) Numerically calculated $I_{\sigma^+}^{\parallel}(x,y)$ and $I_{\sigma^-}^{\parallel}(x,y)$ at the position of the monolayer for different cases. Only halves of the intensities are shown due to symmetry and both sides are normalized by the same value. The left column corresponds to a focused Gaussian beam on a GNP, the middle column - to the case of a plane wave incident on a GNP, and right column - to the focused Gaussian beam in the absence of a GNP. (b) Resultant 2D-DOCP in x-y plane. The dotted circles highlight the projected GNP edge.

We refer to this observation in the main text: *Reference simulations of a tightly focused Gaussian beam without a GNP (see Sec. S6 of the Supporting Information) reveal that $I_{\sigma^-}^{\parallel}$ remains nearly zero, emphasizing that the slight increase of this component observed in the presence of a GNP is not due to high NA itself.*

6. It also looks like the areas with lower or sign-reverted DOCP in Figure 4 should not be very relevant because there is no intensity at those locations, so it is surprising that they have an effect on the observed polarization. Can the authors comment on this?

This question is closely related to the comment of the first reviewer, who also pointed out problems with the interpretation of our 2D DOCP plots. To address these concerns, we have made significant revisions to Figure 4 as well as Figure 5 and the corresponding textual descriptions. These changes were implemented to improve the clarity and accuracy of our findings, addressing the inquiries raised by both reviewers. We kindly ask you to review our response and the modified figures provided above.

7. The manuscript states that the work “unveils a serious impediment of the practical fabrication of resonant valleytronic nanostructures.” Is it an impediment or a constraint that needs to be taken into account when designing such a system for optimal polarization? For example, there could be resonant microcavities where the spatial invariance along the plane of the monolayer does not affect polarized emission in the way that a plasmonic nanoparticle does, so such resonant nanostructures could in principle enhance valley polarization.

The depolarization of PL emission measured and quantitatively modelled in our work can be addressed to the strong spatial sensitivity of emitters distributed in the nearfield of the plasmonic nanoparticle. As a consequence, emitters displaced only by a few 10s of nanometres already show a strongly modified PL response in the farfield compared to emitters below the center of the GNP. Generally, such effects are expected to translate in a similar fashion to other geometries and, as a consequence, pose two major problems for the design and tailoring of nanoantenna-based resonant valleytronic devices: (1) the simulation efforts drastically increase as the spatially averaged response of emitters in a plane needs to be calculated and (2) a priori assumptions on the collective response of a hybrid system based on symmetries may not hold (as seen on the example of this work). We argue that these problems are essential causes for the limited progress that could be observed in recent years in the field of nanoantenna-based valleytronics. Therefore, these findings state an impediment for valleytronic devices based on achiral plasmonic nanoantennas. At the same time, we think that the insights provided in our work will help other researchers to design and tailor new valleytronic devices with focus on nanoantenna-based approaches (e.g. by utilizing all-dielectric nanostructures).

As pointed out by the reviewer, one way to circumvent these problems is to integrate 2D-TMDs with (micro-)cavities where the field gradient along the plane of the crystal can be minimized. However, by changing the platform one also loses the advantages of nanoantenna-based approaches such as much smaller footprints and good coupling to free space radiation. The resulting use-cases therefore will also differ and are beyond the scope of this manuscript which focuses on nanoscopic systems. A brief outlook on how to circumvent these limitations in the framework of nanoantennas was added to the conclusion:

- *[...] While our work specifically shows the challenges and limitations for the realization of valleytronic devices on the basis of resonant plasmonic nanostructures, it hints at pathways to circumvent these limitations. These might range from optimizing the valley selective excitation of 2D-TMDs in the vicinity of resonant nanostructures by structured*

light excitation techniques or exploring alternative material platforms and resonance types for tailoring the collective emission response to nanostructuring of 2D-TMDs for precise emitter position control for the realization of functional nanoscopic valleytronic devices.

8. What is shown in Figure 2c and attributed to absorption? Did the authors measure transmission and reflection in order to calculate absorption? Or is it differential reflection?

The spectra in Figure 2c corresponds to the normalized differential reflectance. We appreciate the reviewer for bringing this to our attention. Initially, we referred to it as "absorption" because the imaginary part of the susceptibility, a measure of absorption, can be directly linked to the differential reflectance through the Kramers–Kronig relation. For a monolayer on a thick substrate, the imaginary part of susceptibility and differential reflectance are directly proportional (see the supplementary information in Roch (14) for a detailed description). However, your comment has prompted us to reconsider this point, as the presence of the spacer layer may affect this relationship. Therefore, to provide better accuracy, we have updated the text and the figure to refer to this spectra as differential reflectance instead of absorption as presented below:

The ~~white-light absorption~~ differential reflectance $\Delta R/R$ spectrum (black curve) of the embedded 1L-MoS₂, which was measured with a tungsten-halogen white light source, shows ...

Fig. 2: Optical microscopy and spectroscopy pre-characterization. (a) Optical brightfield and (b) darkfield microscope image of a prepared substrate incorporating embedded 1L-MoS₂ crystals and being decorated by several GNPs. The red circles indicate the positions of GNPs lying atop an embedded 1L-MoS₂ crystal. (c) Measured cryogenic ($T = 3.8$ K) ~~white-light absorption~~ differential reflectance $\Delta R/R$ (black curve) and PL (red curve) spectrum of an embedded 1L-MoS₂ crystal as well as the white light reflection spectrum of an isolated GNP (blue curve).

Also, the definition of the differential reflectance was added to the Methods section: *The differential reflectance spectrum in Figure 2c was obtained from the reflectance of the coated by silicon oxide 1L-MoS₂ and the reflectance of the coated bare substrate via*

$$\Delta R/R = \frac{R_{\text{MoS}_2} - R_{\text{Subs}}}{R_{\text{Subs}}}. \quad (1)$$

9. The methods section does not specify whether a beamsplitter was used for reflection-configuration photoluminescence measurements. If used, what are the polarizing properties and model of that component?

We have used a non-polarizing 30:70 plate beam splitter (BS) for all our cryogenic measurements. To address the inquiry of the reviewer, we have added the respective SI section (section S1 in the revised version) with the detailed characterization of the beam splitter using the Müller matrix formalism. On top of that we have expanded our analysis to include both the beam splitter and the mirror as critical factors affecting the polarization state of the circularly polarized laser beam. See below the corresponding SI figure:

Fig. S1: Stokes polarimetry in the cryostat setup. (a) Sketch of the optical setup in reflection mode. The numbers in round brackets next to the gray dashed lines denote different positions. (b) Intensity transmission of s- and p-polarized light over a broad spectral range. (c) Polarization ellipses of the 6 analyzed cases averaged over the wavelength range from 650 nm up to 670 nm. Here, "Input" refers to the polarization state measured without a BS while "Output" refers to the polarization state after passing the BS. The circles in the "Output" plot show the theoretically calculated ellipses based on the estimated Müller matrix and the input state. The line color encodes the corresponding DOCP for clear differentiation between left- and right-handed circularly polarized light.

Our analysis is further supported by the polarimetry measurements of the circularly polarized laser that passes through the aforementioned optical components as explained in the subsection S.1.3.

We refer to the section S1 in Methods: *The non-polarizing 30:70 plate beam splitter was utilized for all our cryogenic measurements. The sketch of our setup as well as the detailed characterization of polarization effects of the utilized optical components are provided in Sec. S1 of the Supporting Information.*

10. Supplementary Figure 1 shows an offset for the S2 Stokes parameter (and S1 to a lesser extent), which is not zero and changes sign with the excitation helicity. What is the origin of this apparent artifact? Also, a subtle detail is visible in the S1 and S2 images: the emission is slightly linearly polarized with lobes following the orientation of the polarization, corresponding to enhanced linearly polarized electric fields at different positions around the nanoparticle, which is to be expected for a metal nanoparticle.

Thank you for your insightful comments regarding the analysis of S_1 and S_2 in our manuscript. After we have performed the detailed study of the setup's polarization effects as described in section S1 of Supporting Information, we can confidently attribute the offsets of

S_1 and S_2 to the influence of the utilized optical components. Among other issues, our analysis confirms that the polarization state of the laser light undergoes similar transformations when interacting with these optical components. In the previous version of the manuscript, our data corrections accounted solely for the effects of the beam splitter, leading to an almost zero S_1 . This correction, however, did not adequately explain the observed shift in S_2 . During the revision process, we incorporated the impact of the mirror, particularly the phase shift between s- and p-polarizations, which we found to cause a shift in S_2 proportionally to the S_3 of the incoming light. This addition to our model has significantly improved the accuracy of our results. In the revised SI, section S2 we display the uncorrected data:

Fig. S2: Polarization resolved photoluminescence imaging. (a) Measured confocal scan of the total emission intensity \mathcal{I}_{tot} from embedded 1L-MoS₂ decorated with monodispersed GNPs upon σ^+ excitation and collected through a 660 nm bandpass filter. (b) Optical microscopy image of the same sample. In both images the position of the GNPs is indicated by circles. (c) Measured confocal scans of the Stokes parameters of emission from the same sample upon σ^- (top row) and σ^+ (bottom row) excitation. (d) Corresponding degree of polarization (DOP) for two excitation polarizations. All scale bars in (c) and (d) represent 2 μm .

In the corresponding description we explicitly discuss these offsets:

Furthermore, we observe the offsets of S_1 and S_2 induced by the utilized optical components as detailed in the previous section. These offsets align well with the behaviour of the initially circularly polarized laser light that passes through the same optical components (see subsection S.1.3). At the center of the GNPs, S_2 approaches nearly zero, affected by the DOCP of the circularly polarized PL due to mixing of the Stokes vector components by the mirror.

Additionally, we acknowledge the minor enhancement of the linear polarization component near the nanoparticle, as fairly noticed by the reviewer. However, this enhancement has a minimal impact on the overall degree of polarization. We discuss this effect in the description to Figure S2: *Around the nanoparticles, we observe subtle lobes of enhanced S_1 and S_2 indicating that it induces a mild linear polarization in its vicinity. Nonetheless, complete depolarization is evident at the center, as depicted in Figure S2 (d), where the degree of polarization is calculated using the formula:*

$$\text{DOP} \equiv \sqrt{S_1^2 + S_2^2 + S_3^2}.$$

This calculation reveals that the minor increase in linear polarization around the nanoparticle has a negligible impact on the overall degree of polarization, which is primarily affected by

the decrease in S_3 . Therefore, the alteration in S_3 is directly linked to changes in the degree of polarization.

We believe that these adjustments address the concerns raised and provide a more comprehensive understanding of the phenomena observed. We appreciate your guidance in refining our study and hope that our revisions meet your approval.

11. Do the Lumerical simulations include the presence of the MoS₂ monolayer? This is not specified in the methods.

No, the Lumerical simulations did not include the presence of the MoS₂ monolayer. We have added the explicit statement into the methods section: **We neglected the presence of the monolayer in our simulations.** Thank you for bringing this up. It would indeed be intriguing to investigate the potential influence of the monolayer's refractive index in our simulations.

12. Did the authors specify how they scan in their confocal images? Is the sample scanned or the beam?

In our confocal measurements, the sample is scanned inside the chamber of the cryostat by two piezo-controlled stages. In this way we can ensure that the excitation beam (and particularly its polarization) does not change during the scanning measurements. We have added this information to the revised manuscript as follows:

- "Materials and Methods", section "Optical experiments at cryogenic temperatures" (page 16, lines 323): *Lateral scanning was performed using two piezoelectrical nanopositioners moving the sample. This allows for stable conditions of the focused excitation with a fixed degree of circular polarization.*

13. Can the authors please define explicitly how they calculated 'the cross-sections and error corridors (shaded regions)' in Figure 6?

To address the reviewer's question, we have replaced one sentence of the old manuscript version by a more detailed description of our error calculation method: ~~The cross-sections and error corridors (shaded regions) are obtained by averaging over several GNPs as well as cross-section directions across the GNP.~~ *The cross-sections and error corridors (shaded regions) are computed as the mean value and the standard deviation, respectively. For our sampling we have used the DOCP shown in Figure 3 (c) (scanning) or Figure 4 (b) (wide field) and compared the cross-sections along x- and y- axes of three measured nanoparticles on a single flake (in total 6 samples per measurement scheme)*

14. I cannot see the logic of using the word "however" in the following sentence: "Other nanophotonic structures facilitated the generation of valley-polarized plasmon/photon-exciton polaritons opening new ways for valley control. However, of crucial importance for the development of on-chip valleytronic devices is directional routing ...". The paragraph makes sense without "however".

We have removed the word "However".

15. I believe that the following sentence is inaccurate: 'Finally, by drop-casting and subsequent spin-coating we sparsely distributed monodispersed GNPs'. Usually the term

drop-casting is used for a method that relies on placing a drop of liquid and letting it dry, which is incompatible with spin-coating.

We thank the reviewer for the careful reading and have made the following corrections in the revised manuscript:

- "Sample preparation" (page 5, line 91): ~~Finally, by drop-casting and subsequent spin-coating we sparsely distributed [...]~~
- "Materials and Methods", section "Gold nanoparticle deposition" (page 16, line 307): ~~Subsequently, the diluted suspension was drop-casted spin-coated onto the sample and spin-coated [...]~~

Note typos: non-degrative character, inducable degree of valley-polarization, Our numerical calculations closely agrees, spatial extend.

We thank the reviewer for the careful reading, the typos were removed in the revised manuscript.

Reviewer 3

Tobias Bucher and coauthors investigate spin-valley polarization in hybrid systems composed of gold nanoparticles and monolayer MoS₂. In their optical studies, the combined system is subject to strong depolarization. The experiments seem to have been conducted with care and are well presented. The manuscript is written in a clear, technical language, but is still a good read. The results are convincing. However, I cannot judge the novelty of the work as my focus is more on plasmonics. The combination has been investigated quite a bit and my guess would have been that polarization in such combinations would have been treated earlier. The guess might be wrong though. If I understood the model correctly, the dipolar nature of the exciton is not changed by the presence of the plasmonic nanoparticle? There is quite a body of work on strong coupling effects in 2D excitons strongly coupled to plasmonic cavities. The authors should detail on how to exclude changed emission properties by a changed exciton fine structure.

We thank the reviewer for the positive assessment of our manuscript and for the raised question about coupling phenomena in our plasmonic-TMD hybrid system. As evidenced by a plethora of works in literature around similar geometries, the exciton fine structure and exciton emission properties can be altered in plasmonic-TMD hybrid systems. A common interaction type in this respect is dipole-dipole interaction between the excitons in TMDs, which are of dipolar nature, and induced mirror dipole moments in the respective plasmonic nanostructures as recently theoretically studied by Salzwedel et al. (15) and Greten et al. (16; 17) We would like to point out that the polarization dependence of such coupling phenomena has not attracted the same attention yet and will be of interest for future studies. However, it is worth to note that the dipole-dipole interaction is very sensitive to the spacing between the interacting dipoles (r^{-6}) and substantial differences in the coupling strength can be expected when changing the spacer layer thickness in our system. As shown in section S4 of the Supporting Information, we have analyzed the depolarization effect for spacer thicknesses of 5, 15 (presented in manuscript) and 50 nm (see Fig. S4 below). Despite the large variation of the spacer thickness we always observe the same qualitative result in these samples. Hence, we do not expect

Fig. S4: Influence of the spacer thickness. (a-c) From left to right: Microscope images of the samples with highlighted scanning regions denoted by white squares, confocal scans showing the PL intensities for σ^+ and σ^- excitation, the total PL intensity, and the DOCP obtained for varying spacer thicknesses (5 nm, 15 nm, and 50 nm). The positions of the GNPs are indicated with circles, and short horizontal lines adjacent to these circles mark the locations where the cross-sections depicted in (d) are taken. (d) Cross-sections through the center of the GNPs showing total PL intensity (left) and the DOCP for the samples illustrated in (a-c).

coupling phenomena such as dipole-dipole interaction to play a dominant role as a cause for the observed PL depolarization in our system.

We referred to the respective section S4 of the Supporting Information in the main manuscript as follows:

- "Nearfield excitation polarization": [...] and second, we find the depolarization effect to be robust across all our samples naturally including different spacer thicknesses (see Sec. S4 of the Supporting Information) and GNPs of slightly with naturally varying size and topography.

A brief mentioning of current related works from the field of chiral plasmonics would be beneficial for the manuscript, too.

The concept of chiral Purcell enhancement is briefly introduced in the revised manuscript ("Introduction" page 3, lines 42-47, see also answer to question 2 of Reviewer 2). However,

the direct transfer of these concepts to the field of valleytronics poses new challenges due to the selective coupling to a specific valley. Or in other words, the handedness of the chiral nanostructure is fixed by design while the handedness of the exciton state is variable (as carrier of the valley information). We have added a respective discussion in the revised manuscript and limited our focus on achiral nearfield effects.

From the experimental point of view, the same experiment with off-resonant particles would be very interesting as well as measuring polarization for the combination driven off-resonantly.

In this work, we use resonant GNPs which feature a broad electric dipolar resonance across the emission spectrum of 1L-MoS₂. Due to the resonant excitation used in our experiments, we also have a significant overlap of the laser line with the GNP resonance. To experimentally verify that our results are not just a particularity of the chosen combination of geometry and experimental parameters, we have (1) shown a robust depolarization across various samples including a natural variation of the GNP size by about 5-10% and (2) repeated measurements for samples with varying spacer thickness. However, we find an identical quantitative effect on the DOCP of emission. The variation of the spacer thickness leads to a different overlap of the GNP's nearfield and the 1L-MoS₂ but leaves the resonant character of the GNP unchanged. Studying the system off-resonantly will lead to a similar situation (as for smaller sizes of the GNP the scattering behaviour remains to be dominated by a dipolar response even in the quasi-static limit). For larger GNP sizes, the increased absorption leads to a strong broadening and weakening of the resonance which will also suppress the observation of higher order modes. By using a shorter excitation wavelength, the experimentally observable DOCP of emission from bare 1L-MoS₂ will reduce due to intervalley scattering. However, the qualitative modification of the emission will remain the same as described by our physical model.

References

- [1] E. Hecht, *Optics* (Pearson Education India, 2012).
- [2] S. Yoo and Q.-H. Park, "Chiral light-matter interaction in optical resonators," *Phys. Rev. Lett.* **114**, 203003 (2015).
- [3] Y. Tang and A. E. Cohen, "Optical chirality and its interaction with matter," *Phys. Rev. Lett.* **104**, 163901 (2010).
- [4] Z. Wu, J. Li, X. Zhang, J. M. Redwing, and Y. Zheng, "Room-temperature active modulation of valley dynamics in a monolayer semiconductor through chiral Purcell effects," *Adv. Mater.* **31**, 1904132 (2019).
- [5] F. Caruso, M. Schebek, Y. Pan, C. Vona, and C. Draxl, "Chirality of valley excitons in monolayer transition-metal dichalcogenides," *The J. Phys. Chem. Lett.* **13**, 5894–5899 (2022). PMID: 35729685.
- [6] T. Bucher, A. Vaskin, R. Mupparapu, F. J. F. Löchner, A. George, K. E. Chong, S. Falsold, C. Neumann, D. Y. Choi, F. Eilenberger, F. Setzpfandt, Y. S. Kivshar, T. Pertsch, A. Turchanin, and I. Staude, "Tailoring photoluminescence from MoS₂ monolayers by Mie-resonant metasurfaces," *ACS Photonics* **6**, 1002–1009 (2019).
- [7] Y. Liu, S. C. Lau, W.-H. Cheng, A. Johnson, Q. Li, E. Simmerman, O. Karni, J. Hu, F. Liu, M. L. Brongersma, T. F. Heinz, and J. A. Dionne, "Controlling valley-specific

- light emission from monolayer mos_2 with achiral dielectric metasurfaces,” *Nano Lett.* (2023).
- [8] J. A. Schuller, S. Karaveli, T. Schiros, K. He, S. Yang, I. Kyriassis, J. Shan, and R. Zia, “Orientation of luminescent excitons in layered nanomaterials,” *Nat. Nanotechnol.* **8**, 271–276 (2013).
- [9] M. Brotons-Gisbert, R. Proux, R. Picard, D. Andres-Penares, A. Branny, A. Molina-Sánchez, J. F. Sánchez-Royo, and B. D. Gerardot, “Out-of-plane orientation of luminescent excitons in two-dimensional indium selenide,” *Nat. Commun.* **10**, 3913 (2019).
- [10] G. Wang, C. Robert, M. M. Glazov, F. Cadiz, E. Courtade, T. Amand, D. Lagarde, T. Taniguchi, K. Watanabe, B. Urbaszek, and X. Marie, “In-plane propagation of light in transition metal dichalcogenide monolayers: Optical selection rules,” *Phys. Rev. Lett.* **119**, 047401 (2017).
- [11] K. Kośmider, J. W. González, and J. Fernández-Rossier, “Large spin splitting in the conduction band of transition metal dichalcogenide monolayers,” *Phys. Rev. B* **88**, 245436 (2013).
- [12] A. Kormányos, V. Zólyomi, N. D. Drummond, and G. Burkard, “Spin-orbit coupling, quantum dots, and qubits in monolayer transition metal dichalcogenides,” *Phys. Rev. X* **4**, 011034 (2014).
- [13] C. Robert, B. Han, P. Kapuscinski, A. Delhomme, C. Faugeras, T. Amand, M. R. Molas, M. Bartos, K. Watanabe, T. Taniguchi, B. Urbaszek, M. Potemski, and X. Marie, “Measurement of the spin-forbidden dark excitons in mos_2 and mose_2 monolayers,” *Nat. Commun.* **11**, 4037 (2020).
- [14] J. G. Roch, G. Froehlicher, N. Leisgang, P. Makk, K. Watanabe, T. Taniguchi, and R. J. Warburton, “Spin-polarized electrons in monolayer mos_2 ,” *Nat. nanotechnology* **14**, 432–436 (2019).
- [15] R. Salzwedel, L. Greten, S. Schmidt, S. Hughes, A. Knorr, and M. Selig, “Spatial exciton localization at interfaces of metal nanoparticles and atomically thin semiconductors,” *Phys. Rev. B* **109**, 035309 (2024).
- [16] L. Greten, R. Salzwedel, M. Katzer, H. Mittenzwey, D. Christiansen, A. Knorr, and M. Selig, “Dipolar coupling at interfaces of ultrathin semiconductors, semimetals, plasmonic nanoparticles, and molecules,” *physica status solidi (a)* **221**, 2300102 (2024).
- [17] L. Greten, R. Salzwedel, T. Göde, D. Greten, S. Reich, S. Hughes, M. Selig, and A. Knorr, “Strong coupling of two-dimensional excitons and plasmonic photonic crystals: Microscopic theory reveals triplet spectra,” *ACS Photonics* **11**, 1396–1411 (2024).

Point by point response to the Reviewers: Influence of resonant plasmonic nanoparticles on optically accessing the valley degree of freedom in 2D semiconductors

**T. Bucher, Z. Fedorova, M. Abasifard, R. Mupparapu, M. J. Wurdack,
E. Najafidehaghani, Z. Gan, H. Knopf, A. George, F. Eilenberger, T.
Pertsch, A. Turchanin, and I. Staude**

E-mail: tobias.bucher@uni-jena.de and zlata.fedorova@uni-jena.de

We again thank all the reviewers for thorough reading and their constructive comments. We have read all reviewer's comments carefully and answered them point-by-point below. The color convention in our response remains the same as in the previous revision stage. Namely, the original reviewer comments in **blue**, the corresponding answers in **black**, and the changes to the manuscript in **red** where the struck out parts were removed and the italic parts were newly added. Where appropriate, page and line numbers are given which refer to the respective position in the originally submitted manuscript. For the revised version please refer to the file with tracked changes.

Reviewer 1, Xavier Zambrana-Puyalto

In the revised version of Influence of resonant plasmonic nanoparticles on optically accessing the valley degree of freedom in 2D semiconductors, Tobias Bucher et al. have given many more details about their experimental and theoretical findings. I think that the article has improved as a result. However, after reading the response to all the referees comments, the revised manuscript and the SI, my feeling is that the article still needs to be polished. In my opinion, the most important issues that need to be addressed are the following:

- The authors mention that the decrease in DOCP that is observed goes hand in hand with a reduction of the DOP. That is, below the GNP, the three Stokes parameters $\{S_1, S_2, S_3\} \approx 0$. However, it seems that the Stokes parameters (especially S_3) are clearly different from 0 at the flake positions where the GNP are not present, thus leading to a $DOP > 0.8$. I find this quite puzzling. It seems as if a simple scattering process (photoluminescence emitted photons from 1L-MoS₂ interacting with the GNP) makes the DOP of the emitted light go to zero. A $DOP \approx 0$ is typically a sign of incoherent light. Do the authors understand what is going on? I think that this is a crucial point for this article. Because their theoretical model, based on the exciton emission and GNP scattering, assumes that the light is coherent. That is, the rotating dipole model that is used to understand their measurements is a coherent model. As a result, a reduction of DOCP (or a decrease in S_3) necessarily leads to an increase of $\{S_1, S_2\}$, as the DOP must remain 1. However, the authors do not measure this. This makes me think that the interpretation of their experimental results might be missing something. I am not used to working with the polarization of incoherent or partially coherent light. So I might get this wrong. But linear scattering of coherent light does not change its coherence. This is why I cannot understand why a material that emits with $DOP > 0.8$ would turn into a material that emits with $DOP \approx 0$ upon interaction with a GNP. And then, I thought that maybe what is happening is that basically the intensity that is being measured is almost 0 - due to the

near-field created below the GNP, or due to the confocal nature of their measurement. But looking at Fig. S4 or Fig. 3, this does not seem to be the case. So I am left wondering what is going on. I think that the authors should try to address this point. As the whole motivation of the paper revolves around the fact that $\text{DOCP} \approx 0$, but in fact, the authors also measure that $\text{DOP} \approx 0$, but kind of decided not to pay much attention to that.

We thank the reviewer for bringing this to our attention and helping us to realize the gaps in our explanation. It is important to emphasise that we have assumed the light emitted by the TMD flake to be incoherent both spatially and temporally. The reason is that from each single point in space we collect numerous uncorrelated emission events since the excitonic coherence time (~ 0.1 ps, (1)) is incommensurately shorter than the detector's exposure time (~ 0.1 s). As fairly noted by the reviewer, the rotating dipole model is a coherent model and the computed farfields have well-defined polarization state for each emission direction. However, our assumption of temporal incoherence implies that there is no fixed phase relation between the light emitted in different directions even though it was emitted from the same point in space. As a result, when we average the contribution of each rotating dipole over all emission angles, we integrate the farfield intensities rather than complex farfield vectors. As a consequence, the integrated intensities do not represent a coherent state of light anymore and allow for the degree of polarization to be less than 1.

To clarify this point we have added the following sentence to the section "*Farfield emission polarization*":

In our calculations we assume that the emission from 1L-MoS₂ is both spatially and temporally incoherent (see Methods for more details).

Further, we add more details to the methods section: *Each FDTD simulation provides the field components in spherical coordinates $\{E_\theta(r, \mathbf{u}), E_\phi(r, \mathbf{u}), E_r(r, \mathbf{u})\}$, where r is the emitter displacement and $\mathbf{u} = (u_1, u_2)^T$ denotes the emission direction in terms of a direction unit vector as defined in Lumerical FDTD Solutions. [42] An angular filter was applied to the resulting field restricting the accepted emission angles to fit the finite numerical aperture of our objective ($|\mathbf{u}| < \text{NA}$, $\text{NA}=0.9$). Next, we transformed the filtered field given in spherical coordinates $\{E_\theta, E_\phi, E_r\}$ into the helical basis $E_{\sigma^\pm} = E_\theta \pm iE_\phi$. The corresponding intensities are given by $\mathcal{I}_{\sigma^\pm}^K(r, \mathbf{u}) = |E_{\sigma^\pm}^K(r, \mathbf{u})|^2$, where we have added the valley index K and dependence on the emitter displacement. In the following we assume that emission from each point in space is temporally incoherent because we observe the system on timescales ~ 0.1 s incommensurately longer than the exciton coherence times ~ 0.1 ps (1). Consequently, there is no fixed phase relation between the radiation into different directions and we average by integrating the intensities (not fields) over all emission angles $\mathcal{I}_{\sigma^\pm}^K(r) \propto \iint \mathcal{I}_{\sigma^\pm}^K(r, \mathbf{u}) d\mathbf{u}$.*

To ensure that our model accurately predicts the depolarizing behavior, we have included a detailed step-by-step computation of all the Stokes parameters (S_1 and S_2) and the degree of polarization in the newly added SI section S7. In Figure S7, we present results for homogeneously distributed incoherent point emitters (analogous to Fig. 5 in the main text), where we observe that the linear Stokes components can reach up to 0.59 near the GNP, while the DOP decreases to 0.31 at a distance of 35 nm. In Figure S8, we provide the final result that accounts for the nearfield excitation conditions and the optical resolution for both circular excitation polarizations. Due to the finite optical resolution and mixing of K and K' emitters, the maximum values of S_1 and S_2 reduce to less than 0.04, consistent with the measurements presented in section S2. These results further

Fig. S7: Linear polarization properties of incoherent σ^+ -rotating dipoles coupled to a GNP (a) Relation between the direction unit vector coordinates used by far field projections and the field components in spherical coordinates. The circle limits the NA of the objective. (b) Sketch of the system's geometry showing that K and K' emitters are related by reflection around the symmetry axis (here, x -axis). (c) Calculated farfield intensities \mathcal{I}_0^K and \mathcal{I}_{90}^K and their sum $\mathcal{I}_{\text{tot}}^K$. The dotted circle denotes the projected GNP edge. (d) Spatially-dependent S_1^K . (e) Radial dependence of the DOP for homogeneously distributed incoherent K -emitters. The shaded region corresponds to the GNP size.

confirm that the reduction in DOP near the nanoparticle is primarily due to the decrease in DOCP. We refer to Sec.S7 in the main text in the end of the section "*Full model for optically addressing the valley degree of freedom*": *Additionally, in Sec. S7 of the Supporting Information, we provide detailed computations of the expected values for all Stokes components and the total degree of polarization. These results further validate that our model accurately predicts the overall depolarizing effect of the nanoparticle, in strong agreement with experimental observations.*

- I think that Referee 2 was right in pointing out that there were some issues with Fig 1 of the SI. The authors have created a new section S.1 in the SI, which I like. I think that it is a good addition to the paper. But I am wondering, if the authors are trying to measure the polarization properties, why do they base their measurements and analysis on the detection of some Stokes parameters that mix orthogonal states? I am referring here to the results presented in the table of S.1.3. In my opinion, the authors should have added some extra quarter and half wave-plates to compensate for the polarization modification caused by all the optical elements after the microscope objective. That is, typically they should make sure that Stokes parameters of the laser are $\{S_1, S_2, S_3\} = \{0, 0, \pm 1\}$ both before focusing (position 1) and after collection. As mentioned in S.1.1, non-polarizing beam-splitters do not tend to affect much the polarization state of beams, therefore this could be measured after the BS as well, in case measuring it in position 1 was not possible (I am talking about the reflected beam). If this was the case, then they should make sure that this is still the case at position 3, i.e. after the mirror. This will not happen, as explained in S.1, since the 45 degree mirror changes polarization. So then, after the mirror, they should add some polarization compensation so that this is still respected. That is, after the compensation elements (probably a system of QWP-HWP-QWP), an

Fig. S8: Calculated Stokes parameters and DOP. Numerically computed spatial variation of Stokes parameters and the resulting degree of polarization in the vicinity of a GNP for σ^+ and σ^- excitation polarization. Here, the scanning measurement scheme is considered. The dotted circle denotes the projected GNP edge.

incoming laser with a Stokes vector $\{S_1, S_2, S_3\} \approx \{0, 0, \pm 1\}$ should remain invariant for both left and right CP reflected laser beams. And same for linear polarizations. Ideally, the same should hold true for the photo-luminescence wavelengths. But depending on the QWPs and HWPs used, the change of wavelength could affect more or less this orthogonality. This is something that needs to be taken into account. For example, if we look at Fig.S3, and we are using a set-up that does not respect orthogonality, we cannot be sure the polarization contrast between the PL and the laser is a real physical effect or it is just the result of different chromatic effects given by the optical elements before the Stokes parameters measurement took place.

We thank the reviewer for his insightful suggestions, and we fully acknowledge the importance of careful consideration of polarization modifications introduced by optical elements. These insights will guide our future experiments and we will continue to refine our setup to further improve polarization detection.

However, the setup that has been used for the experiments presented in this paper has been rigorously characterized, particularly with respect to the polarization optics. We have carefully measured the Müller matrix of the beam splitter, estimated the influence of the mirror, and considered the small effects of their mutual misalignments by measuring the polarization modifications of almost perfectly circularly-polarized laser light. We ensured that our measured S_3 stays proportional to the S_3 leaving the sample within acceptable margins of error.

Regarding potential chromatic artifacts, we have taken steps to minimize them. Specifically, we used superachromatic waveplates from Thorlabs (SAQWP05M-700 for the wavelength range 325 - 1100 nm) and verified that the beam splitter's response shows minimal wavelength dependency within the 600-750 nm range (Fig.S1 (b)). Additionally, we do not expect significant spectral variation in the mirror's optical response, as silver does not exhibit strong resonances in this range.

While no experimental setup can be ideal, we are confident that our measurements reflect real physical effects rather than artifacts introduced by the optical system. We have devoted considerable effort ensuring the robustness of our polarization measurements and

confirming that our results accurately represent the phenomena under investigation.

- Related to the previous comment, in Fig. S3 we observe that the PL emission for left and right CP illumination has very similar values for longer wavelengths. Again, is this a detection (chromatic) issue, or is this supposed to be like that? Is this one of the reasons why the DOCP is 0.8 for a bare 1L-MoS₂ flake? In theory it should almost be 1, right?

We suppose that the reviewer meant "detection" instead of "illumination" because in Fig.S3 the illumination polarization does not change. The PL observed at longer wavelengths (>670 nm) primarily originates from defect-trapped excitons within the bandgap, which emit predominantly unpolarized light (see e.g. (2)). This explains why the PL for both left and right circularly polarized detection appears similar at the longer wavelengths. As explained above, we do not expect any considerable chromatic artifacts from our optical setup.

Concerning the DOCP of a bare 1L-MoS₂ flake, it can vary considerably across different samples from 32% up to 100% for the same temperature and excitation wavelength as we have used in our experiments (compare (2) and (3)). In Ref. (4) such a wide range of values was quantitatively explained by a model that takes into account the energy of the longitudinal acoustic phonons and the emission energy of the A-exciton. The latter is very sample-dependent and is influenced by numerous factors, including the dielectric environment, the presence of defects, and the specific fabrication process. Thus, the measured DOCP of 0.8 is within the expected range.

Apart from this, I also have a number of comments:

1. In lines 16-17, it is stated that that the light emitted from the nanoparticle position. I understand what the authors mean, but to avoid any misunderstandings it would be better to write that the nanoparticle scatters light. Emission, in this article, carries the meaning of photo-luminescence, and thus absorbing light at one frequency and emitting at a different one. This is not what the nanoparticle does.

We have modified the sentence according to the reviewer's suggestion: ... *our cryogenic photoluminescence microscopy reveals that the light emitted from the nanoparticle position scattered by the nanoparticle is largely unpolarized, i.e. we observe depolarization.*

2. In line 21, I would change strongly reduces to 'is strongly reduced'.
The suggested revisions have been incorporated: *In doing so, we find that the farfield degree of polarization strongly reduces is strongly reduced in the hybrid system...*
3. In lines 18-23 - unless I do not properly understand these two sentences - the authors make a reference to their near-field excitation + scattering model. But as I mentioned before, I do not think that this models predicts any kind of depolarization - understanding depolarization as a reduction of DOP. The model predicts a reduction of the DOCP, but it says nothing about the reduction of DOP. I think that the authors should try to explain the reduction of DOP, but if they don't, then they should at least rephrase these sentences to make it clear that their efforts are focused on the DOCP decrease (and not in the DOP reduction).

We have addressed this comment by answering the first question of the reviewer and confirmed that our model does predict the reduction of DOP in the SI section S7.

4. In lines 22-23, I do not exactly know what the authors mean.

In that sentence, our intention was to emphasize that predicting the polarization behavior of a hybrid system requires more than just considering a single emitter at the symmetry point of the structure, as was commonly done previously. Instead, it is crucial to account for the radially-dependent excitation and emission processes. However, in the revised version of the manuscript, this point was expanded to additionally highlight the significance of emission incoherence.

We rigorously study the nature of this phenomenon numerically considering the monolayer-nanoparticle interaction at different levels including *spatially-dependent excitation and emission processes*. When assuming the spatial and temporal incoherence of valley-selective emission, ~~In doing so~~, we find that the farfield degree of polarization ~~strongly reduces~~ is strongly reduced in our hybrid system ~~when including excitons emitting from outside of the system's symmetry point, which in combination with depolarisation at the excitation level causes the observed effect~~.

5. In line 33, maybe the sentence can form that may be used could be changed to 'can form and may be used'.

The relevant adjustment has been made: *...excitons with distinct spin states can form that and may be used...*

6. In line 59, this has already been mentioned. It might be a bit confusing to call the valley excitons chiral or to talk about the handedness of their emission. They emit with a well-defined total AM, and as a result, they have (approximately) one handedness in one semi-space, and the opposite handedness in the other semi-space.

We thank the reviewer for careful reading. We corrected this issue by removing the word "chiral".

7. In line 93, it is the only place in the article where the ellipticity of the GNP. Was this measured? Or it is just an assumption that they cannot be perfect spheres?

We have studied the sizes, shapes and distribution of GNPs and have added the section "Scanning electron microscope imaging of deposited gold nanoparticles" to the Supporting Information:

We have studied the size, shape and distribution of gold nanoparticles after deposition by means of scanning electron microscope (SEM) imaging. For this, we have deposited GNPs on a separate glass substrate coated with 15 nm of indium tin oxide as conductive layer to prevent charging of the sample during the imaging. Figure S2 (left) shows the respective top-view SEM image of a typical sample region. Our deposition process results in a homogeneous distribution of mostly isolated GNPs with few clusters of two or more GNPs. By locating 1168 individual GNPs using a circle detection algorithm, we have obtained an average GNP size of (220 ± 15) nm. On the right side we show a zoomed in image of the region indicated by the red box in order to identify the shape of individual GNPs. Slight deviations due to a finite ellipticity can occur for individual GNPs. Generally, elliptical nanoparticles can introduce a finite degree of linear polarization of the PL from emitters in their vicinity. However, in our PL measurements,

Fig. S9: Scanning electron microscope imaging. Top-view scanning electron microscope image of gold nanoparticles deposited onto a glass substrate coated with indium tin oxide. A zoomed in image of the region indicated by the red box is shown on the right side.

we did not notice any significant enhancement of the S1 or S2 Stokes parameter at the position of the GNPs. Hence, the GNPs can be approximated as spherical with reasonable accuracy. Note that clusters of more than one GNP can also readily be distinguished from isolated GNPs by optical darkfield microscopy due to their different shape. Similarly, GNPs of significantly different sizes can be distinguished in darkfield microscopy by their different color impression due to the spectral shift of the plasmonic resonance. This allowed us to limit our PL measurements to 1L-MoS₂ decorated only with isolated GNPs with similar sizes.

8. In line 94, instead of a predominating, I would write something like 'predominantly'. We have substituted "predominating" by "predominantly": **This leads us to the conclusion that the observed effect predominantly represents a predominating depolarization.**
9. In line 114 and in many other places we are told about the 1L-MoS₂ crystals. Isn't this a bit of a weird way of calling them? In general, crystals are 3D structures. I would simply call the 1L-MoS₂ layer a '1L-MoS₂ flake'

We thank the reviewer for his suggestion. While we understand that the term "crystal" traditionally refers to 3D structures, in the context of 2D materials, the term "2D crystal" is widely used in the literature to describe monolayer materials with an ordered atomic arrangement, including few-layer transition metal dichalcogenides (5). The term "crystal" in this case emphasizes the periodicity and atomic order, regardless of the dimensionality. Remarkably, even one of the earliest papers on few-layer MoS₂ from 1966 also refers to these structures as "crystals" (6). Therefore, we believe that referring to 1L-MoS₂ as 2D crystals is appropriate and aligns with established terminology.

10. In line 125-126, I would write see Methods when the differential reflectance $\Delta R/R$ magnitude is introduced. I was not sure what that meant, but then I could find it in Methods.

We have revised this part accordingly:

Next, we characterized the optical properties of the prepared sample at cryogenic temperature. as shown in Figure 2c (see Methods). The Figure 2c shows the measured differ-

ential reflectance $\Delta R/R$ spectrum $\Delta R/R$ (black curve) of the embedded 1L-MoS₂, which was measured ~~with~~using a tungsten-halogen white light source (see Methods).

11. In line 129, shouldn't the absorption peaks be 'reflection peaks'?

We have revised this part accordingly:

The pronounced ~~absorption~~ reflectance peaks...

12. In line 130, maybe it would be good to give a reference that explains what the A- and B-excitons are.

We have added [11] as reference for the notation of A- and B- excitons.

...peaks at 645 nm and 595 nm wavelength which are related to the A- and B-excitonic resonances formed at the direct bandgap in the K/K' points of the Brillouin zone [11].

13. In line 148, I guess that this is sort of understood, but the σ^+ polarization is not the polarization that pumps the sample, but rather the polarization of the laser beam before it is focused. Maybe this could be specified by writing ' σ^+ polarization before focusing '.

To clarify, the σ^+ polarization refers to the polarization of the laser beam that actually pumps the sample. This polarization is measured before the objective, as indicated at position (1) in Fig. S1 of the SI. After reflection from the sample, the sign of the Stokes parameter S_3 flips, and it flips again upon reflection from the mirror. Thus, at position (3), we measure the same σ^+ polarization as before focusing. Furthermore, the photoluminescence (PL) measured at position (3) exhibits the same sign of S_3 as the reflected laser. To avoid any ambiguity we refer to the SI section where we explain it in the details and specify our notations.

We have detailed the polarization control and notation of our experimental setup in Sec. S1 of the Supporting Information. In this notation, the DOCP of the incoming excitation laser and the measured PL emission will have equal sign.

14. In lines 182, the authors mention that *the excitation rate of carriers in valleys K and K' is proportional to the intensity of the σ^+ and σ^- polarized components of the external fields*. Here, I assume that external fields mean incident field + scattered. However, in line 194-195, it is stated that *Figure 4a shows the resulting total in-plane intensities generated by a focused Gaussian beam (left) and a plane wave (right)*. I am pretty sure that the authors refer to the field that is generated as a superposition of illumination and scattering. But one could get confused, so I think that they should specify that they are talking about their external fields, which are illumination + scattered field.

This point has been clarified and modified as requested:

Figure 4a shows the resulting total (incident + scattered) in-plane intensities generated by a focused Gaussian beam (left) and a plane wave (right).

15. In line 204, it is stated that *Notably, this cross-polarized component is slightly larger for the focused beam.* I guess that the comparison is relative. That is, they have normalized the maximum of the $I_{\sigma^+}^{\parallel}$ to 0.8 (or 1) for both illuminations and they are comparing how the other components is respect to the dominant one. However, the sentence could possibly mislead somebody to believe that the comparison is in absolute terms, i.e. in absolute emitted power. But that wouldn't make sense, as there will be much more power emitted in the Gaussian illumination case, if both Gaussian beam and plane wave are carrying the same power. Anyway, I think that it would be good to make sure that this is understood.

This concern has been addressed in the updated version.

Notably, when normalized to the maximum of the total in-plane intensity, this cross-polarized component is slightly larger for the focused beam.

Note, that the way we normalize these plots was specified in the caption of Figure 4b. The normalization is chosen such that the sum of both curves peaks to 1.

16. In Fig. 4, I think that it is a bit confusing because if one looks at the intensity plots of (a), you get the impression that there is no intensity right in the centre of symmetry. However, we see that the intensity is not 0 in (b). Then, in Fig. S6 this is clear. But maybe a reference to Fig. S6 could be given so that one is not under the impression that there is a 0 of intensity in the centre?

We have clarified this in the revised manuscript with the following sentence:

In both cases, the maximum intensity is concentrated in a ring around the GNP, while directly below the GNP the intensity is lower but non-zero (see Fig. S6 of the Supporting Information for reference).

17. In line 226, it says Figure 4b, but I think that it should be Figure 4c.

We have corrected this typo:

Figure 4c shows the respective measured DOCP image of PL from the same 1L-MoS₂ sample as shown before and measured with an average excitation power of 200 μW

18. In line 229, we are told that DOCP = 0.72 ± 0.02 when there are no nanoparticles. This is related to what I already mentioned before in the beginning of the review, but why isn't this value almost 1? Maybe a comment should be written about it?

As mentioned earlier, the DOCP of a bare monolayer 1L-MoS₂ flake can vary significantly due to factors such as the dielectric environment, defects, and the specific fabrication process, as discussed in Ref. (4). To make this clear to a broader audience, we have added the following sentence to the manuscript to the section *Polarization-resolved cryo-PL measurements*, where this value appears for the first time:

For 1L-MoS₂ without GNPs, we measured an average DOCP of 0.71 ± 0.03, which is consistent with values reported in the literature. The DOCP in 1L-MoS₂ has been shown to vary significantly from 32% up to 100%, even under the same experimental conditions due to sample-specific factors such as the dielectric environment, defects, and fabrication processes [33].

19. In lines 230-231, we are told about the prediction of ΔDOCP from their near-field simulations. But I am not sure if we have been told what these ΔDOCP simulations are computed, once we have obtained the near-fields. In methods, some details are given about how to compute the whole model, but I do not think that we are told how this intermediate step is computed.

Indeed, the description of how the predicted ΔDOCP is computed from the near-field simulations could benefit from further clarification. The step that may have been unclear is that we directly relate a change of the DOCP of the in-plane nearfield to a change of the DOCP of the farfield, but no further intermediate steps were involved in the process. To ensure that our computation method is transparent, we have revised the section "Nearfield excitation polarization" as follows:

~~While the lower ΔDOCP is qualitatively in line with the prediction from our nearfield simulations, the observed difference is, however, still significant. For a quantitative~~

~~comparison of both excitation schemes on the basis of our numerical simulations, we need to take into account the finite optical resolution in our experiments. For a quantitative prediction of the Δ_{DOCP} for both excitation schemes we have to relate the degree of valley polarization η to the DOCP of emission in the farfield. In order to do so solely based on the numerically calculated excitation nearfield, we initially neglect scattering of the emission by the GNP. Additionally, we take into account the finite optical resolution in our experiments. [...]~~

~~[...] Importantly, during scanning the beam position relative to the GNP center changes during the scanning such that the calculated nearfield intensity distribution $I_{\sigma\pm}^{\parallel}(r, x, y)$ and the respective exciton density distribution $n_{K/K'}(r, x, y)$ become dependent on the displacement distance r of the beam from the center of the GNP. Consequently, we obtain the averaged helical intensities $\tilde{I}_{\sigma\pm}(r) \propto \iint I_{\sigma\pm}^{\parallel}(r, \xi, \eta) d\xi d\eta$ by integrating over the intensity spot for each beam position. Therefore, the averaged helical excitation intensities at a distance r from the center of the GNP can be estimated as $\tilde{I}_{\sigma\pm}(r) \propto \iint I_{\sigma\pm}^{\parallel}(r, \xi, \eta) d\xi d\eta$, where $I_{\sigma\pm}^{\parallel}(r, x, y)$ are the respective nearfield intensities resulting from the excitation with the focused beam that is displaced by the distance r from the GNP center. Finally, the numerically predicted farfield PL DOCP we estimate the DOCP of PL in the farfield for each measurement setting is computed using Equation 5 both excitation schemes by substituting $\tilde{\mathcal{I}}_{\sigma\pm} \rightarrow \tilde{I}_{\sigma\pm}$ in Equation 5 and evaluating it for $r = 0$.~~

20. In lines 239-240, I do not fully understand why the authors do this integral and where they use it. I think that more details should be given about it.

The respective paragraph has been revised. Please refer to the answer of comment 19 above.

21. In lines 243-244, I do not fully understand how these results are obtained. I think that more details should be given about it too.

The respective paragraph has been revised. Please refer to the answer of comment 19 above.

22. In lines 273-274, it is not accurate that a rotating dipole emits circularly polarized light into different half spaces. It is approximately true in the far-field, though. That is, approximately it emits light of one helicity into one semi-space and the opposite helicity in the other semi-space. Polarization and polarization in the far-field (helicity) are not the same thing. The GNP, when illuminated by a tightly focused Gaussian beam that was CP before focusing, also behaves as a rotating dipole, and clearly does not create a near-field that is circularly polarized, as both CPs and also a \hat{z} component are present. However, in the far-field, it is approximately true that that its radiation pattern is also similar to the one shown in Fig.5(a).

This concern has been addressed as suggested by the reviewer.

~~Due to its fixed axis of rotation, \vec{p}_K approximately emits light of one helicity into one half-space and the opposite helicity in the other half-space emits circularly polarized light with opposite handedness into different half-spaces matching the PL polarization properties of valley-selective excitons in bare 1L-TMDs (see Sec. S3 of the Supporting Information).~~

23. In Figure 5, could the authors mention something about their current normalization? Now the total intensity goes to 1.5, which is a bit unconventional, even if there is no problem to leave it like this provided some details are given.

We would like to clarify that all far-field intensities in Figure 5b are normalized to the intensity at infinite displacement ($r = \infty$). This corresponds to the case where the rotating dipole is so far from the GNP that its field no longer interacts with the scatterer. Consequently, as r increases, the orange curve, representing the total intensity $\mathcal{I}_{\sigma^+}^K + \mathcal{I}_{\sigma^-}^K$, asymptotically approaches 1. The value of 1.5 indicates that the total intensity emitted by a dipole at a specific distance from the GNP center is 1.5 times higher than the total intensity emitted in the absence of the GNP.

The details of this normalization are already provided in the main text:

- Section "Farfield emission polarization": *Here, all the farfield intensities are normalized to the emission of a rotating electric dipole without GNP, i.e. at infinite distance.*
- Additionally, the caption of Figure 5b explicitly states *The farfield intensity is normalized to the intensity obtained without a GNP.*

We hope this addresses the concern.

24. In lines 321-324, I guess that the sentence *Similarly, we have [...] are obtained as [...] is missing a comment on the fact that this depends on the radial position, right? In particular, when the emitter is far away from the GNP, the DOCP seems to be 1 (Fig.5(b)).*

We have added the argument r and emphasised the radial dependency of the farfield intensities.

Similarly, we have discussed above that excitons located in the valley K (or K') at a distance r from the GNP's center lead to both σ^+ and σ^- polarized PL intensities in the farfield which we denoted as $\mathcal{I}_{\sigma^\pm}^K(r)$ (or $\mathcal{I}_{\sigma^\pm}^{K'}(r)$), respectively, where the farfield intensities from emitters in opposite valleys are obtained as $\mathcal{I}_{\sigma^\pm}^{K'}(r) = \mathcal{I}_{\sigma^\mp}^K(r)$.

25. In line 389-390, it is stated a cylindrically symmetric system conserves the circular polarization state in the direction out of the substrate. Similar comments are spread across the article. I think that generally speaking they are correct. But if read carefully, it seems as if the authors were claiming that cylindrical symmetry is related to preservation of circular polarization. And this is not true, in general. It is true that a cylindrically symmetric scatterer preserves circular polarization in the forward direction, at 0° [1, 2]. And it is also true that it preserves the total AM. But again, as shown by the authors in Fig. S6, a GNP does not preserve circular polarization (especially in the near-field). The sentence would be approximately correct if they talked about the far-field polarization state in the backward semi-space. I say approximately correct because it is not literally true that there is no scattered light with the opposite polarization in the backward semi-space. But it is true that it is negligible. Anyway, I think that the authors need to be more careful with these sort of statements.

We have slightly modified the sentences of this kind throughout the text to ensure they remain correct.

- In the abstract *Contrary to the simple intuition suggesting that a centrosymmetric nanoresonator mostly preserves the degree of circular polarization in the forward and backward scattered farfield by angular momentum conservation...*
- In the section *"Farfield emission polarization"* *By symmetry, the nanoparticle-on-substrate geometry is expected to largely preserve the circular polarization of valley-selective emission in the forward and backward scattered farfield.*
- In the conclusion *We reported a robust and strong reduction in the DOCP mediated by the GNPs which is in contrast to the expectation that a cylindrically symmetric system predominantly conserves the circular polarization in the farfield in the direction out of the substrate.*

26. In the subsection **Optical experiments at cryogenic temperatures** (as well as in the main text), the authors do not mention anything about how they obtained the spectral measurements of the GNP's scattering. Could they give some information about it?

We thank the reviewer for pointing out the missing information regarding the experimental details. We have revised the respective part of the Methods section as follows:

~~For white light spectroscopic measurements, a stabilized tungsten-halogen white light source was used. The differential reflectance spectrum in Figure 2c was obtained from the reflectance of the coated by silicon oxide and the reflectance of the coated bare substrate via Reflectance spectra were measured from 1L-MoS₂ as well as from isolated GNPs by limiting the illumination area using an iris aperture. The differential reflectance spectra in Figure 2c were then obtained by referencing to the reflectance of a bare region of the coated substrate via~~

$$\Delta R/R = \frac{R_{\text{MoS}_2/\text{GNP}} - R_{\text{Subs}}}{R_{\text{Subs}}}. \quad (1)$$

Differential reflectivity spectroscopy is an established method to characterize surfaces and thin-films incorporating nanoparticles (7; 8) and has also been utilized to characterize the scattering behaviour of single optical nanoantennas (9; 10). For clarity, we have relabelled the curve in Figure 2c (~~Seat-GNP~~ $\Delta R/R$, *GNP*) to properly refer to the used measurement technique (see below).

Fig. 2: Optical microscopy and spectroscopy pre-characterization. (a) Optical brightfield and (b) darkfield microscope image of a prepared substrate incorporating embedded 1L-MoS₂ crystals and being decorated by several GNPs. The red circles indicate the positions of the GNPs lying atop overlapping with an embedded 1L-MoS₂ crystal. (c) Measured cryogenic ($T = 3.8\text{K}$) differential reflectance spectra $\Delta R/R$ of embedded 1L-MoS₂ (black curve) and of isolated GNPs on a bare region of the substrate (blue curve), and PL spectrum (red curve) spectrum of an embedded 1L-MoS₂ crystal as well as the white light reflection spectrum of an isolated GNP (blue curve).

We have further revised the section "Sample preparation" in the manuscript as follows:

Next, we characterized the optical properties of the prepared sample at cryogenic temperature, as shown in Figure 2c (see Methods). The Figure 2c shows the measured differential reflectance $\Delta R/R$ spectrum $\Delta R/R$ (black curve) of the embedded 1L-MoS₂, which was measured with using a tungsten-halogen white light source (see Methods). τ shows We observe two distinct peaks [...]

We further verified the resonant character of the deposited GNPs at the central emission wavelength of the embedded 1L-MoS₂ crystals by measuring its For comparison, we show the averaged white light reflectivity spectrum (blue curve) of two separate GNPs deposited on a bare region of the coated substrate. The GNPs exhibits an electric dipolar resonance [...] provides sufficient spectral overlap confirming the resonant character of the deposited GNPs at the central emission wavelength of the embedded 1L-MoS₂.

27. In line 511, depended should be change by dependent.

We have corrected this typo.

28. In eq.(4), the authors do an integral as a function of $d\xi d\eta$ but no details are provided about these two variables or the meaning of the integral. This is a methods subsection, so more details could be provided to make it easier for the reader to understand what they did.

In our analysis, we introduced the variables ξ and η to denote the spatial coordinates within each simulation, as distinct from the beam position coordinates. These variables represent the spatial grid over which the near-field intensities are computed. We hope that the following explanation will enhance the clarity of the Methods section.

In case of the confocal scanning configuration, the local exciton densities $n_{K/K'}(r, x, y)$

become ~~depende~~ dependent on *the beam displacement from the projected GNP's center r . the position of the excitation beam relative to the GNP.* In the experiment, we capture the PL signal that passes through the circular detection polarizer by integrating over a spot on the camera for each beam position. Likewise, in simulations we obtain σ^\pm PL intensities by integrating over the corresponding farfield intensity distributions for each r . Therefore, the detected σ^\pm PL intensities at the distance r from the GNP center becomes

$$\tilde{\mathcal{I}}_{\sigma^\pm}(r) \propto \iint n_K(r, \xi, \eta) \cdot \mathcal{I}_{\sigma^\pm}^K(\xi, \eta) + n_{K'}(r, \xi, \eta) \cdot \mathcal{I}_{\sigma^\pm}^{K'}(\xi, \eta) d\xi d\eta, \quad (2)$$

where ξ and η are two spatial coordinates used in the simulations.

29. Maybe this is trivial and should be clear to anybody reading the paper, but for whatever reason it is not clear to me. Also, I do not think that it is very relevant, but just letting the authors know that I am not sure of what their equation (3) does to the field simulations obtained with a plane wave. I didn't understand it in line 329, and I did still not understand it in Methods. Maybe this could be explained a bit better?

Thank you for raising this point. Equation (3) represents the effect of applying a Gaussian point-spread function, which we use as a simple model to account for the finite optical resolution in our system. Essentially, the point-spread-function is convoluted with the near-field intensity distribution, simulating the blurring effect that occurs due to the limited resolution of the optical setup.

We recognize that a Gaussian point-spread function is a common approximation and may not be ideal for all applications (11), but in our case, it provides a sufficiently accurate representation of the optical resolution's impact. We hope this clarification helps to explain our analysis.

Now, moving to the SI. So the numeration of the lines starts at 1 again.

1. In lines 7-8, it is mentioned that the polarization state is evaluated in position 1. Is the polarization evaluated there, or also modified? That is, how do the authors make sure that the polarization before their objective lens is CP? In their Fig.S1 they do not show any polarization optics elements to do that. Also, how do the authors measure polarization? With a polarimeter? With QWPs and linear polarizers and a camera/powermeter?

Polarization is only measured at this position, not modified. We do not show the polarization optics in our sketch because we insert it only during the alignment procedure and then remove. The quality of circular polarization is assessed with a help of a linear polarizer and a powermeter.

2. In Fig. S2(a), I think that the scale is wrong right? Maybe it is 5 μm , but not 2.
Thank you for the careful check. Yes, it should be 5 μm and we have corrected the scale accordingly.

3. In Fig.S2(a), I read what the authors replied to me, but don't they think that this PL image is weird? Ok, using CVD can create differences in photo-luminescence. That is fine. But the differences in Fig.S2(a) seem to be huge, from like 0.2 to 1. Did they observe this in all the other flakes? If they did, could they just write some sentences in the main text and/or in the SI about it?

Thank you for your comment. Variations in PL distribution across different flakes are common in CVD-grown samples and can arise due to factors such as specific growth

conditions or the presence of strain. (12) Indeed, we have observed similar variations in other flakes. Importantly, however, note that our spatially and polarization resolved PL measurements allow us to clearly distinguish between (1) natural variations in the PL intensity across the bare flake due to factors like growth conditions or strain and (2) local modifications of the PL intensity correlating with the positions of deposited GNPs.

4. In lines 155-156, in my opinion, the sentence It is evident that the discussed in the main text ring-like pattern below the GNP emerges at [...] is confusing. I would re-write it.

We have modified this sentence as follows: *A ring-like intensity pattern forms near the GNP at distances less than 50 nm, while at larger distances, the intensity distribution resembles that of a Gaussian beam.*

Again, sorry for the long review. But I hope that these comments will help the authors to improve their work.

Reviewer 2

The authors have addressed my concerns for the most part and increased the clarity and accuracy of the manuscript. There are a few manuscript modifications that could be tweaked for extra accuracy:

1. Two issues with the implied link between superchiral fields and valley polarization are still present in the rephrased introduction, which now reads: ‘In the field of chiral plasmonics, the modulation of the valley-pseudospin is commonly discussed on the basis of chiral Purcell enhancement (15) involving superchiral nearfields. (16) While such chiral metamaterials have been proven to selectively modulate the valley dynamics and can lead to a sizable DOCP from 2D-TMDs even at room temperature,(17) a fundamental challenge arises when transferring these concepts to valleytronics.’ Firstly, even though it has been used a few times in papers, it is not correct to talk about chiral Purcell effect for valley polarization: reference 15 does not deal with valley polarization but with spontaneous emission from chiral quantum emitters. Secondly, phrasing the problem with applying superchiral fields to valleytronics as an issue only due to the use of chiral nanostructures instead of achiral ones is not correct: the reality is that there is no experimental proof that valley polarization should respond to superchiral fields at all. In fact, the evidence presented in this manuscript proves that there is no reason to expect any relation between them. Therefore, I find that the implied connection between superchirality and valley polarization is highly problematic as it goes against the conclusions of the article.

We agree with the reviewer that no sufficient experimental proof for a direct coupling of superchiral fields to the valley DOF is provided by the previously cited references. Hence, we have revised this section as follows:

~~In the field of chiral plasmonics, the modulation of the valley pseudospin is commonly discussed on the basis of chiral Purcell enhancement^[15] involving superchiral nearfields.^[16] While such chiral metamaterials have been proven to selectively modulate the valley dynamics and can lead to a sizable DOCP from 2D-TMDs even at room temperature,^[17] a fundamental challenge arises when transferring these concepts to valleytronics. The chiral asymmetry permits only coupling to interacting objects of the same handedness (e.g. chiral molecules). In 2D-TMDs, however, the handedness of chiral valley excitons is defined by the spin angular momentum of excitation^[18] which can take two possible states (\pm). Consequently, nanostructures with equal responses to a valley excitonic state and its mirror image, namely achiral nanostructures, emerge as favorable choice for valley-based information processing. For~~

example, chiral plasmonic nanostructures and metasurfaces have been shown to favour one emission helicity over the other, suggesting a potential pathway to achieve room-temperature valley polarization by making use of superchiral nearfields.^{[15][16]} However, the direct link between valley-selective excitation of the material and the observed farfield PL polarization contrast remains unclear, and definitive proof that valley polarization can respond to superchiral fields is still lacking.

In parallel, achiral nanostructures characterized by symmetric responses to the emission from both valleys emerged as promising tools for valley-based information processing.

2. Regarding excitation of valley polarization with in-plane field components only, the original version said '... only the tangential components of external fields will contribute to the excitation of the material (33)'. The revised version says '... refers to the in-plane components of the external fields.' avoiding any mention of its relevance to the problem at hand and removing the reference. I think it would be more accurate to say, closer to the original phrasing, that 'mostly the in-plane components of the external fields contribute to the excitation of the material (33)'.

We have revised the text as recommended by the reviewer and put back the reference to Raziman *et al.*, *ACS Photonics* 2019 6 (10), 2583-2589. We agree that this reference is highly relevant here, as it establishes the connection between the intensity of the in-plane field components and the resulting exciton density. See below the modified version

In 1L-MoS₂, the out-of-plane contributions from spin-forbidden dark excitons are negligible without strong external magnetic fields as shown by Robert et al.[34]. Therefore, mostly the in-plane components of the external fields, denoted by the superscript ||, contribute to the excitation of the material.^[33] In equilibrium the local exciton densities $n_K(x,y)$ and $n_{K'}(x,y)$ will be proportional to $I_{\sigma^+}^{\parallel}(x,y)$ and $I_{\sigma^-}^{\parallel}(x,y)$, respectively ~~where the superscript || refers to the in-plane components of the external field.~~

3. For reproducibility, it is good practice to indicate supplier and model numbers for optical components that can introduce measurement artefacts. Please provide that information for the non-polarising beam splitter, mirror and objective.

We thank the reviewer for this suggestions. In order to improve our manuscript in terms of reproducibility of the polarization resolved measurements, we have added the missing information about the optical components to the Methods section of the manuscript:

Optical experiments were conducted at cryogenic temperatures ($T = (3.8 \pm 0.1)$ K) using a closed-cycle helium cryostation (Montana Instruments s50) equipped with a high numerical aperture optical access (400x/0.9NA Zeiss 422392-9900-000, EC Epiplan-Neofluar 100x/0.90 DIC Vak objective) in reflection geometry. ~~The~~ A non-polarizing 30(R):70(T) plate beam splitter (Chroma) was utilized for all our cryogenic measurements. ~~The~~ A sketch of our optical setup as well as ~~the~~ a detailed characterization of ~~the polarization~~ polarizing effects of the utilized optical components are provided in Sec. S1 of the Supporting Information. [...] The polarization of the excitation beam was prepared by a linear polarizer (Thorlabs LPVISC050-MP2) and quarter-wave phase plate (Thorlabs WPMQ05M-633) by monitoring the degree of circular polarization of the collimated excitation beam before entering the objective. In detection, the polarization was analyzed by a combination of super-achromatic quarter-wave phase plate (Thorlabs SAQWP05M-700) and a linear polarizer (Thorlabs LPVIS100-MP2).

as well as to section S1 of the Supporting Information: Starting from line 5 of the SI: ~~In this setup, the incoming laser beam is first deflected~~ *initially is passed through a linear polar-*

izer (LP, Thorlabs LPVISC050-MP2) and a quarter-wave plate (QWP, Thorlabs WPMQ05M-633), then is reflected by a non-polarizing 30(R):70(T) plate beam splitter (BS, Chroma) and ~~then directed~~ subsequently is focused onto the sample inside the cryostat via an objective lens (Zeiss 422392-9900-000, EC Epiplan-Neofluar 100x/0.90 DIC Vak objective) inside the vacuum chamber of the cryostat. The circular polarization state of the laser is routinely evaluated immediately after the BS, at a point marked as position 1 in Figure S1. The PL emitted from the sample is recollected by the same objective, transmitted through the BS, and then redirected by a silver mirror (Thorlabs PF10-03-P01). Subsequently, the PL is analysed by a superachromatic QWP (SAQWP, Thorlabs SAQWP05M-700) and a LP (Thorlabs LPVIS100-MP2).

Reviewer 3

Based on the extensive feedback from referee one and two (and my short comments), the authors have revised their manuscript extensively. On a side note, referee comments and replies were an educational reading. I still cannot judge novelty from the 2D materials perspective, but this left aside, the present manuscript represents an accessible, educational description on the author's studies of coupling plasms to the valley degree of freedom in 2D semiconductors. I can imagine that content and presentation will receive positive response from the readership of Nature Communications and support publication as is.

We thank the reviewer for the positive assessment of our revised manuscript.

References

- [1] P. Dey, J. Paul, Z. Wang, C. Stevens, C. Liu, A. Romero, J. Shan, D. Hilton, and D. Karauskaj, "Optical coherence in atomic-monolayer transition-metal dichalcogenides limited by electron-phonon interactions," *Phys. review letters* **116**, 127402 (2016).
- [2] K. F. Mak, K. He, J. Shan, and T. F. Heinz, "Control of valley polarization in monolayer MoS₂ by optical helicity," *Nat. nanotechnology* **7**, 494–498 (2012).
- [3] H. Zeng, J. Dai, W. Yao, D. Xiao, and X. Cui, "Valley polarization in MoS₂ monolayers by optical pumping," *Nat. nanotechnology* **7**, 490–493 (2012).
- [4] G. Kioseoglou, A. Hanbicki, M. Currie, A. Friedman, D. Gunlycke, and B. Jonker, "Valley polarization and intervalley scattering in monolayer MoS₂," *Appl. Phys. Lett.* **101** (2012).
- [5] A. V. Kolobov and J. Tominaga, *Two-dimensional transition-metal dichalcogenides*, vol. 239 (Springer, 2016).
- [6] R. Frindt, "Single crystals of MoS₂ several molecular layers thick," *J. Appl. Phys.* **37**, 1928–1929 (1966).
- [7] C. Humbert, O. Pluchery, E. Lacaze, A. Tadjeddine, and B. Busson, "Optical spectroscopy of functionalized gold nanoparticles assemblies as a function of the surface coverage," *Gold Bull.* **46**, 299–309 (2013).
- [8] L. Bossard-Giannesini, H. Cruguel, E. Lacaze, and O. Pluchery, "Plasmonic properties of gold nanoparticles on silicon substrates: Understanding fano-like spectra observed in reflection," *Appl. Phys. Lett.* **109** (2016).
- [9] M. Kaniber, K. Schraml, A. Regler, J. Bartl, G. Glashagen, F. Flassig, J. Wierzbowski, and J. J. Finley, "Surface plasmon resonance spectroscopy of single bowtie nano-antennas using a differential reflectivity method," *Sci. Reports* **6** (2016).

- [10] J. U. Esparza, A. R. Dhawan, R. Salas-Montiel, W. D. de Marcillac, J. M. Frigerio, B. Galias, and A. Maître, “Differential reflectivity spectroscopy on single patch nanoantennas,” *Appl. Phys. Lett.* **117** (2020).
- [11] S. Stallinga and B. Rieger, “Accuracy of the gaussian point spread function model in 2d localization microscopy,” *Opt. express* **18**, 24461–24476 (2010).
- [12] S. Kataria, S. Wagner, T. Cusati, A. Fortunelli, G. Iannaccone, H. Pandey, G. Fiori, and M. C. Lemme, “Growth-induced strain in chemical vapor deposited monolayer mos2: Experimental and theoretical investigation,” *Adv. Mater. Interfaces* **4** (2017).

Point by point response to the Reviewers: Influence of resonant plasmonic nanoparticles on optically accessing the valley degree of freedom in 2D semiconductors

**T. Bucher, Z. Fedorova, M. Abasifard, R. Mupparapu, M. J. Wurdack,
E. Najafidehaghani, Z. Gan, H. Knopf, A. George, F. Eilenberger, T.
Pertsch, A. Turchanin, and I. Staude**

E-mail: tobias.bucher@uni-jena.de and zlata.fedorova@uni-jena.de

We again thank all the reviewers for thorough reading and their constructive comments. We have read all reviewer's comments carefully and answered them point-by-point below. The color convention in our response remains the same as in the previous revision stage. Namely, the original reviewer comments in **blue**, the corresponding answers in **black**, and the changes to the manuscript in **red** where the struck out parts were removed and the italic parts were newly added. Where appropriate, page and line numbers are given which refer to the respective position in the originally submitted manuscript. For the revised version please refer to the file with tracked changes.

Reviewer 1, Xavier Zambrana-Puyalto

In the revised version of *Influence of resonant plasmonic nanoparticles on optically accessing the valley degree of freedom in 2D semiconductors*, Tobias Bucher *et al.* have properly addressed all the comments from the referees. I have a few comments, mostly regarding their new section S.7. But other than that, I do not think that I need to further review their work. So I am happy for it to get published.

We thank the reviewer for the positive evaluation and the careful review that helped greatly to improve the quality of our manuscript.

- In the lines 461-464 of the main text, the authors mention that they assume that the emissions from each point in space are temporally incoherent. Also, in the rebuttal letter, they comment that they have assumed the light emitted by the TMD flake to be incoherent both spatially and temporally. However, in Fig. S8 they show some simulations where the DOP is 1 for all points of space except for the points close to the GNP. Does it make sense to the authors that a spatial and temporal incoherent emission yields a DOP= 1? As mentioned in my previous review, I have not worked much with incoherent light sources, so maybe I am missing something here. But I would expect that the emission of a completely incoherent emitter yields DOP= 0, or at least DOP \neq 0. I think that it would be good that the paper clarified this.

We would like to give two examples which might clarify how we can observe an incoherent ensemble of emitters and a non-zero degree of polarization (DOP) of emission at the same time:

1. Let us consider a thermal source emitting incoherent radiation, which can be modeled as an ensemble of uncorrelated linear dipoles. The emission from this source is both temporally and spatially incoherent, and the orientation of the dipoles is random, resulting in no defined state of polarization over time, i.e., DOP = 0. However,

if we pass the emitted light through a perfect linear polarizer, the degree of polarization becomes 1, despite the fact that the coherence properties of the light remain unchanged, provided the ensemble of dipoles is sufficiently large and randomized.

2. Alternatively, we could imagine selecting a subset of emitters whose dipole moments are aligned along a single axis. Even though the direction of the dipoles is fixed, their oscillation phases are random, yielding spatial incoherence. Since the emitter lifetimes are much shorter than the detection time, the emitted light is temporally incoherent as well. Despite the incoherence, the degree of polarization in this case can also approach 1.

In the context of 1L-TMDs, the material's symmetry leads to valley-contrasting optical selection rules due to angular momentum conservation. As a result, the emitted light possesses angular momentum that is determined by the valley index and exciton momentum, causing it to be predominantly circularly polarized. Nevertheless, the photoluminescence remains incoherent in both space and time. The degree of polarization in this case is close to but not exactly 1, as intervalley scattering, which is discussed in detail in the main manuscript, reduces it.

- In the Numerical simulations section (and in Section S.7), I believe that the notation that the authors use could lead to confusion. Maybe the notation is clear if you have a Lumerical FDTD simulation in front of you, but I guess that this will not be the case for most of readers such as myself. I would say that it is a non-conventional notation, where for example a unit vector can have a modulus which is different from 1. I have checked reference 42, and maybe now it is a bit clearer. But even then, reference 42 seems to show some coordinate transformations where \mathbf{u} has three components, whereas in the text, it only has two. And then, in the text we are presented with the three polarization components in spherical coordinates $\{E_\theta, E_\phi, E_r\}$, which depend on r and \mathbf{u} . Could it be that they are referring to the field emitted by a dipole which is displaced at a distance r from the center of symmetry of the problem and whose dipolar moment has a certain orientation on the plane of the TMD given by \mathbf{u} ? Or maybe they compute $\{E_\theta, E_\phi, E_r\}$ for all possible \mathbf{u} 's, meaning all possible emission directions, as if it was a plane wave decomposition? But then I don't understand why it would only have two components (or dimensions) instead of 3. Anyway, I find their notation quite confusing, and I think that any reader would benefit from a better explanation of what they are exactly doing. Specially, cause then we look at Figure S8(a) and we see a plot which looks as if \mathbf{u} was related to the angle of a cone, but then their little blue E_θ, E_ϕ seem to indicate something else? It is important to define exactly what all these components mean, as they are used in other formulas, such as $E_{\sigma^\pm} = E_\theta \pm iE_\phi$. Anyway, I understand that this is an experimental paper, but I think that the authors should pay more attention at the notation and formulas that they use. In particular, as I think that I mentioned in some of my previous reviews, it is not very clear when they are talking about polarization in real space:

$$\mathbf{E}(\mathbf{r}) = E_\theta(\mathbf{r})\hat{\theta} + E_\phi(\mathbf{r})\hat{\phi} + E_r(\mathbf{r})\hat{\mathbf{r}} = E_x(\mathbf{r})\hat{\mathbf{x}} + E_y(\mathbf{r})\hat{\mathbf{y}} + E_z(\mathbf{r})\hat{\mathbf{z}} \quad (1)$$

with $\{\hat{\theta}, \hat{\phi}, \hat{\mathbf{r}}\}$ or $\{\hat{\mathbf{x}}, \hat{\mathbf{y}}, \hat{\mathbf{z}}\}$ being the polarization vectors; or they are talking about polarization in momentum space:

$$\mathbf{E}(\mathbf{r}) = \int d\mathbf{k} [E_{\theta_k}(\mathbf{k})\hat{\theta}_k + E_{\phi_k}(\mathbf{k})\hat{\phi}_k] e^{i\mathbf{k}\cdot\mathbf{r}} \quad (2)$$

with $\{\hat{\theta}_k, \hat{\phi}_k\}$ being the polarization vectors in the momentum space, which can be combined as $\mathbf{e}_\pm = \hat{\phi}_k \pm \hat{\theta}_k$ to yield the two helicity vectors \mathbf{e}_\pm . My feeling is that my confusion with this \mathbf{u} notation might have to do with the fact that it is expressed in a way that looks like a real space representation, but they might be talking about a momentum space representation.

We thank the reviewer for the careful read of the newly added Sec. S7. Following the questions raised by the reviewer, we believe the confusion results mostly from the following aspects:

1. Indeed, the directional unit vector $\mathbf{u} = (u_x, u_y, u_z)^T$ has three components. However, when extracting the farfield using Lumerical's built-in function, it is typically represented in dependence on the in-plane components of the unit vector u_x and u_y . The out-of-plane component, u_z , is not treated as an independent variable since $u_z = \sqrt{1 - u_x^2 - u_y^2}$. We acknowledge that this explanation was missing, therefore, we have now added a proper definition of \mathbf{u} in spherical coordinates to the main text, Methods section $\mathbf{u}(\theta, \phi) = (\sin \theta \cos \phi, \sin \theta \sin \phi, \cos \theta)^T$, $\theta \in [0, \pi], \phi \in [0, 2\pi]$ denotes the emission direction in terms of a direction unit vector as defined in Lumerical FDTD Solutions (1).

Furthermore, we have updated Figure S7 with a new sketch (a) illustrating the relationship between the directional unit vector, and the Cartesian and spherical coordinate systems (see below). Finally, we have rewritten the integration of the farfield

New version of the Figure S7 (a). *Hemispherical surface with a radius of 1 m centered at the GNP (not in scale) on which the farfields are computed. The sketch shows the relation between the used Cartesian $\{x, y, z\}$ and spherical coordinate systems $\{\theta, \phi, r\}$ as well as the direction unit vector \mathbf{u} .*

intensities over all accepted emission angles in terms of spherical coordinates as follows

$$\mathcal{I}_{\sigma^\pm}^K(x_e) \propto \iint \mathcal{I}_{\sigma^\pm}^K(x_e, \mathbf{u}(\theta, \phi)) \sin \theta d\theta d\phi \text{ (see Ref. (1) for details on the farfield integration).}$$

2. Previously, we have used the character r to denote the displacement of the excitation beam in the confocal scheme, the displacement of the emitter and as a spatial coordinate. To avoid such ambiguities, we revised the manuscript as follows: We use a Cartesian coordinate system in the spatial domain of our sample including nearfield results. The displacements of the focused laser beam and the emit-

ter are defined along the x -axis, and their corresponding offsets are denoted by x_b and x_e , respectively. These offsets act as parameters in quantities such as the valley exciton densities $n_{K/K'}(x_b, x, y)$ or the farfield components of a single emitter $\{E_\theta(x_e, \mathbf{u}), E_\phi(x_e, \mathbf{u}), E_r(x_e, \mathbf{u})\}$. In SI Section S.7, where we study the linear polarization properties of the system, we introduce arbitrary positions for the emitter and beam as (x_e, y_e) and (x_b, y_b) , respectively. The symbol r now exclusively denotes the radial component in the spherical coordinate system used for the farfields. All the instances of a notation change are marked in the revised manuscript.

3. In our emission calculations we are only concerned with the farfield results obtained from the GNP system with a rotating dipole emitter placed at position $\mathbf{r} = (x_e, 0, 0)$. In the farfield, the position dependence reduces to an overall phase factor which does not influence the intensity observable in our experiments. Hence, the two descriptions for the spherical unit vectors as discussed by the reviewer, $\{\hat{\theta}, \hat{\phi}, \hat{\mathbf{r}}\}$ and $\{\hat{\theta}_k, \hat{\phi}_k\}$, will be identical as the radial field component vanishes, i.e. $E_r(\mathbf{r}) = 0$.
- The authors mention in point 13 of their rebuttal letter that *To clarify, the σ^+ polarization refers to the polarization of the laser beam that actually pumps the sample. This polarization is measured before the objective, as indicated at position (1) in Fig. S1 of the SI.[..]*. This is incorrect. When you focus a Gaussian beam with a σ^+ polarization with a microscope objective with NA= 0.9, you get some intensity in the other two polarization components σ^- and $\hat{\mathbf{z}}$. This is shown in [1], Fig.3. The same effect is shown in Fig.3.10 of [2] for a linearly polarized beam which is tightly focused. Of course, if the focused beam (with three polarization components) is reflected and then it is re-collimated by the same microscope objective, you obtain the same σ^+ polarization (or linear) after the objective. But that does not mean that this is the polarization that you have in the focal plane. I think that this should be specified in the article, so that it is clear that they do not control the polarization state at the focal plane where the 2D material is, but rather before the microscope objective.

As fairly pointed out by the reviewer, in experiments we do not have direct control over the polarization state of the excitation beam in the focal plane. Importantly for this study, however, is the fact that the absorption by 1L-MoS₂ is predominantly related to the in-plane components of the external electromagnetic field. Hence, we numerically calculated the 2D-DOCP of a tightly focused Gaussian beam in the focal plane as shown in Sec. S6 of the Supporting Information. For the in-plane field components we find the 2D-DOCP to be nearly 1 across the whole focal plane such that the influence of σ^- polarized components is negligible. We stated this point in the Methods section of the main text as follows:

[...] Note that in experiments we do not have direct access to the polarization state of the excitation beam in the focal plane. Hence, we numerically calculated the 2D-DOCP for a tightly focused Gaussian beam (see. Sec. S6 of the Supporting Information) which is nearly 1. Note further that the out-of-plane component for a tightly focused beam is non-zero, however, it does not contribute significantly to the absorption by 1L-MoS₂ as discussed in the Results section.[...]

- In the first sentence of section S.7 of the SI, the authors make reference to section S3. I think that they meant S2.

We have corrected the typo such that in the revised version we properly refer to Sec. S2.

- In lines 197 and 200 of SI, the authors write that ϕ and β are calculated as 'atan2'. I would just write 'atan', as 'atan2' is just typically the name of a numerical implementation of the atan function.

There is an important difference between writing $\text{atan}\left(\frac{x}{y}\right)$ and $\text{atan2}(y,x)$. The first one is a function of one variable giving the principal value of the arctangent in the range $(-\pi/2, \pi/2)$, while the latter is a two-variable function that considers the signs of both y and x to determine the correct quadrant of the angle in $(-\pi, \pi]$. As an alternative to 'atan2', one can find in the literature the usage of an argument of a complex number, e.g. $\text{atan2}(y,x) = \text{Arg}(x + iy)$. However, for clarity, we have decided to stick with $\text{atan2}(y,x)$ as it is widely used in several programming languages and unambiguous. To the SI we have added the following clarification:

... by rotating the coordinate system at an angle $\phi_e = \text{atan2}(y_e, x_e)$, with $\text{atan2}(y,x)$ denoting the four-quadrant inverse tangent of $\frac{y}{x}$.

References

- [1] A. Canada, "Ftdt product reference manual," (2024).

Review of NCOMMS-24-13553-T

Xavier Zambrana-Puyalto

In their article *Influence of resonant plasmonic nanoparticles on optically accessing the valley degree of freedom in 2D semiconductors*, Tobias Bucher *et al.* measure the effect that a gold nanoparticle (GNP) has on the degree of circular polarization (DOCP) of the photoluminescence of a monolayer of MoS₂. The authors measure the emitted DOCP of a 1L – MoS₂ flake upon two different illumination schemes: i) a tightly focused circularly polarized Gaussian beam, and ii) a circularly polarized plane wave. They observe that the DOCP of the 1L – MoS₂ emission is completely determined by the handedness of the incident circular polarization (CP). At the same time, the authors observe that the DOCP is drastically reduced in the areas where GNP have been drop-casted on top of the 1L – MoS₂ (which is shielded by a 15 nm layer of silica). After presenting their experimental results, the authors build a physical model that reproduces their experimental findings. Their model takes into account the role of the GNPs in exciting the 1L – MoS₂, as well as in tailoring its emission. Finally, the authors compare their model with the experimental data (Fig. 6), and obtain a good agreement, thus validating their theoretical interpretation.

In general terms, I think that the article is correct, yet I have found it quite confusing at times. I needed to read it multiple times in order to make sure that I understood it properly. I think that part of my confusion stems from the use that the authors do of certain words or concepts which turn out to be crucial to understand their light-matter interaction model. Next I list some of the concepts that, in my opinion, should be clarified in the next version of the manuscript, so that their physical model becomes also clearer.

- Depolarization. The article revolves around this concept, but it is not clear to me what the authors mean. Between lines 136-142, the authors state "*the observed reduction in the DOCP is equivalent to a reduction of the total degree of polarization...*". Then, in Fig.1 of the supplementary information (SI), the three Stokes parameters of their measurements are given which seem to lead us towards the idea that what is actually happening is that the degree of polarization (DOP) is being reduced. But then, the whole article deals with *changes* in the DOCP, rather than with the decrease in DOP. In my opinion, these points need to be clarified:

- i) What do the authors mean by "depolarization"? If an optical field goes from being CP to being linearly or elliptically polarized, has this field been depolarized? Note that this is what typical polarization optics elements do, e.g. linear polarizers, wave-plates, etc. Microscope objectives also do this, they transform the polarization of a beam. But they not introduce any randomness or decoherence effects that are the typical cause of unpolarized light [1, Ch.12]. In my opinion, this is what "depolarization" is about, the

decrease of DOP due to unpolarized light or random phases. But of course, I also think that it is fine if the authors use it in a different way, I would just like them to clearly state what they mean.

- ii) Do the authors believe that what their hybrid system is doing is changing the DOP and the DOCP at the same time? Or just the DOCP? Or maybe just the DOP? Because, as far as I can see, the whole physical model of the article is based on the assumption that DOP is constant (and equal to 1) at all times. This is what I actually expect, by the way, as their illumination is a HeNe laser. But then, as I mentioned before, the authors also state that the DOP is reduced (and they also provide evidence of that), so then we are left with the question - well, if something is making the DOP to be reduced, then what is it? And why is this not being studied or taken into account in order to explain the experimental results?
- Helicity. I am aware that the term helicity is used in various ways in the literature. And of course, I think that it fine that the authors use it in the way that they think is the most correct. But again, I think that they should clearly state what they mean, otherwise it could get confusing for some readers like me. For me, helicity has to do with the handedness (or the sign of CP) of a field in the momentum space [2]. Formally, it is defined as the normalized projection of the total angular momentum (AM) into the direction of the linear momentum, i.e. $\Lambda = \frac{\mathbf{J} \cdot \mathbf{P}}{|\mathbf{P}|}$ [3, p.136]. As a result, an optical field has a helicity +1 when its decomposition into plane waves yields that all the plane waves are left CP [2, p.4]. Instead, I think that the authors use helicity to talk about CP (in real space). Of course, when we only have one plane wave, or we have a paraxial beam, both things coincide. But they do not coincide when we are talking about dipoles, tightly focused beams or near fields. In particular, in lines 165-166, the authors say that *we observe a strong modification of the 2D-DOCP at larger distances even leading to field components of opposite helicity*. I guess that what they mean is that they have decomposed the field in three polarizations on that plane, and they have observed that the polarization changes depending on the point of the plane. But then, later on, the authors use helicity to talk about the rotation direction of an electric dipole: in line 213, they say *rotating mirror dipole of the same helicity*. But the emitted field of a rotating electric dipoles is definitely not CP. Here, I understand that they talk about helicity of emitting electric dipole to talk about the sign of a rotating dipole of the form $\mathbf{p}_{\pm} = \mathbf{p}_x \pm i\mathbf{p}_y$. The authors also mention in lines 204-205 that *By symmetry, the nanoparticle-on-substrate geometry is expected to preserve the helicity of valley-selective chiral emission*. Here, I am not sure of what they mean, as the two helicity meanings that have been previously given could make sense. Do they mean that a single sphere will preserve the CP handedness in the forward direction in the far field, or maybe they mean that the GNP will have an induced electric dipole that will be of the same sign \pm as the emitting rotating dipole? Notice that there are no general conditions for polarization preservation. Cylindrical symmetry is associated to the preservation of AM,

but not polarization. In fact, a rotating electric dipole of the kind $\mathbf{p}_{\pm} = \mathbf{p}_x \pm i\mathbf{p}_y$ emits fields with a well-defined AM= ± 1 [4, 5], and this will certainly be preserved by the substrate-GNP system if the dipole is centred. Instead, the condition for helicity (understood as handedness in the momentum space) preservation is duality symmetry [6], which has to do with the material properties of the matter, and not cylindrical symmetry.

- Chirality. Chirality is a term that tends to have a more established meaning. Typically, we say that an object is chiral when you cannot superimpose it with any of its mirror images. As the authors write in line 66, a single GNP is achiral. I agree with that. But then, chirality is mostly used in the article to refer to the emission of 1L – MoS₂. For example, in line 67 ”*specific chiral emission from a monolayer of 1L – MoS₂*”, or in line 117 ”*valley-specific chiral emission*”. When it comes to optical fields, chirality can be used as helicity (handedness in momentum space), consequently there are only two possible handednesses that an optical field can have, + or -. Or left and right. It is well-known that a CP plane wave is chiral, and it has two different handednesses, left and right. But things get more complicated when we have optical fields that are superpositions of plane waves, such as the emission of dipole. When I read that the emission of 1L – MoS₂ is chiral, what I understand is that the emission of 1L – MoS₂ can be decomposed into plane waves, and that all these plane waves have the same handedness. Now, I do not know if this is what they authors meant or not, but this is not the case if we model the emission of 1L – MoS₂ as a rotating electric dipole of the kind $\mathbf{p}_{\pm} = \mathbf{p}_x \pm i\mathbf{p}_y$. In fact, it can be proved that such a dipole is achiral, i.e. that the emitted field can be superimposed with its mirror image. Or in other words, that its field can be decomposed 50-50% into fields with left-right handedness [5, p.3]. I suspect that the reason why this gets confusing is because the authors are only interested in the emission of 1L – MoS₂ in one semi-space. And indeed, if we only consider one semi-space, then we could consider that the field emitted by a \mathbf{p}_{\pm} electric dipole is mostly formed by plane waves with the \pm handedness. I have depicted this in Fig. R1. In Fig. R1a), I show the near-field of a rotating dipole of the kind $\mathbf{p}_{\pm} = \mathbf{p}_x \pm i\mathbf{p}_y$. It is observed that it emits light all over space. Then, in Fig. R1b) and c), I depict the near-field of rotating dipoles (with the same + rotating sign) that emit light only with left (or +) and right (or -) handedness respectively. To create these dipoles, we need to superimpose electric and magnetic rotating dipoles with the same rotating sign +. And then, it is easy to see that the dipole in a) is the addition of the dipoles in b) and c). In fact, this is also plotted by the authors in Fig. 5 of the manuscript, but for the far-field. That is, effectively, a rotating dipole emits light with one handedness in one semispace, and it emits light the other handedness in the other semispace. Consequently, it is achiral but has a well-defined AM equal to \pm [5, p.3].

To wrap up this general comment about concepts/wording - my intention, of course, is not to try to impose any kind of wording, but rather to show the authors that their current wording, which is necessary to understand their physical model, does not seem to be very consistent throughout the

FIG. R1. a) Power emitted by a rotating electric dipole $\mathbf{p}_+ = \mathbf{p}_x + i\mathbf{p}_y$. b) Power emitted by a rotating chiral dipole with left handedness created as a in-phase superposition of rotating electric and magnetic dipoles, i.e. $\mathbf{p}_+ + \mathbf{m}_+$, with \mathbf{m}_+ being a rotating magnetic dipole. c) Power emitted by a rotating chiral dipole with right handedness, i.e. $\mathbf{p}_+ - \mathbf{m}_+$. All plots are done at 632nm, for an area of $1.2 \times 1.2 \mu\text{m}^2$. The black region in the center of the plot is avoided because emitting dipolar fields diverge at the emitting point.

paper and could lead to misunderstandings. As I mentioned, I think that I partially struggled to understand the paper because of this.

Next, I move on to comment some other issues. I find Fig.4 confusing, which is problematic, because it is what summarizes the excitation model of the authors. First of all, for propagating beams, I am used to talking about *transverse* and *longitudinal* polarization components. That is, if a beam is propagating in the z direction, and we decompose it into three polarizations components, e.g. $\mathbf{E} = E_x\hat{\mathbf{x}} + E_y\hat{\mathbf{y}} + E_z\hat{\mathbf{z}}$ or $\mathbf{E} = E_{\sigma^+}\hat{\sigma}^+ + E_z\hat{\mathbf{z}} + E_{\sigma^-}\hat{\sigma}^-$, then I would consider $\hat{\mathbf{z}}$ to be the longitudinal component of polarization, and $\{\hat{\mathbf{x}}, \hat{\mathbf{y}}\}$ or $\{\hat{\sigma}^+, \hat{\sigma}^-\}$ would be the transverse components. The authors use the expression "tangential" components. I believe that they are referring to the $\{\hat{\sigma}^+, \hat{\sigma}^-\}$ components. But then, when I look at the intensity and the DOCP plots, I get very confused. For whoever has worked with nanoparticles and Gaussian beams, I think that it would be shocking to see that the result of this interaction is a doughnut beam. Typically, one would expect something similar to a Gaussian beam, or an Airy-type pattern. And then, because I know that the longitudinal component of a tightly focused Gaussian beam looks quite similar to a doughnut beam, I started wondering if what the authors meant by "tangential" was longitudinal. I had to go and simulate a similar system (a tightly focused Gaussian beam) interacting with a sphere to see what is happening - see Fig. R2a). In Fig. R2a), I plot the total transverse field, i.e. scattered + incoming, on a $x - z$ plane of $1 \times 1 \mu\text{m}^2$. We observe that indeed a vortex beam is formed at the plane at 15 nm from the tip of the sphere (the beam is travelling from negative to positive z 's). But already at 100 nm from the tip of the sphere, the field resembles that of a Gaussian beam. In my opinion, there needs to be a discussion about all this in the revised version of the manuscript. By the way,

FIG. R2. a) Plot of $|E_{\sigma^+}|^2 + |E_{\sigma^-}|^2$ for a $x-z$ plane of $1 \times 1 \mu\text{m}^2$ size. The inset is a $x-y$ plane of $1 \times 1 \mu\text{m}^2$ size at 15 nm of the tip of the sphere. b) DOCP in the same $x-y$ plane as the inset. All plots are done with a CP Gaussian beam with a $\hat{\sigma}^+$ polarization before focusing. The wavelength is $\lambda = 632$ nm. The GNP has a radius of 100 nm.

I also simulated the DOCP in the near-field using the definitions given in the manuscript, but I did not get the same results as the authors - see Fig. R2b). It can be seen that the results differ quite a bit from the ones shown in the paper, but I understand that they simulated a sphere lying on an infinite glass layer, so obviously the result is different. Not sure if it makes sense that the two systems behave so differently - in my plot the DOCP is always positive, and in the plot in Fig. 4 of the paper we see a DOCP going to negative values - but anyway, I cannot say much more about it. What I am especially puzzled with is the DOCP for the plane wave case. It looks as the DOCP plot was just the DOCP of the incoming plane wave and no scattering had been taken into account. Surely the near-field scattered by the GNP will alter the DOCP of the incoming plane wave and will yield some components in the opposite polarization $\hat{\sigma}^-$. Could the authors double-check this? In fact, the image seems to be in contradiction with the text, since in line 160 is mentioned that the GNP leads to a slight reduction of the DOCP even for the plane wave case, but the DOCP seems to be constant and equal to 1 across the plane. This constant DOCP plot is one of the reasons why I am quite confused with Fig. 4, as it made me think that these plots were showing just the incoming beams, and not the incoming beam + scattering.

In the article, it is also mentioned that Fig.3(a) is the result of confocal microscopy. Whereas, I assume that the DOCP shown in Fig. 4(b) is not a confocal measurements, right? That is, to measure DOCP for the plane wave excitation, pictures of the whole flake are taken (under a plane wave, or more accurately a weakly focused Gaussian beam) and the polarization analysis is done on the whole image right? For this case, it is clear to me that the polarization that all the points in

the flake will feel is the same. However, in the confocal case, I am less sure about that, especially if the scan on the sample is done with a scanning laser beam, instead of with a scanning stage. If the sample is scanned with a stage, then all the pixels of the image, corresponding to different positions on the plane of the sample, feel the same polarization distribution, given by the tightly focused Gaussian beam. But I am not sure that this is the case when the scan is done by a scanning laser. Could the authors comment on this?

A little bit related to this, in lines 178-180 we are told about this Gaussian smoothing. Does this just mean that a plane wave is not used, but rather a very weakly focused Gaussian beam?

Also, in Fig.1(a) in the SI - why is there so little photoluminescence coming from the center of the flake? Is this due to photobleaching? In fact, could the authors comment on the power that was used to do the experiment? How does this affect the emission of 1L - MoS₂?

In the sample preparation, it is mentioned that the GNP are drop-casted and spin-coated. What is the purpose of the spin-coating? And does the polymer or resist get removed afterwards?

In line 161 - what does *inducible* mean? I guess that it should be inducible. Also in line 189, there's a typo - calculated instead of calculated.

Some final comments:

- Fig. 6 is, in a way, what "proves" that the modelling done is correct. Yet, I think that very few details are provided so that we can try to reproduce their results. Unless I am mistaken, in order to get Fig. 6, what the authors have done is: i) They have computed the intensities $|E_{\sigma+}(x, y)|^2$ and $|E_{\sigma-}(x, y)|^2$, which are computed as illumination + scattering for both illumination schemes. ii) They have assumed that the emission of 1L - MoS₂ is given by \mathbf{p}_+ and \mathbf{p}_- -like dipoles, whose probability to emit is proportional to the intensities $|E_{\sigma+}(x, y)|^2$ and $|E_{\sigma-}(x, y)|^2$. These probabilities to emit in \mathbf{p}_+ and \mathbf{p}_- states are proportional to what they call local exciton densities $n_K(x, y)$ and $n_{K'}(x, y)$. iii) They have taken the dipolar emission caused by the local exciton densities $n_K(x, y)$ and $n_{K'}(x, y)$ and have convoluted their (x, y) distribution on the plane with Fig. 5(b). Fig 5(b) models the DOCP that a far-field detector measures as a result of a \mathbf{p}_+ emission in a glass substrate located at 15 nm below the sphere contact point with the glass, and at a transverse distance r of that contact point. The authors do not show it, but obviously there must be another counterpart of Fig. 5(b) with an emitting \mathbf{p}_- dipole and negative values of DOCP. Now, my feeling is that the article would be clearer if the reader was guided through this process more thoroughly. For that, the authors would need to show and quantify (when possible) all the intermediate results. The article is relatively long (14 pages), and the section regarding Fig. 6, which verifies all the rest, is only one and a half

pages. Also, we are told about an averaging over GNP sizes for the theoretical curve. But we are not told what sort of sizes they have averaged upon. Also, what about the experimental results? Are they also averaged or they are just the points for one of the GNP? If they are not averaged, does it make sense to average the theoretical results instead of measuring the GNP size and applying some error bars to the measurement and do the simulation with different sizes within the error bars? In summary, I think that more details should be given in this part.

- Finally, after reading the article, I am left a little bit thinking that we have not gotten many insights out of this experiment. The authors have done an experiment and in principle they have managed to reproduce the results with a physical model. That is good work, of course. But what can we get out of this? We have learned that this highly symmetric system does not preserve DOCP (contrary to what some could expect), and we have learned how to model this complex light-matter interaction problem to get good matching with the experimental data. But what is the underlying reason that is making this happen apart from a convolution of effects? If we were to vary the system slightly, should we go through the whole simulation process to predict the outcome? Or can we build some intuition based on these results? For example, what would have happened if the particle was not cylindrically symmetric? Or if the particle did not behave like an electric dipole, but rather as a combination of electric and magnetic? Also, what would it happen if the 1L – MoS₂ flake was at a different distance. As seen in Fig. R2a), the fact that a doughnut beam is obtained at a plane at 15 nm from the contact point of the sphere is a bit of a surprising near-field condition that already disappears at 100 nm from the contact point of the sphere. And it can be checked that if the particle would have been made of TiO₂ (just to name a common material for nanophotonics), the near-field intensity would have been very similar to a Gaussian beam. In fact, one could intuitively think that what is happening here is that the 1L – MoS₂ is located at a precise distance of the GNP such that the intensity on that plane (right below the GNP) is zero for a large area of roughly $100 \times 100 \text{ nm}^2$. And as a result, the DOCP and DOP that the authors measure are 0 basically because there is no emission. Whereas if that near-field intensity distribution was more Gaussian-like, which is what intuitively one would expect, then the values of DOCP and DOP would be much better preserved. I am not saying that this is what is happening exactly, but basically, as the article stands right now, I do not think that we get any clear big picture of what is happening. And thus, we are left with the idea that if we were to change any of the particularities of the experiment that I mentioned above (particle shape or material and distance), that we would need to re-think the whole experiment another time.

I am sorry for the long review. But I hope that these comments will help the authors to improve

their work.

-
- [1] B. E. Saleh and M. C. Teich, *Fundamentals of photonics*, Vol. 22 (Wiley New York, 1991).
 - [2] I. Fernandez-Corbaton, X. Zambrana-Puyalto, and G. Molina-Terriza, Helicity and angular momentum: A symmetry-based framework for the study of light-matter interactions, *Phys. Rev. A* **86**, 042103 (2012).
 - [3] W.-K. Tung, *Group Theory in Physics* (World Scientific, Singapore, 1985).
 - [4] X. Zambrana-Puyalto and N. Bonod, Tailoring the chirality of light emission with spherical si-based antennas, *Nanoscale* **8**, 10441 (2016).
 - [5] A. G. Lampryanidis, X. Zambrana-Puyalto, C. Rockstuhl, and I. Fernandez-Corbaton, Directional coupling of emitters into waveguides: A symmetry perspective, *Laser & Photonics Reviews* **16**, 2000516 (2022).
 - [6] I. Fernandez-Corbaton, X. Zambrana-Puyalto, N. Tischler, X. Vidal, M. L. Juan, and G. Molina-Terriza, Electromagnetic duality symmetry and helicity conservation for the macroscopic maxwell's equations, *Phys. Rev. Lett.* **111**, 060401 (2013).

Review of NCOMMS-24-13553A

Xavier Zambrana-Puyalto

In the revised version of *Influence of resonant plasmonic nanoparticles on optically accessing the valley degree of freedom in 2D semiconductors*, Tobias Bucher *et al.* have given many more details about their experimental and theoretical findings. I think that the article has improved as a result. However, after reading the response to all the referees comments, the revised manuscript and the SI, my feeling is that the article still needs to be polished. In my opinion, the most important issues that need to be addressed are the following:

- The authors mention that the decrease in DOCP that is observed goes hand in hand with a reduction of the DOP. That is, below the GNP, the three Stokes parameters $\{S_1, S_2, S_3\} \approx 0$. However, it seems that the Stokes parameters (especially S_3) are clearly different from 0 at the flake positions where the GNP are not present, thus leading to a $DOP > 0.8$. I find this quite puzzling. It seems as if a simple scattering process (photo-luminescence emitted photons from 1L – MoS₂ interacting with the GNP) makes the DOP of the emitted light go to zero. A $DOP \approx 0$ is typically a sign of incoherent light. Do the authors understand what is going on? I think that this is a crucial point for this article. Because their theoretical model, based on the exciton emission and GNP scattering, assumes that the light is coherent. That is, the rotating dipole model that is used to understand their measurements is a coherent model. As a result, a reduction of DOCP (or a decrease in S_3) necessarily leads to an increase of $\{S_1, S_2\}$, as the DOP must remain 1. However, the authors do not measure this. This makes me think that the interpretation of their experimental results might be missing something. I am not used to working with the polarization of incoherent or partially coherent light. So I might get this wrong. But linear scattering of coherent light does not change its coherence. This is why I cannot understand why a material that emits with $DOP > 0.8$ would turn into a material that emits with $DOP \approx 0$ upon interaction with a GNP. And then, I thought that maybe what is happening is that basically the intensity that is being measured is almost 0 - due to the near-field created below the GNP, or due to the confocal nature of their measurement. But looking at Fig. S4 or Fig. 3, this does not seem to be the case. So I am left wondering what is going on. I think that the authors should try to address this point. As the whole motivation of the paper revolves around the fact that $DOCP \approx 0$, but in fact, the authors also measure that $DOP \approx 0$, but kind of decided not to pay much attention to that.
- I think that Referee 2 was right in pointing out that there were some issues with Fig 1 of the SI. The authors have created a new section S.1 in the SI, which I like. I think that it is a good addition to the paper. But I am wondering, if the authors are trying to measure the polarization properties, why do they base their measurements and analysis on the detection of some Stokes

parameters that mix orthogonal states? I am referring here to the results presented in the table of S.1.3. In my opinion, the authors should have added some extra quarter and half wave-plates to compensate for the polarization modification caused by all the optical elements after the microscope objective. That is, typically they should make sure that Stokes parameters of the laser are $\{S_1, S_2, S_3\} = \{0, 0, \pm 1\}$ both before focusing (position 1) and after collection. As mentioned in S.1.1, non-polarizing beam-splitters do not tend to affect much the polarization state of beams, therefore this could be measured after the BS as well, in case measuring it in position 1 was not possible (I am talking about the reflected beam). If this was the case, then they should make sure that this is still the case at position 3, i.e. after the mirror. This will not happen, as explained in S.1, since the 45 degree mirror changes polarization. So then, after the mirror, they should add some polarization compensation so that this is still respected. That is, after the compensation elements (probably a system of QWP-HWP-QWP), an incoming laser with a Stokes vector $\{S_1, S_2, S_3\} \approx \{0, 0, \pm 1\}$ should remain invariant for both left and right CP reflected laser beams. And same for linear polarizations. Ideally, the same should hold true for the photo-luminescence wavelengths. But depending on the QWPs and HWPs used, the change of wavelength could affect more or less this orthogonality. This is something that needs to be taken into account. For example, if we look at Fig.S3, and we are using a set-up that does not respect orthogonality, we cannot be sure the polarization contrast between the PL and the laser is a real physical effect or it is just the result of different chromatic effects given by the optical elements before the Stokes parameters measurement took place.

- Related to the previous comment, in Fig. S3 we observe that the PL emission for left and right CP illumination has very similar values for longer wavelengths. Again, is this a detection (chromatic) issue, or is this supposed to be like that? Is this one of the reasons why the DOPC is 0.8 for a bare 1L – MoS₂ flake? In theory it should almost be 1, right?

Apart from this, I also have a number of comments:

1. In lines 16-17, it is stated that *that the light emitted from the nanoparticle position*. I understand what the authors mean, but to avoid any misunderstandings it would be better to write that the nanoparticle scatters light. Emission, in this article, carries the meaning of photo-luminescence, and thus absorbing light at one frequency and emitting at a different one. This is not what the nanoparticle does.
2. In line 21, I would change *strongly reduces* to 'is strongly reduced'.
3. In lines 18-23 - unless I do not properly understand these two sentences - the authors make a reference to their near-field excitation + scattering model. But as I mentioned before, I do not think that this models predicts any kind of depolarization - understanding depolarization as a reduction of DOP. The model predicts a reduction of the DOCP, but it says nothing about the reduction of DOP. I think that the authors should try to explain the reduction of DOP, but

if they don't, then they should at least re-phrase these sentences to make it clear that their efforts are focused on the DOCP decrease (and not in the DOP reduction).

4. In lines 22-23, I do not exactly know what the authors mean
5. In line 33, maybe the sentence *can form that may be used* could be changed to 'can form and may be used'.
6. In line 59, this has already been mentioned. It might be a bit confusing to call the valley-excitons chiral or to talk about the handedness of their emission. They emit with a well-defined total AM, and as a result, they have (approximately) one handedness in one semi-space, and the opposite handedness in the other semi-space.
7. In line 93, it is the only place in the article where the ellipticity of the GNP. Was this measured? Or it is just an assumption that they cannot be perfect spheres?
8. In line 94, instead of *a predominating*, I would write something like 'predominantly'.
9. In line 114 and in many other places we are told about the 1L – MoS₂ crystals. Isn't this a bit of a weird way of calling them? In general, crystals are 3D structures. I would simply call the 1L – MoS₂ layer a '1L – MoS₂ flake'.
10. In line 125-126, I would write see Methods when the differential reflectance $\Delta R/R$ magnitude is introduced. I was not sure what that meant, but then I could find it in Methods.
11. In line 129, shouldn't the *absorption peaks* be 'reflection peaks'?
12. In line 130, maybe it would be good to give a reference that explains what the A- and B-excitons are.
13. In line 148, I guess that this is sort of understood, but the σ^+ polarization is not the polarization that pumps the sample, but rather the polarization of the laser beam before it is focused. Maybe this could be specified by writing ' σ^+ polarization before focusing'.
14. In lines 182, the authors mention that *the excitation rate of carriers in valleys K and K' is proportional to the intensity of the σ^+ and σ^- polarized components of the external fields*. Here, I assume that external fields mean incident field + scattered. However, in line 194-195, it is stated that *Figure 4a shows the resulting total in-plane intensities generated by a focused Gaussian beam (left) and a plane wave (right)*. I am pretty sure that the authors refer to the field that is generated as a superposition of illumination and scattering. But one could get confused, so I think that they should specify that they are talking about their external fields, which are illumination + scattered field.

15. In line 204, it is stated that *Notably, this cross-polarized component is slightly larger for the focused beam..* I guess that the comparison is relative. That is, they have normalized the maximum of the $I_{\sigma_+}^{\parallel}$ to 0.8 (or 1) for both illuminations and they are comparing how the other components is respect to the dominant one. However, the sentence could possibly mislead somebody to believe that the comparison is in absolute terms, i.e. in absolute emitted power. But that wouldn't make sense, as there will be much more power emitted in the Gaussian illumination case, if both Gaussian beam and plane wave are carrying the same power. Anyway, I think that it would be good to make sure that this is understood.
16. In Fig. 4, I think that it is a bit confusing because if one looks at the intensity plots of (a), you get the impression that there is no intensity right in the centre of symmetry. However, we see that the intensity is not 0 in (b). Then, in Fig. S6 this is clear. But maybe a reference to Fig. S6 could be given so that one is not under the impression that there is a 0 of intensity in the centre?
17. In line 226, it says Figure 4b, but I think that it should be Figure 4c.
18. In line 229, we are told that $\text{DOPC} = 0.72 \pm 0.02$ when there are no nanoparticles. This is related to what I already mentioned before in the beginning of the review, but why isn't this value almost 1? Maybe a comment should be written about it?
19. In lines 230-231, we are told about the prediction of Δ_{DOCP} from their near-field simulations. But I am not sure if we have been told what these Δ_{DOCP} simulations are computed, once we have obtained the near-fields. In methods, some details are given about how to compute the whole model, but I do not think that we are told how this intermediate step is computed.
20. In lines 239-240, I do not fully understand why the authors do this integral and where they use it. I think that more details should be given about it.
21. In lines 243-244, I do not fully understand how these results are obtained. I think that more details should be given about it too.
22. In lines 273-274, it is not accurate that a rotating dipole emits circularly polarized light into different half spaces. It is approximately true in the far-field, though. That is, approximately it emits light of one helicity into one semi-space and the opposite helicity in the other semi-space. Polarization and polarization in the far-field (helicity) are not the same thing. The GNP, when illuminated by a tightly focused Gaussian beam that was CP before focusing, also behaves as a rotating dipole, and clearly does not create a near-field that is circularly polarized, as both CPs and also a $\hat{\mathbf{z}}$ component are present. However, in the far-field, it is approximately true that that its radiation pattern is also similar to the one shown in Fig.5(a).

23. In Figure 5, could the authors mention something about their current normalization? Now the total intensity goes to 1.5, which is a bit unconventional, even if there is no problem to leave it like this provided some details are given.
24. In lines 321-324, I guess that the sentence *Similarly, we have [...] are obtained as [...]* is missing a comment on the fact that this depends on the radial position, right? In particular, when the emitter is far away from the GNP, the DOCP seems to be 1 (Fig.5(b)).
25. In line 389-390, it is stated *a cylindrically symmetric system conserves the circular polarization state in the direction out of the substrate*. Similar comments are spread across the article. I think that generally speaking they are correct. But if read carefully, it seems as if the authors were claiming that cylindrical symmetry is related to preservation of circular polarization. And this is not true, in general. It is true that a cylindrically symmetric scatterer preserves circular polarization in the forward direction, at 0° [1, 2]. And it is also true that it preserves the total AM. But again, as shown by the authors in Fig. S6, a GNP does not preserve circular polarization (especially in the near-field). The sentence would be approximately correct if they talked about the far-field polarization state in the backward semi-space. I say approximately correct because it is not literally true that there is no scattered light with the opposite polarization in the backward semi-space. But it is true that it is negligible. Anyway, I think that the authors need to be more careful with these sort of statements.
26. In the subsection **Optical experiments at cryogenic temperatures** (as well as in the main text), the authors do not mention anything about how they obtained the spectral measurements of the GNP's scattering. Could they give some information about it?
27. In line 511, depended should be change by dependent.
28. In eq.(4), the authors do an integral as a function of $d\xi d\eta$ but no details are provided about these two variables or the meaning of the integral. This is a methods subsection, so more details could be provided to make it easier for the reader to understand what they did.
29. Maybe this is trivial and should be clear to anybody reading the paper, but for whatever reason it is not clear to me. Also, I do not think that it is very relevant, but just letting the authors know that I am not sure of what their equation (3) does to the field simulations obtained with a plane wave. I didn't understand it in line 329, and I did still not understand it in Methods. Maybe this could be explained a bit better?

Now, moving to the SI. So the numeration of the lines starts at 1 again.

1. In lines 7-8, it is mentioned that the polarization state is evaluated in position 1. Is the polarization evaluated there, or also modified? That is, how do the authors make sure that the

polarization before their objective lens is CP? In their Fig.S1 they do not show any polarization optics elements to do that. Also, how do the authors measure polarization? With a polarimeter? With QWPs and linear polarizers and a camera/powermeter?

2. In Fig. S2(a), I think that the scale is wrong right? Maybe it is 5 μm , but not 2.
3. In Fig.S2(a), I read what the authors replied to me, but don't they think that this PL image is weird? Ok, using CVD can create differences in photo-luminescence. That is fine. But the differences in Fig.S2(a) seem to be huge, from like 0.2 to 1. Did they observe this in all the other flakes? If they did, could they just write some sentences in the main text and/or in the SI about it?
4. In lines 155-156, in my opinion, the sentence *It is evident that the discussed in the main text ring-like pattern below the GNP emerges at [..]* is confusing. I would re-write it.

Again, sorry for the long review. But I hope that these comments will help the authors to improve their work.

-
- [1] I. Fernandez-Corbaton, X. Zambrana-Puyalto, N. Tischler, X. Vidal, M. L. Juan, and G. Molina-Terriza, Electromagnetic duality symmetry and helicity conservation for the macroscopic maxwell's equations, *Phys. Rev. Lett.* **111**, 060401 (2013).
 - [2] I. Fernandez-Corbaton, Forward and backward helicity scattering coefficients for systems with discrete rotational symmetry, *Optics express* **21**, 29885 (2013).